# Acetylation of histone H2B marks active enhancers and predicts CBP/p300 target genes

**Takeo Narita** [1,2], **Yoshiki Higashijima**[1,2], **Sinan Kilic** [1], **Tim Liebner** [1], **Jonas Walter** [1] & **Chunaram Choudhary** [1] ✉

Chromatin features are widely used for genome-scale mapping of enhancers. However, discriminating active enhancers from other *cis*-regulatory elements, predicting enhancer strength and identifying their target genes is challenging. Here we establish histone H2B N-terminus multisite lysine acetylation (H2BNTac) as a signature of active enhancers. H2BNTac prominently marks candidate active enhancers and a subset of promoters and discriminates them from ubiquitously active promoters. Two mechanisms underlie the distinct H2BNTac specificity: (1) unlike H3K27ac, H2BNTac is specifically catalyzed by CBP/p300; (2) H2A–H2B, but not H3–H4, are rapidly exchanged through transcription-induced nucleosome remodeling. H2BNTac-positive candidate enhancers show a high validation rate in orthogonal enhancer activity assays and a vast majority of endogenously active enhancers are marked by H2BNTac and H3K27ac. Notably, H2BNTac intensity predicts enhancer strength and outperforms current state-of-the-art models in predicting CBP/p300 target genes. These findings have broad implications for generating fine-grained enhancer maps and modeling CBP/p300-dependent gene regulation.

*Cis*-regulatory enhancers are instrumental in activating signal-induced and developmentally regulated genes. Candidate enhancers are identified using DNase I hypersensitivity (DHS), enhancer RNA (eRNA) transcription, massively parallel reporter assays (MPRA) and enrichment of chromatin marks, such as H3K4me1 and H3K27ac[1–3]. Among these, H3K27ac is widely used, including by the ENCODE, BLUEPRINT and NIH Epigenome Roadmap consortia[4–6].

While H3K27ac and other markers have been very useful, some challenges remain that hamper a deeper understanding of enhancer-dependent gene regulation: (1) distal enhancers are thought to encompass heterogeneous groups and thousands of them apparently lack H3K27ac[7–10]. The fraction of endogenously active enhancers that lack H3K27ac is unclear; (2) H3K27ac poorly discriminates between proximally occurring active enhancers and promoters[3,11,12]; (3) a modest validation rate in orthogonal assays has given the

notion that chromatin marks are poor predictors of endogenously active enhancers[13–15]; and (4) a key goal of mapping enhancers is to estimate their functional impact on gene activation. However, predicting native enhancer strength and enhancer target genes is challenging[16,17].

Previous studies showed that (1) in addition to H3K27ac, CBP/p300 catalyzes H2B N-terminus multisite lysine acetylation (H2BNTac)[18], (2) lysine deacetylase inhibition strongly increases H2BNTac[19], (3) some of the H2BNTac sites occur at a relatively high stoichiometry[20] and (4) CBP/p300 activity kinetically controls enhancer-mediated transcription activation[21]. In this study, we show that H2BNTac sites distinctively mark active enhancers and discriminate them from other candidate *cis*-regulatory elements. H2BNTac-marked regions show a high validation rate in orthogonal enhancer activity assays. Importantly, H2BNTac most accurately defines locus-specific

[1]Department of Proteomics, The Novo Nordisk Foundation Center for Protein Research, Faculty of Health and Medical Sciences, University of Copenhagen, Copenhagen, Denmark. [2]These authors contributed equally: Takeo Narita, Yoshiki Higashijima. ✉e-mail: chuna.choudhary@cpr.ku.dk

CBP/p300 activity and enhancer strength and outperforms H3K27ac in predicting CBP/p300-regulated genes and the degree of their dependency on CBP/p300 activity.

## Results

### Multiple H2BNTac sites occupy the same genomic regions

H2BNTac sites are similarly regulated by CBP/p300[18] (Supplementary Fig. 1a), yet the reported genome occupancy patterns of H2BNTac sites are dissimilar from each other[22,23] (Supplementary Note 1). To resolve this conundrum, we systematically compared H3K27ac and H2BNTac genomic occupancy and regulation by chromatin immunoprecipitation followed by sequencing (ChIP–seq). We tested all commercially available H2BNTac site monoclonal antibodies. Six antibodies, targeting H2BK5ac, H2BK12ac, H2BK16ac and H2BK20ac, passed quality control in ChIP–quantitative PCR (qPCR) analyses (Supplementary Table 1). Antibodies targeting the same sites are distinguished by their clone's names as H2BK5ac[EP857Y], H2BK5ac[D5H1S], H2BK20ac[EPR859] and H2BK20ac[D7O9W].

H2BNTac occupancy was analyzed in mouse embryonic stem cells (mESCs) treated with or without the CBP/p300 catalytic inhibitor A-485 (ref. [24]). A roughly similar number of peaks were identified for H3K27ac, H2BK5ac[EP857Y], H2BK16ac, H2BK20ac[EPR859] and H2BK20ac[D7O9W], but fewer peaks were identified for H2BK5ac[D5H1S] and H2BK12ac (Supplementary Fig. 1b). This difference was not due to a lower sequencing depth but due to lower H2BK5ac[D5H1S] and H2BK12ac peak intensity (Supplementary Fig. 1b). Regardless, most H2BNTac site peaks overlapped with each other, and 92–99% of H2BK5ac[D5H1S] and H2BK12ac peaks overlapped with H2BK16ac and H2BK20ac (Supplementary Fig. 2a). The height of overlapping H2BNTac peaks was significantly greater than nonoverlapping peaks. Most (71–92%) H3K27ac and H2BNTac peaks occurred in assay for transposase-accessible chromatin (ATAC) accessible regions, and ATAC-nonoverlapping peaks were low-abundant (Supplementary Fig. 2b). After completing the initial analyses, we obtained an antibody-recognizing H2BK11ac. H2BK11ac peaks also extensively overlapped (84–96%) with other H2BNTac sites (Supplementary Fig. 2c), showing that most H2BNTac sites mark the same genomic regions.

### H2BNTac partially overlaps with H3K27ac

In intergenic regions, most H3K27ac peaks were co-occupied by H2BNTac but in promoter regions, a large fraction of H3K27ac peaks lacked H2BNTac (Supplementary Fig. 3). Overall, 91% of H3K27ac peaks overlapped with H2BK5ac[EP857Y], 72% overlapped with H3K9ac and 38–62% with other H2BNTac sites (Fig. 1a). H2BK5ac[EP857Y] is an outlier among H2BNTac antibodies and the reason for this is discussed below. Compared to H3K27ac overlapping peaks, the height of nonoverlapping H2BNTac peaks was significantly lower. This shows that almost all strongly marked H2BNTac+ regions are marked with H3K27ac but many abundantly marked H3K27ac+ regions lack H2BNTac.

### H2BNTac distinctly correlates with other chromatin marks

We analyzed H2BNTac association with chromatin marks associated with promoters (H3K4me3, H3K9ac), enhancers (H3K4me1), both at promoters and enhancers (MED1, H3K27ac) or Polycomb-repressed regions (H3K27me3). H3K4me3 positively correlated with H3K9ac, MED1 and H3K27ac but negatively correlated with H3K27me3 (Fig. 1b and Supplementary Fig. 3). H2BK5ac[D5H1S], H2BK12ac, H2BK16ac and H2BK20ac strongly correlated with each other (Spearman's $\rho$ = 0.84–0.93; Pearson's $r$ = 0.85–0.94) but modestly correlated (Spearman's $\rho$ = 0.30–0.72; Pearson's $r$ = 0.24–0.73) with H3K27ac, H3K9ac and MED1, and weakly or negatively correlated (Spearman's $\rho$ = −0.55–0.20; Pearson's $r$ = −0.53–0.15) with H3K4me3 and H3K27me3. H3K27me3 was more strongly negatively correlated with H3K27ac than H2BNTac, probably because of the mutual exclusivity of H3K27ac and H3K27me3. Notably, in intergenic regions, H2BNTac positively correlated with

H3K27ac, H3K9ac and MED1; however, at active promoters, H2BNTac showed little correlation with any of these marks (Fig. 1b and Supplementary Fig. 3). Inspection of individual loci confirmed distinct occupancies of H2BNTac and H3K27ac (Fig. 1c).

The occupancy profile of H2BK5ac[EP857Y] was different from other H2BNTac antibodies; H2BK5ac[EP857Y] showed the strongest correlation with H3K27ac (Supplementary Fig. 4). Some histone modification antibodies can cross-react with unintended histone sites[25]. A high sequence similarity near H2BK5 and H3K27, and H2BK20 and H2BK120 (Supplementary Fig. 5a), prompted us to evaluate the specificity of the H2BK5ac, H2BK20ac and H3K27ac antibodies used in this or previous studies[22,23]. We observed cross-reactivity of H2BK5ac[EP857Y], H2BK5ac[polyclonal] and both H2BK120ac[polyclonal] but not of H2BK20ac antibodies (Supplementary Fig. 5b–e). H2BK5ac[D5H1S] showed no measurable cross-reactivity in quantitative image-based cytometry; however, in ChIP–seq, it showed a stronger correlation with H3K27ac than other H2BNTac site antibodies (Supplementary Fig. 4), indicating that H2BK5ac[D5H1S] may weakly recognize H3K27ac. The observed cross-reactivities can be rationalized by the sequence similarities and can explain the unexpectedly strong correlations between the ChIP–seq profiles of H2BK5ac[EP857Y] and H3K27ac and between H2BK20ac and H2BK120ac[22,23]. Because of cross-reactivity, H2BK5ac[EP857Y] was excluded from defining the H2BNTac signature.

### H2BNTac proficiently marks distal enhancers

Promoter-distal MED1 binding marks candidate enhancers[26,27]. We grouped acetylation site peaks into quartiles and determined the fraction of peaks mapping to MED1+ intergenic regions. H2BK16ac, H2BK20ac and H3K27ac marked most MED1+ intergenic regions (Fig. 2a). H2BK12ac and H2BK5ac[D5H1S] marked a smaller fraction of MED1+ regions, reflecting limited coverage of these marks in ChIP–seq (Supplementary Fig. 1b). H3K27ac and H2BNTac marked virtually all MED1+ superenhancers[27] (Fig. 2b). These results show that H3K27ac and H2BNTac are similarly sensitive in detecting MED1+ candidate enhancers.

mESC enhancers are bound by NANOG and OCT4 (ref. [28]). NANOG and OCT4 bound a larger fraction of H2BNTac+ than H3K27ac+ regions (Supplementary Fig. 6). In the top quartile, 88–97% of H2BNTac+ regions were bound by NANOG or OCT4. Notably, NANOG or OCT4 binding was similar in promoter and distal H2BNTac+ regions, whereas NANOG or OCT4 binding was much lower in H3K27ac+ and MED1+ promoters than in distal regions.

### H2BNTac poorly marks constitutively active promoters

In mESCs, most (90–98%) active promoters were marked by H3K27ac, MED1 and H3K9ac but a large portion of promoters lacked H2BNTac (Fig. 2c). In the top quartile, most H3K4me3, H3K9ac, H3K27ac and MED1 peaks occurred in promoters, whereas most H3K4me1 and H2BNTac peaks occurred in distal regions (Fig. 2d). The relative abundance of chromatin marks discriminated promoters from candidate enhancers. In H3K27ac+ regions, the H3K9ac:H3K27ac ratio was higher in promoters than in distal regions; the H3K4me3:H3K27ac ratio was even higher in active promoters than in distal regions (Fig. 2e). Notably, the H2BNTac:H3K27ac ratio showed an opposite pattern; the H2BNTac:H3K27ac ratio was much higher in distal regions than active promoters. Analysis of H3K9ac+ and H2BK20ac+ regions further confirmed these differences (Supplementary Fig. 7a,b).

Confirming distinct specificities, H3K27ac contiguously marks promoters and upstream regions in *Eif4a2*, *Actb* and *Taf1*, whereas H2BNTac only marks the upstream regions near *Eif4a2* and *Actb* (Supplementary Fig. 8a). The *Taf1* upstream region (H3K27ac+H2BNTac−) had high H3K4me3 and high nascent transcription, indicating that it is probably an unannotated alternative promoter of *Taf1*. Some DHS+H3K4me3+ regions (Supplementary Fig. 8b) were previously interpreted as candidate enhancers[27,29] but the lack of H2BNTac

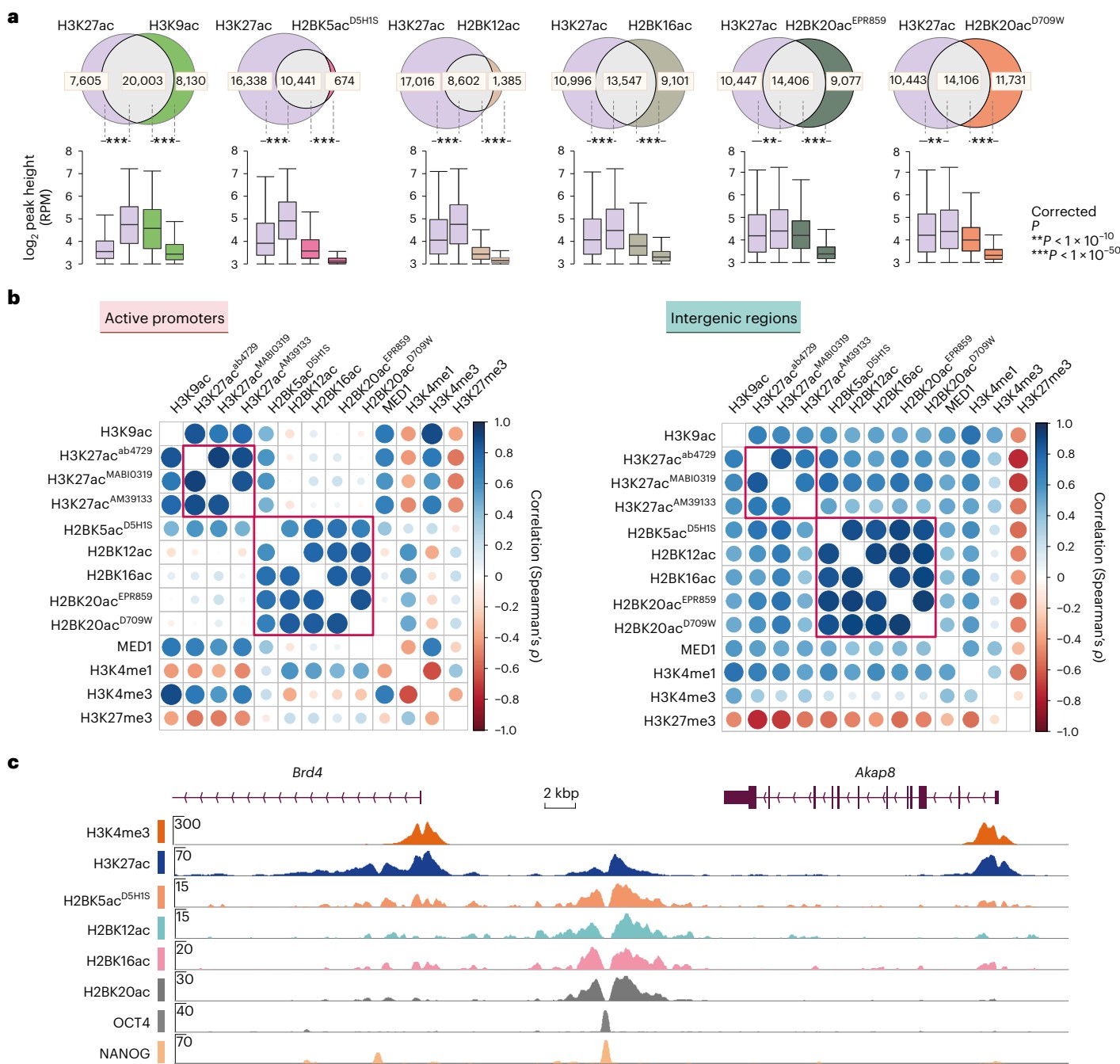

**Fig. 1 | Most H2BNTac site peaks overlap with H3K27ac. a**, Venn diagrams showing the overlap between the number of ChIP−seq peaks identified for the indicated histone acetylation marks. The box plots below each Venn diagram show the ChIP signal intensity of the overlapping and nonoverlapping peaks. One H2BNTac peak can overlap with more than one H3K27ac peak or vice versa, which leads to slight differences in the total number of H3K27ac peak counts in the different Venn diagrams. The box plots display the median, upper and lower quartiles; the whiskers show the 1.5× interquartile range (IQR). Number of ChIP−seq biological replicates: H2BK5ac^D5H1S (*n* = 4); H3K27ac (*n* = 3); H2BK12ac (*n* = 2); H2BK16ac (*n* = 2); H2BK20ac^EPR859 (*n* = 2); H2BK20ac^D709W (*n* = 1)

and H3K9ac (*n* = 1). Two-sided Mann−Whitney *U*-test, adjusted for multiple comparisons with the Benjamini−Hochberg method; **$P < 1 \times 10^{-10}$, ***$P < 1 \times 10^{-50}$. **b**, Correlation among H2BNTac sites and the other indicated chromatin marks. Pairwise correlation (Spearman's $\rho$) was determined using the normalized ChIP−seq counts, using the universe of all peaks. Left, correlation at the active promoter regions (±1 kb from the TSS). Right, correlations at intergenic regions. Color intensities and the size of the circles indicate correlation (Spearman's $\rho$). **c**, Representative genome browser tracks showing differential occupancy of H3K27ac and the indicated H2BNTac marks.

led us to realize that these are not enhancers, rather promoters of small nucleolar RNA genes.

In ChromHMM-defined states[30,31], H2BNTac was enriched comparably or higher than H3K27ac in candidate enhancers; however, in promoters, H2BNTac was enriched less prominently than H3K27ac

(Supplementary Fig. 9). ChromHMM states predicted with or without H2BNTac were largely similar (Supplementary Fig. 10). A notable difference is that, without including H2BNTac, the model predicted two states with strong H3K4me3 enrichment, one with H3K4me1 enrichment and the other without. When H2BNTac was included in

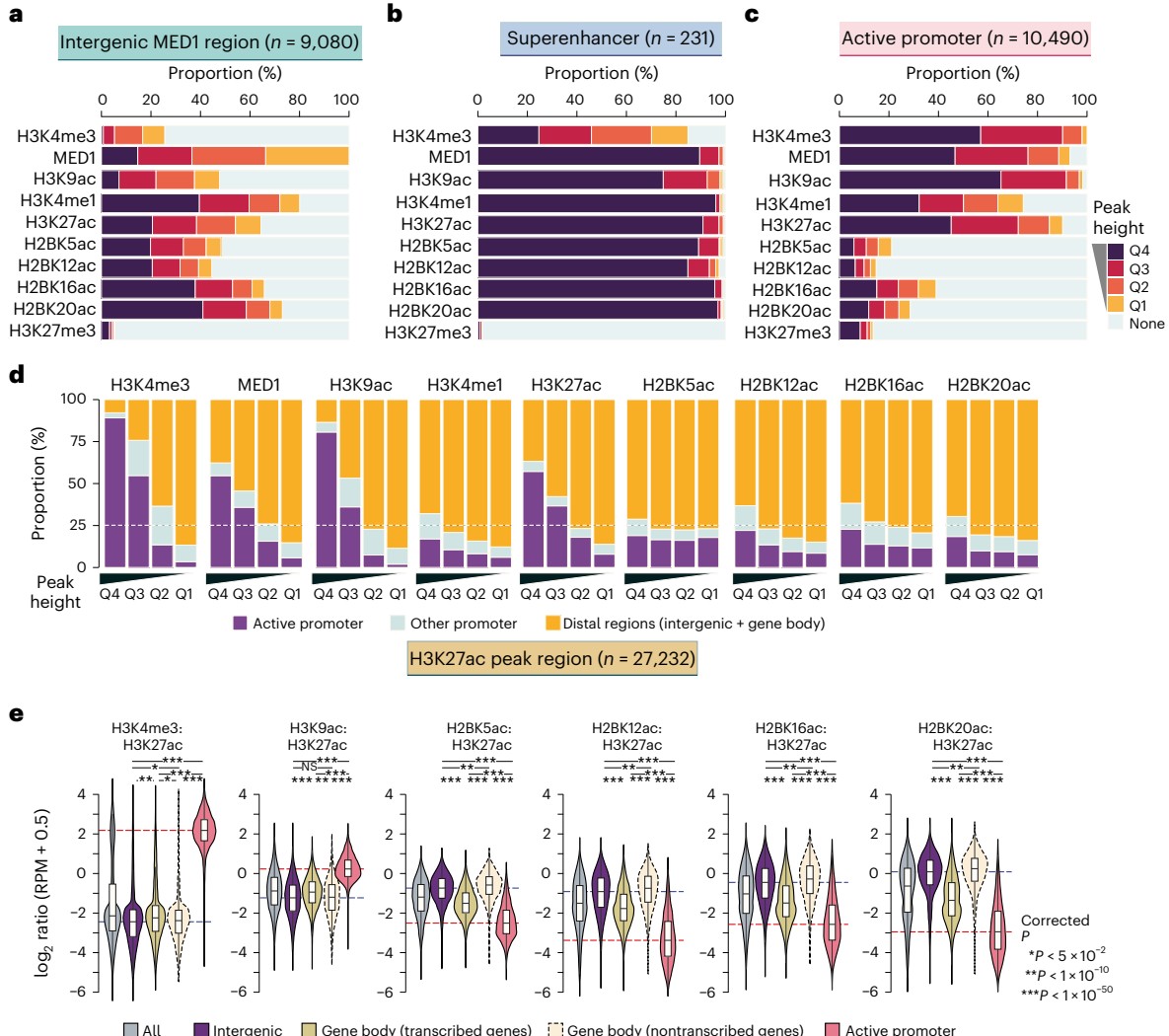

**Fig. 2 | H2BNTac prominently marks enhancers and discriminates them from active promoters. a–c**, Fraction of the indicated chromatin mark peaks mapping to MED1-occupied intergenic regions (**a**), MED1-occupied superenhancers (**b**) and active promoters (**c**). Based on peak height, the indicated chromatin marks are grouped into quartiles (Q4–Q1). Active promoters (±1 kb from the TSS) are defined as those mapping to genes expressed in mESCs (5-ethynyluridine (EU) RNA sequencing (RNA-seq) transcripts per million (TPM) ≥ 2) and marked with H3K4me3. **d**, Fraction of the indicated chromatin marks mapping to the specified genomic regions. For each chromatin mark, peaks were grouped into quartiles (Q4–Q1); in each category, the fraction of peaks mapping to the indicated genomic regions is shown. Active promoters are defined as in **a** and the remaining gene TSS are classified as other promoters. **e**, The H2BNTac signal is higher in distal candidate enhancers than in active promoters. Shown are the relative ChIP signal intensities of the H3K27ac and H2BNTac sites in H3K27ac-occupied regions. H3K27ac peak regions are grouped into the following categories: all, all peaks; intergenic, peaks occurring outside promoters and gene bodies; gene body (transcribed genes, TPM ≥ 2), peaks occurring within actively transcribed gene bodies; gene body (nontranscribed genes), peaks occurring within the nontranscribed gene body. Active promoters were defined as in **a**. The dotted lines indicate the median ratio at intergenic and active promoter regions. The box plots display the median, upper and lower quartiles; the whiskers show 1.5× IQR. Two-sided Mann–Whitney $U$-test, adjusted for multiple comparisons using the Benjamini–Hochberg method; NS, not significant, $P ≥ 0.05$, *$P < 0.05$, **$P < 1 × 10^{-10}$, ***$P < 1 × 10^{-50}$.

predicting chromatin states, the H3K4me1-enriched transcriptional start site (TSS) state was merged with the enhancer state that contained both H3K27ac and H2BNTac marks. This confirms the differential enrichment of H3K27ac and H2BNTac in regions strongly marked by H3K4me3.

### CBP/p300 and histone deacetylases 1 and 2 reversibly control H2BNTac

To investigate locus-specific regulation of H2BNTac and H3K27ac, we inhibited CBP/p300 using A-485 (ref. 24). A-485 more strongly reduced H2BNTac in promoters and actively transcribed gene body regions than in distal regions (Fig. 3a and Supplementary Fig. 11a,b). H2BNTac is increased by class I deacetylase inhibitors in vivo and histone

deacetylases (HDACs) 1 and 2 can deacetylate it in vitro[19,32]. By depleting the endogenous HDACs 1 and 2 using the degradation tag approach[33], we confirmed that H2BNTac is deacetylated by HDACs 1 and 2 (Supplementary Fig. 12a,b).

### Transcription shapes H2BNTac genomic occupancy

Interestingly, H2BNTac:H3K27ac and H2BNTac:H3K9ac ratios were lower in the actively transcribed gene body regions than in intergenic regions (Fig. 2e). The H2BNTac:H3K27ac ratio was inversely associated with gene transcription; the ratio was lowest in genes with the highest expression (Fig. 3b). To investigate the link between H2BNTac and active transcription, transcription was inhibited using the CDK9 inhibitor NVP-2 (2 h, 100 nM)[34]. NVP-2 increased

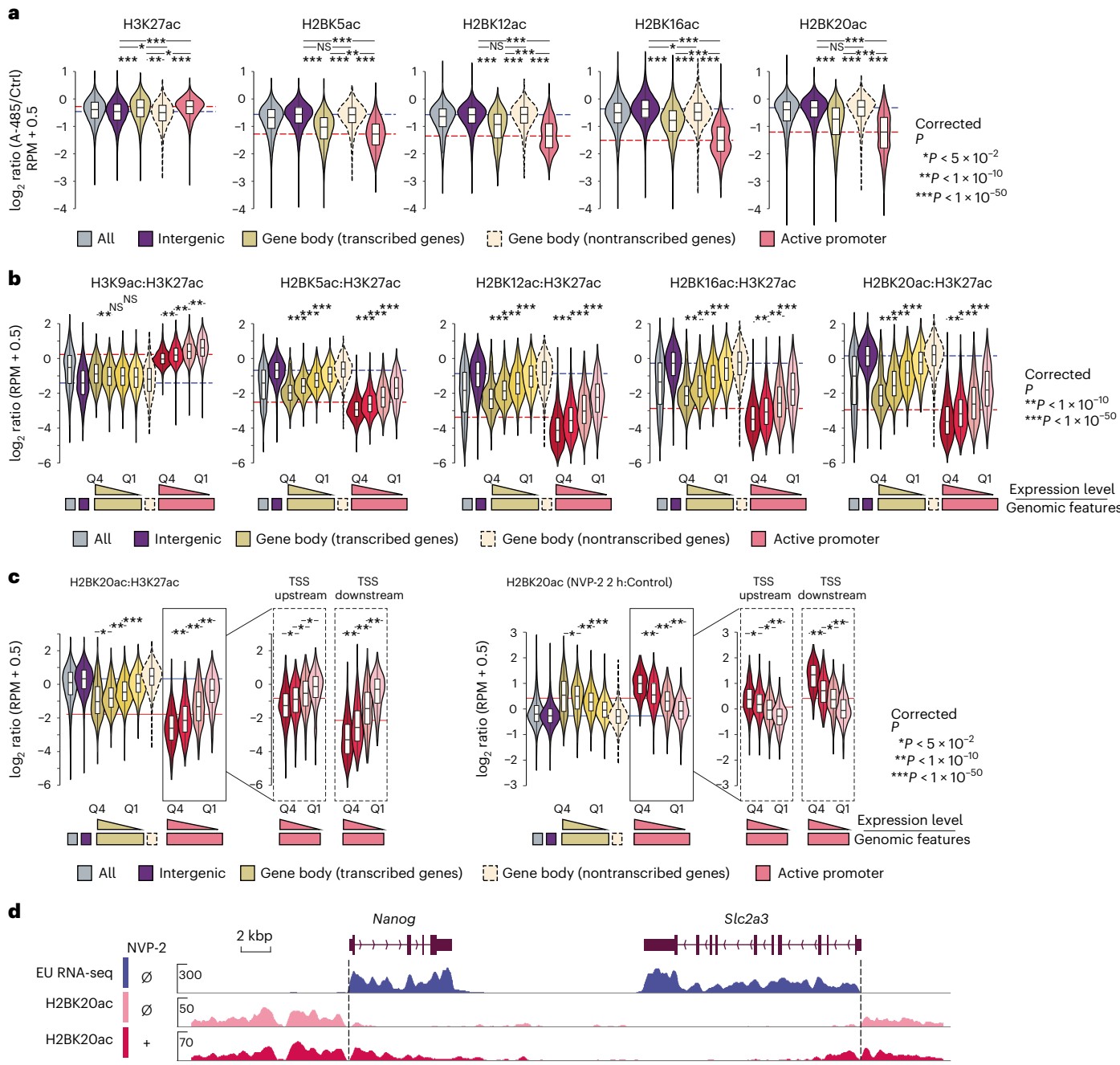

**Fig. 3 | CBP/p300 regulates global H2BNTac and the H2BNTac signal is lower in actively transcribed regions. a**, Relative ratio of H3K27ac and H2BNTac site ChIP signal intensity in untreated and A-485-treated (15 min) cells. Different genomic regions were classified as defined in Fig. 2e. The dotted lines indicate the median ratio at intergenic and active promoter regions. **b**, Ratio of the indicated histone marks in H3K27ac-marked regions. Based on the nascent transcription levels, the corresponding genes, gene body and active promoter regions were further subdivided into quartiles. **c**, Shown are the relative ratio of H2BK20ac:H3K27ac in untreated conditions (left) and the relative change in H2BK20ac in NVP-2-treated and untreated cells (right). H2BK20ac was analyzed in untreated and NVP-2-treated (2 h) cells using ChIP–seq (*n* = 1). Different

genomic regions were classified as defined in Fig. 2e. Based on the nascent transcription levels, the corresponding genes, gene body and active promoter regions were further subdivided into quartiles. Moreover, in active promoters, TSS upstream (0 ± 2 kb from the TSS) and TSS downstream (0 ± 2 kb from TSS) regions were analyzed separately. **d**, Representative genome browser tracks showing the differential increase in H2BK20ac in intergenic and transcribed gene body regions. The box plots display the median, upper and lower quartiles; the whiskers show the 1.5× IQR. Two-sided Mann–Whitney *U*-test, adjusted for multiple comparisons using the Benjamini–Hochberg method; *$P < 0.05$, **$P < 1 \times 10^{-10}$, ***$P < 1 \times 10^{-50}$.

the H2BK20ac:H3K27ac ratio more strongly in actively transcribed than nontranscribed regions (Fig. 3c,d and Supplementary Fig. 12c). The H2BK20ac increase directly corresponded to transcription; H2BK20ac was increased more strongly in highly transcribed regions.

The H2BK20ac:H3K27ac ratio was different in the TSS upstream and downstream regions and NVP-2 treatment increased H2BK20ac more strongly in the TSS downstream than in upstream regions (Fig. 3c). This shows that ongoing transcription causes loss of H2BNTac. In retrospect,

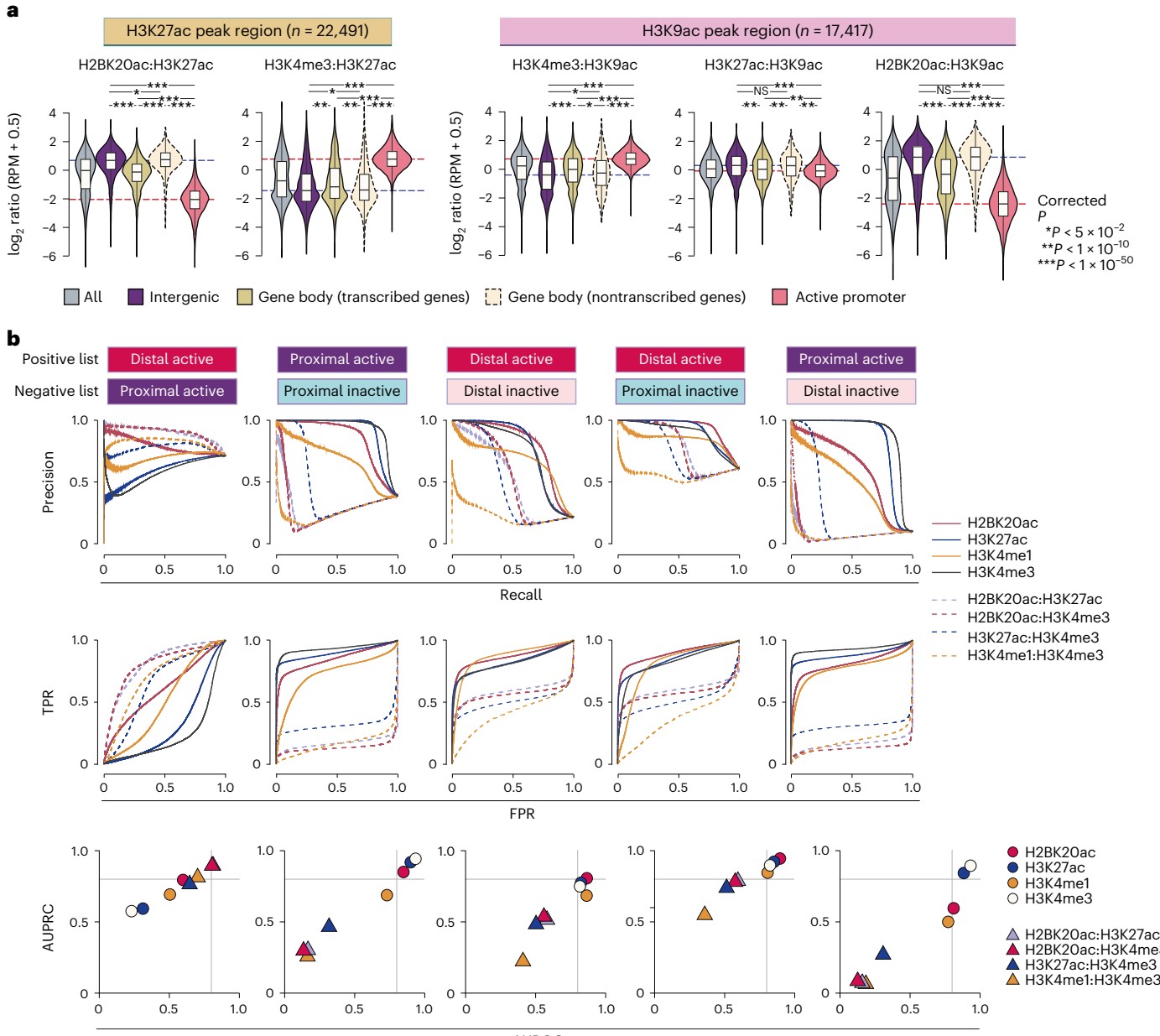

**Fig. 4 | H2BNTac segregates enhancers and active gene promoters in human cells. a**, Relative ChIP–seq signal intensities of the indicated histone marks at the H3K27ac⁺ (left) or H3K9ac⁺ (right) peak regions. The H3K27ac and H3K9ac peak regions were further grouped into the indicated categories, as specified in Fig. 2e. Active promoters are defined as TSS-proximal regions (±1 kb from the TSS) of actively transcribed (RNA-seq, TPM ≥ 2) genes in K562 cells and whose promoters are marked with H3K4me3. Number of ChIP–seq biological replicates: H3K4me3 ($n = 2$); H2BK20ac ($n = 2$); H3K9ac ($n = 2$); H3K27ac ($n = 1$). The box plots display the median, upper and lower quartiles; the whiskers show the 1.5× IQR. Two-sided Mann–Whitney $U$-test, adjusted for multiple comparisons using the Benjamini–Hochberg method; *$P < 0.05$, **$P < 1 \times 10^{-10}$, ***$P < 1 \times 10^{-50}$.
**b**, Combination of H2BNTac and other histone marks discriminate transcriptionally active and inactive PINTS regions. Transcriptionally active and inactive regions in K562 cells were defined based on RNA expression evidence in K562 and non-K562 cell lines[14]. PINTS regions were further classified as proximal or distal as defined by Yao et al.[14] (Methods). The discriminatory potential of different histone marks was analyzed by comparing ChIP–seq signal enrichment within ±500 bp of PINTS region centers. The true positive rate (TPR) and false positive rate (FPR), and AUPRC and AUROC are shown. The curves were generated under the different thresholds of peak enrichment or peak enrichment ratios.

## Enhancer specificity of H2BNTac in human cells

To confirm H2BNTac specificity in human cells, we used H2BK20ac as a representative H2BNTac mark and performed ChIP–seq in K562 cells. Like mESCs, in the H3K27ac⁺ regions, the H2BK20ac:H3K27ac ratio was much higher in distal regions than in active promoters (Fig. 4a). The H2BK20ac:H3K27ac ratio discriminated distal candidate enhancers from active promoters much more clearly than the H3K27ac:H3K9ac ratio. The H2BK20ac:H3K27ac ratio was lower in actively transcribed gene body regions than in distal regions. The H2BK20ac:H3K27ac ratio was the inverse of the H3K4me3:H3K27ac ratio in distal enhancers and active promoters. Similarly, in H3K9ac⁺ regions, the H3K27ac:H3K9ac ratio was only modestly higher at distal candidate enhancers than promoters, whereas the H2BK20ac:H3K9ac ratio was much higher in enhancers (Fig. 4a). Differential H3K27ac and H2BNTac occupancy was also confirmed in several human loci (Supplementary Fig. 13a).

## H2BNTac aids in discriminating different CREs

Next, we investigated the usefulness of different chromatin features in discriminating between active and inactive CREs. As a reference, we used transcriptionally active and inactive regions defined by peak identifier for nascent transcript starts (PINTS)[14]. Proximal and distal PINTS regions correspond to the TSS of annotated genes and candidate enhancers, respectively. PINTS regions were classified into four categories: (1) proximal active (active PINTS regions within 500 bp of the TSS); (2) proximal inactive (inactive PINTS regions within 500 bp of the TSS); (3) distal active (active PINTS regions >500 bp away from the TSS); and (4) distal inactive (distal PINTS regions >5 kb away from TSS and >2 kb away from any distal active regions in K562) (Methods). Among individual marks, H3K4me3 best separated proximal active from inactive regions (area under the precision–recall curve (AUPRC) = 0.94; area under the receiver operating characteristics (AUROC) = 0.93), while H2BK20ac best separated distal active from inactive regions (AUPRC = 0.87; AUROC = 0.81) (Fig. 4b). Notably, high H2BK20ac:H3K4me3 and H2BK20ac:H3K27ac ratios outperformed the H3K4me1:H3K4me3 ratio in separating distal active from proximal active regions. For example, in the *Ahnak* upstream region, H3K4me3 and H3K27ac strongly marked active promoters, whereas H2BNTac prominently marked *Ahnak* proximal candidate enhancers (Supplementary Fig. 13b). The corresponding region in K562 also displayed differential occupancy of these marks (Supplementary Fig. 13b). These results demonstrate that the genomic occupancy profiles of H2BNTac, H3K27ac, H3K4me3 and H3K4me1 are nonidentical. While none of the marks displayed absolute specificity, H2BNTac complemented other marks and afforded valuable information in confidently discerning candidate active enhancers.

## Validation of H2BNTac⁺ candidate enhancers by MPRA

To validate the enhancer activity of H2BNTac⁺ regions, we used MPRA-defined candidate enhancers[37]. From 25,609 MPRA⁺ peaks, only natively accessible (ATAC⁺) peaks (*n* = 10,497) were considered probable endogenously active enhancers. The number of ATAC⁺MPRA⁺ peaks was smaller than the ATAC⁻MPRA⁺ peaks, yet a greater fraction of ATAC⁺MPRA⁺ peaks overlapped with H2BNTac⁺ regions (Fig. 5a and Supplementary Fig. 14a). Overall, 32% of H3K27ac⁺ and 32–54% of H2BNTac⁺ regions overlapped with MPRA peaks. Within H3K27ac peaks, MPRA peaks showed greater overlap with the H3K27ac⁺H2BNTac⁺ than H3K27ac⁺H2BNTac⁻ peaks (Supplementary Fig. 14b,c). Of note, the number of ATAC⁺MPRA⁺ peaks was smaller than the number of H2BNTac⁺ regions; hence, ATAC⁺MPRA⁺ was not expected to validate all H2BNTac⁺ regions. The MPRA library covered approximately 83% of the genome (with an approximate 9× coverage)[37]. Assuming that the input library was not biased, the expected maximum validation rate in any quartile would be approximately 83%. Remarkably, of the top 1,000 H2BNTac⁺ regions, 80–90% overlapped with ATAC⁺MPRA⁺ regions (Fig. 5a), indicating that the validation rate in the top-ranked peaks reached near maximum. This validation rate of H2BNTac⁺ candidate enhancers compares favorably with the validation rate (29%) of eRNA-defined enhancers[14].

## H2BNTac⁺ candidate enhancer validation by eRNA expression

eRNA transcription is a surrogate of endogenously active enhancers[13,14]. A recent study cataloged hundreds of thousands of endogenously active TSS of annotated genes and candidate enhancers[14]. We combined PINTS-identified TSS from more than 110 cell types and used this as a reference to globally validate the enhancer activity of H2BNTac⁺ regions. Of the reference PINTS regions, 50,885 were active in K562 and the remaining 154,093 were inactive in K562 but active in other cell types or tissues (Fig. 5b). Based on their expression in K562, PINTS regions were classified as active or inactive. Following the original classification[14], PINTS regions were classified as proximal (TSS of annotated genes) or distal (TSS of candidate enhancers).

Overall, approximately 79–85% of H2BK20ac⁺ and H3K27ac⁺ regions overlapped with active PINTS regions (Fig. 5b). Among the top 25% of H3K27ac⁺ and H2BK20ac⁺ regions, virtually all overlapped with active PINTS regions, showing that almost all abundantly marked H2BNTac⁺ or H3K27ac⁺ regions are actively transcribed. Inspection of genome browser tracks confirmed excellent concordance between H2BNTac with active PINTS regions (Supplementary Fig. 15). These analyses showed that the validation rate of H2BNTac⁺ candidate enhancers is comparable or higher than enhancers defined by other features[13,14,38–41].

## H3K27ac⁻ and H2BNTac⁻ noncanonical enhancers are rare

Next, we aimed to get an estimate of the active H3K27ac⁻ noncanonical enhancers[7–10,42]. As a reference enhancer set, we used MPRA-defined mESC enhancers[37]. We assumed that (1) enhancer scoring in MPRA is not biased to specific chromatin modifications, (2) ATAC⁺MPRA⁺ regions represent a set of candidate enhancers that are probably active in vivo and (3) ATAC⁺MPRA⁺H3K27ac⁻ or ATAC⁺MPRA⁺H2BNTac⁻ regions represent endogenously active noncanonical enhancers. Strikingly, 88% (9,216 out of 10,497) of all distal ATAC⁺MPRA⁺ peaks overlapped with H2BK20ac or H3K27ac (Fig. 6a). In the top quartile of highly active ATAC⁺MPRA⁺ enhancers, more than 94% overlapped with H2BK20ac⁺ or H3K27ac⁺ regions. The ATAC⁺MPRA⁺ peaks that did not overlap with H3K27ac or H2BNTac had low chromatin accessibility (Fig. 6b). This indicates that most endogenously active enhancers are marked with H3K27ac or H2BNTac.

To further confirm this, we used PINTS-defined candidate active enhancers as a reference. The H2BK20ac:H3K27ac ratio was significantly higher at distal PINTS regions than at proximal PINTS ones (Fig. 6c). We hypothesized that H3K27ac⁺ or H2BK20ac⁺ distal active PINTS regions represent canonical enhancers and H3K27ac⁻ or H2BK20ac⁻ distal active PINTS regions represent noncanonical active enhancers. Seventy-five per cent of all distal active PINTS regions were marked by H3K27ac or H2BK20ac, whereas just 4% of distal inactive PINTS regions were marked by H3K27ac or H2BK20ac (Fig. 6d), even though the number of inactive PINTS regions is approximately three times larger than active PINTS regions (Fig. 5b). Nonoverlapping active PINTS regions had low accessibility (Fig. 6e). These analyses suggest that H3K27ac or H2BNTac mark most of the endogenously active enhancers. Noncanonical active enhancers are either rare or occur in regions that have very low DNA accessibility.

## H2BNTac correlates with CBP/p300-dependent gene regulation

Gene expression correlates with the abundance of active chromatin marks at promoters[43]. H3K27ac, H3K9ac and MED1 abundance positively correlated with gene expression, but interestingly, H2BNTac showed no clear relationship (Supplementary Fig. 16a). Unlike H3K27ac, H3K9ac and MED1, promoter H2BNTac abundance was associated with cell type specificity; genes with a higher H2BNTac signal were expressed in fewer cell types (Supplementary Fig. 16b). A-485-induced gene downregulation correlated with the decrease in promoter H3K27ac but H2BNTac showed poor correlation (Supplementary Fig. 17a), probably because H2BNTac is decreased globally but only a subset of genes is downregulated.

It would be highly useful if chromatin mark abundance could predict a gene's dependence on CBP/p300 activity. We noted that the H2BNTac:H3K27ac ratio was much higher in promoters of A-485-downregulated genes than in unregulated genes (Fig. 7a). To test the association between H2BNTac and CBP/p300-dependent gene activation, genes were ranked based on MED1, H3K27ac, H3K9ac and H2BNTac ChIP signal in promoters; within each rank category, the fraction of A-485-downregulated genes was determined. Globally, approximately 13–14% of genes were downregulated (twofold or greater) after 1 h of CBP/p300 inhibition[21] (Fig. 7b). MED1 and H3K9ac abundance showed no relationship with A-485-induced gene regulation. H3K27ac abundance was modestly associated with CBP/p300-dependent gene

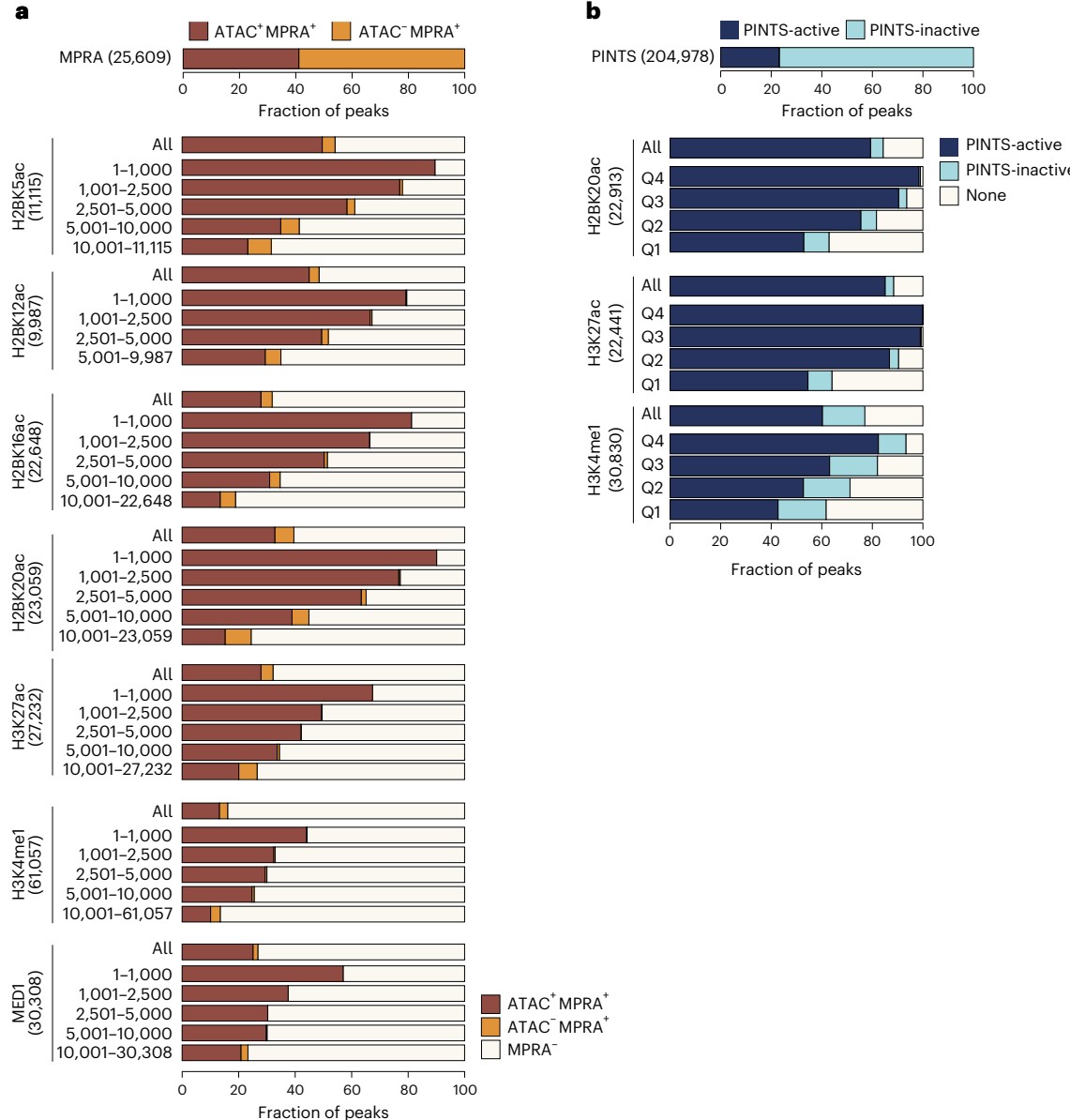

**Fig. 5 | H2BNTac⁺ regions show a high validation rate in orthogonal assays. a**, Validation of enhancer activity of H2BNTac⁺ regions by MPRA. Shown are the fraction of H2BNTac, H3K27ac, H3K4me1 and MED1 regions overlapping with ATAC⁺MPRA⁺ or ATAC⁻MPRA⁺ regions in mESCs. The top bar chart shows the fraction of MPRA peaks with or without the ATAC signal. The lower bar charts show the fraction of the indicated chromatin mark ChIP–seq peaks that overlap with the ATAC⁺MPRA⁺ and ATAC⁻MPRA⁺ regions. The indicated chromatin marks were ranked based on peak height and grouped into the indicated rank categories; the overlap with MPRA regions is shown within each rank category and for all peaks (All). **b**, Most H2BNTac⁺ regions are actively transcribed in vivo. PINTS-active regions refer to regions that are actively transcribed in K562 cells; PINTS-inactive regions refer to regions that are not transcribed in K562 cells but transcribed in other human cells and tissues[14] (Methods). The top bar chart shows the fraction of PINTS regions that are active or inactive in K562 cells. The lower bar charts show the fraction of the indicated chromatin mark ChIP–seq peaks that overlap with PINTS-active and PINTS-inactive regions. None, no PINTS classification was available for these regions.

regulation. Among the top 100 H3K27ac⁺ genes, 35% were downregulated; however, among the top 1,000, only 20% were downregulated, which is only marginally higher than the average (13%). H2BNTac showed a stronger association with A-485-induced gene downregulation. In the top 100 highest H2BNTac⁺ promoters, 78–94% were downregulated; in the top 500, 67–78% were downregulated; and in the top 1,000, 56–70% were downregulated (Fig. 7b). Indeed, the overall distribution of H2BNTac⁺ genes was shifted toward downregulation. A-485-induced gene downregulation correlated more strongly with H2BNTac than H3K27ac promoter intensity (Spearman's $\rho = -0.52$ to $-0.65$ versus $-0.10$) (Fig. 7c). p300 enrichment in A-485-downregulated

and not changed promoters was not appreciably different, except that p300 binding was somewhat elevated in the regions flanking A-485-downregulated TSS (Supplementary Fig. 17b). Also, the H2BN-Tac signal appeared to extend beyond the nucleosomes flanking the A-485-downregulated TSS (Fig. 7d).

### Promoter H2BNTac signal predicts CBP/p300 target genes

By integrating H3K27ac, DHS and Hi-C data, the activity-by-contact (ABC) model can predict enhancer targets[44]. We asked if the model can be adapted to predict CBP/p300 target genes. The ABC score was calculated pairwise for each enhancer and target gene. A-485 treatment

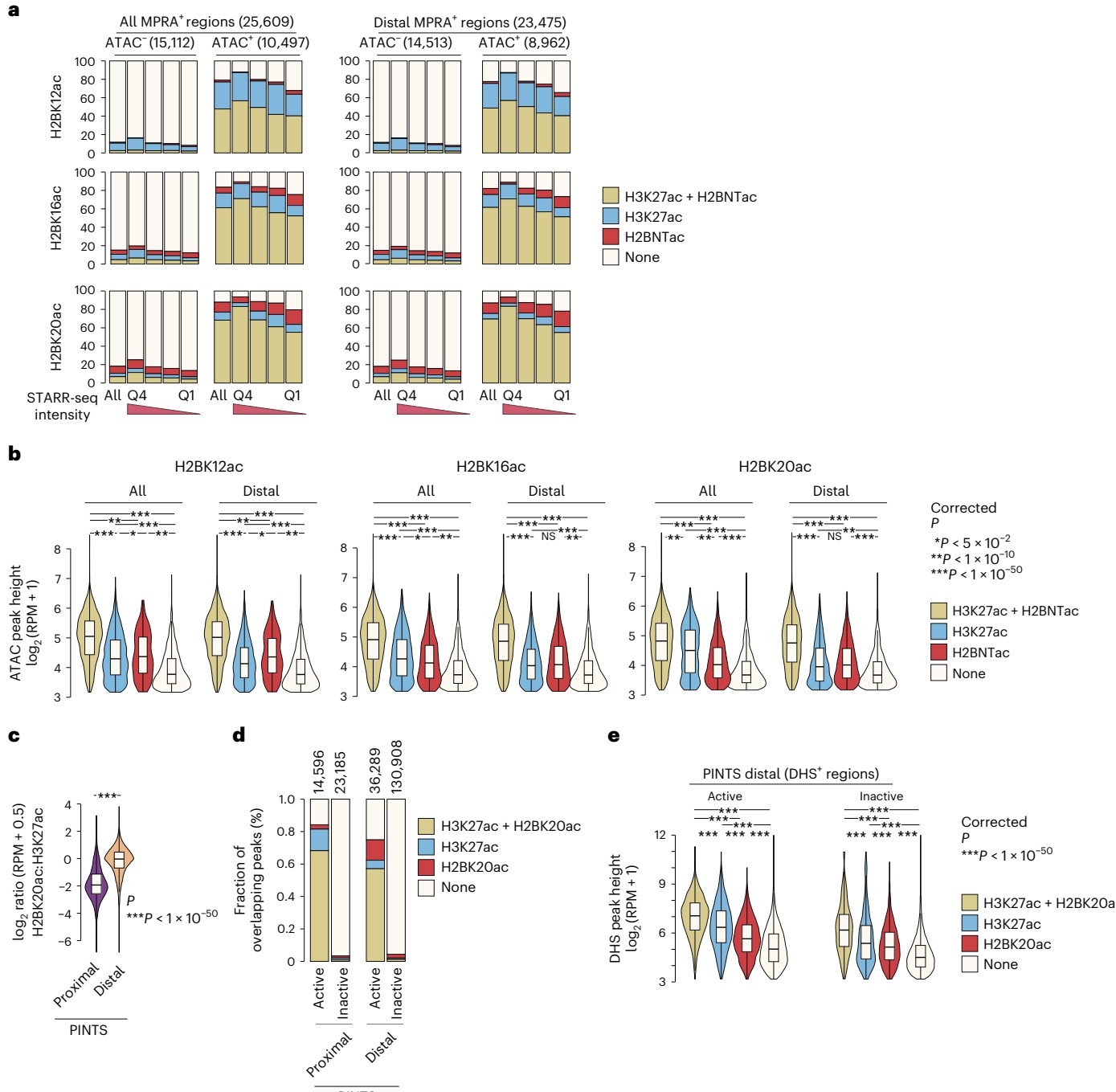

**Fig. 6 | The most active enhancer regions are marked by H2BNTac and H3K27ac. a**, Most ATAC⁺MPRA⁺ regions are marked by H2BNTac or H3K27ac. MPRA regions were grouped into ATAC⁺ and ATAC⁻. Within each group, MPRA⁺ regions were subgrouped into quartiles based on the self-transcribing active regulatory region sequencing (STARR-seq) signal. Shown is the overlap of H2BNTac or H3K27ac in the indicated groups of MPRA regions. All, all MPRA⁺ regions; distal MPRA⁺ regions, MPRA⁺ regions mapping more than ±1 kb away from annotated TSS regions. **b**, H2BNTac or H3K27ac nonoverlapping MPRA⁺ regions have low DNA accessibility. MPRA⁺ATAC⁺ regions were grouped based on the overlap of the indicated histone acetylation marks; within each group, ATAC peak height was analyzed. MPRA⁺ATAC⁺ regions overlapping with the histone marks have much higher accessibility than regions lacking these marks. **c**, The

relative H2BK20ac abundance of H2BK20ac is higher in distal PINTS regions. Shown is the ratio of H2BK20ac:H3K27ac ChIP–seq signal in proximal and distal PINTS regions in K562 cells. **d**, PINTS regions were grouped into proximal and distal, and further subgrouped into active and inactive (Methods). Shown is the fraction of PINTS regions overlapping with H2BK20ac or H3K27ac in K562 cells. **e**, Distal PINTS-active regions were grouped based on the occurrence of the indicated histone marks. Within each category, DHS peak height was determined. Distal PINTS-active regions marked with H2BK20ac or H3K27ac have much higher accessibility than regions lacking these marks. The box plots display the median, upper and lower quartiles; the whiskers show 1.5× IQR. Two-sided Mann–Whitney $U$-test, adjusted for multiple comparisons using the Benjamini–Hochberg method; *$P < 0.05$, **$P < 1 \times 10^{-10}$, ***$P < 1 \times 10^{-50}$.

did not impact the activity of individual enhancers but inhibited CBP/p300 function globally. Therefore, to predict CBP/p300 target genes, we used summed nominal ABC scores of all enhancers connected to

a gene. ABC scores were calculated using ATAC followed by sequencing (ATAC–seq), Hi-C contact frequency and H3K27ac or H2BNTac ChIP signal. Our positive set included genes that were downregulated

(greater than twofold) after CBP/p300 inhibition; the negative set included genes that remained unaffected (<1.2-fold regulation) by CBP/p300 inhibition[21].

The H3K27ac-based ABC predicted A-485-downregulated genes with a good accuracy (AUPRC = 0.58, AUROC = 0.76) (Fig. 8a,b). Substituting H3K27ac with H2BNTac slightly increased the prediction performance (AUPRC = 0.62–0.64, AUROC = 0.78–0.79), showing that the ABC model can be adapted to predict CBP/p300 targets with an accuracy that is similar to predicting enhancer–gene pairs from CRISPR interference (CRISPRi) data[44]. Micro-C provides better resolution than Hi-C[45,46]. However, substituting Hi-C with Micro-C only marginally improved the model's performance (Fig. 8a,b).

To further compare the usefulness of H3K27ac and H2BNTac in the context of the ABC model, we used two differently sized CRISPRi-mapped enhancer–gene pair datasets[44,47] and calculated ABC scores using H3K27ac or H2BK20ac (while keeping the other parameters unchanged). On the smaller dataset with a limited number of enhancer–gene pairs, H3K27ac-based ABC scores performed well (AUPRC = 0.65); as reported originally[44], substituting H3K27ac with H2BK20ac slightly improved the performance (AUPRC = 0.69) (Supplementary Fig. 17c). H3K27ac only partly contributed to the ABC scores; promoter proximal regions, where H2BK20ac and H3K27ac abundance is most different, were excluded from predicting enhancer-dependent gene regulation. This may explain why substituting H3K27ac with H2BNTac has a minor impact on the ABC model's performance. Surprisingly, on the larger dataset of enhancer–gene pairs, the model performed poorly (AUPRC = 0.25–0.27), regardless of whether we used H3K27ac or H2BK20ac. This indicates that the ABC model's performance was more heavily influenced by the size or type of CRISPRi dataset used, than the use of H3K27ac or H2BK20ac.

Because H2BNTac promoter intensity correlated with A-485-dependent gene regulation (Fig. 7), we asked if ChIP–seq signals in promoters could predict CBP/p300 target genes. We determined the enrichment of H3K4me3, H3K4me1, H3K27ac and H2BNTac in promoters (±1 kb from the TSS). The promoter ChIP signal predicted A-485-downregulated genes in the following order: H3K4me3 < H3K27ac < H3K4me1 < H2BNTac. Strikingly, in this comparison, H2BNTac promoter intensity predicted CBP/p300 regulated genes better than the ABC model. H2BNTac promoter intensity alone, without involving DHS or three-dimensional (3D) genome contact data, provided the highest prediction accuracy (AUPRC = 0.83–0.84, AUROC = 0.92) (Fig. 8a,b). These results show that CBP/p300 contribution to gene activation is better reflected in the promoter intensity of H2BNTac than H3K27ac. We propose that the promoter H2BNTac intensity can be used to estimate a CBP/p300 activity score ('A score'). The A score offers a simple and accurate measure for predicting strongly regulated CBP/p300 targets and their quantitative dependencies.

## Discussion

The number of H3K27ac genome-wide mapping studies has increased linearly since H3K27ac was linked to enhancers[11,26], whereas H2BNTac

has remained largely unstudied (Supplementary Fig. 18). Through rational follow-up of the CBP/p300-regulated acetylome[18], we rigorously established H2BNTac as a distinctive and valuable signature of enhancers (Fig. 8c).

Systematic comparisons allowed us to present a coherent, genome-scale map of H2BNTac site occupancy and locus-specific regulation. We cannot entirely rule out possible cross-reactivity of H2BNTac antibodies, but the specificity of the H2BNTac signature is supported by multiple lines of evidence (Supplementary Note 2). We confirmed the enhancer specificity of H2BK20ac but found no concrete evidence of a distinct class of strong H2BK20ac⁺ enhancers that lack H3K27ac[9]. Our analyses conclusively showed that H2BK20ac is not an outlier among H2BNTac sites and all analyzed H2BNTac sites indistinguishably marked the same genomic regions. Our results identified H2BNTac as the most distinctive marker of CBP/p300 activity, offered a mechanistic explanation for the differences in H2BNTac and H3K27ac genomic occupancy and revealed a yet underappreciated role of transcription-coupled nucleosome exchange in shaping H2BNTac occupancy[35,36].

The identification of a distinctive H2BNTac signature allowed us to address two fundamental questions that are frequently discussed in contemporary reviews[12,17,42,48]: (1) How reliable are histone acetylation marks for predicting candidate enhancers?; and (2) What fraction of endogenously active enhancers remain undetected by histone acetylation marks?

Using MPRA- and eRNA-defined candidate enhancers, we demonstrated a high validation rate of H2BNTac⁺ enhancers. We found that virtually all highly acetylated regions are actively transcribed. The differences in validation rate are caused by the abundance of histone marks and the depth of analyses. Highly acetylated regions have a higher validation rate than weakly acetylated regions (Fig. 5). If the depth of ChIP–seq analyses is limited, only highly active enhancers get detected and the validation rate is higher. For example, because fewer H2BK12ac peaks were detected, the validation rate of H2BK12ac⁺ regions was higher than that of H2BK20ac⁺ regions. Similarly, with increasing depth of eRNA detection, the validation rate of eRNA-defined enhancers decreases from approximately 70% to approximately 30%[13,14], which is no higher than the validation rate of candidate enhancers defined by histone marks. The differences in the depth of analyses can, at least partially, rationalize differences in validation rates in previous studies[13,14,38–41]. Overall, we conclude that histone acetylation marks are reasonably sensitive and accurate in sampling most of the enhancers. An advantage of MPRA- and eRNA-centric approaches is that they offer higher resolution and can map enhancer positions more precisely than histone marks. Histone modifications have the practical advantage that DNA is more stable than RNA, and ChIP–seq is relatively cheaper and less laborious.

It has been suggested that there are different classes of distal enhancers[9,10,22,23,42]. We used thousands of ATAC⁺MPRA⁺ and eRNA-defined enhancers to get an approximation of endogenously active noncanonical enhancers. If active noncanonical enhancers were

---

**Fig. 7 | H2BNTac marks cell type-specific promoters and predicts CBP/p300 target genes in mESCs. a**, Ratio of H2BNTac sites and H3K27ac in the promoters of the indicated CBP/p300-regulated gene categories. mESC genes were grouped into A-485-downregulated (Down), slightly downregulated or not changed (NC) categories, as defined by Narita et al.[21] (EU RNA-seq: *n* = 5 biological replicates). The ratio of the indicated histone marks was determined by normalized ChIP–seq counts in promoters (within ±1 kb of the TSS) of the respective gene categories. The box plots display the median, upper and lower quartiles; the whiskers show the 1.5× IQR. Two-sided Mann–Whitney *U*-test, adjusted for multiple comparisons using the Benjamini–Hochberg method; *$P < 0.05$, **$P < 1 \times 10^{-10}$, ***$P < 1 \times 10^{-50}$. **b**, Genes expressed in mESCs were ranked based on the ChIP–seq signal of the indicated marks in promoter regions (within ±1 kb of the TSS). Shown are the composite nascent RNA transcription profiles for all genes, as well

as the top 100, 500 and 1,000 genes with the highest abundance of the indicated mark. Within each group, the fraction of A-485-downregulated genes is indicated. Change in nascent transcription was determined by EU RNA-seq in mESCs treated without or with A-485 (1 h)[21]. The dotted lines indicate a twofold downregulation of nascent transcription after A-485 treatment. **c**, Correlation between the ChIP–seq signal of the indicated chromatin marks in promoter regions and A-485-induced nascent transcription changes in mESCs (Spearman's *ρ*). **d**, Aggregate plots showing the average ChIP signal of the indicated marks in the specified classes of A-485-regulated gene promoters in mESCs. A-485-regulated genes[21] are classified as follows: NC, transcription decreased by less than 1.2-fold after A-485 treatment; slightly downregulated (transcription decreased by ≥1.5-fold after A-485 treatment); downregulated (transcription decreased by twofold or greater after A-485 treatment).

widespread, we would expect to find many ATAC⁺MPRA⁺ peaks that are H3K27ac⁻ or H3K27ac⁻ regions that express eRNA. Eighty-eight per cent of distal ATAC⁺MPRA⁺ peaks and 75% of distal active PINTS⁺ regions, overlapped with H3K27ac⁺ or H2BK20ac⁺ regions (Fig. 6). The nonoverlapping ATAC⁺MPRA⁺ and active PINTS⁺ regions have no or low chromatin accessibility and are unlikely to represent endogenously active enhancers. We do not rule out their existence, but we did not find strong evidence for a large number of H3K27ac⁻

or H2BNTac⁻ endogenously active noncanonical enhancers. This realization is important for estimating the extent to which canonical and noncanonical enhancers contribute to global gene regulation.

One of the salient findings of our work is the discovery that H2BNTac is a useful marker for predicting CBP/p300 target genes (Fig. 8). H2BNTac is also a good predictor of enhancer strength[49], showing that H2BNTac is not just 'yet another enhancer marker' but offers genuinely complementary advantages to the existing marks.

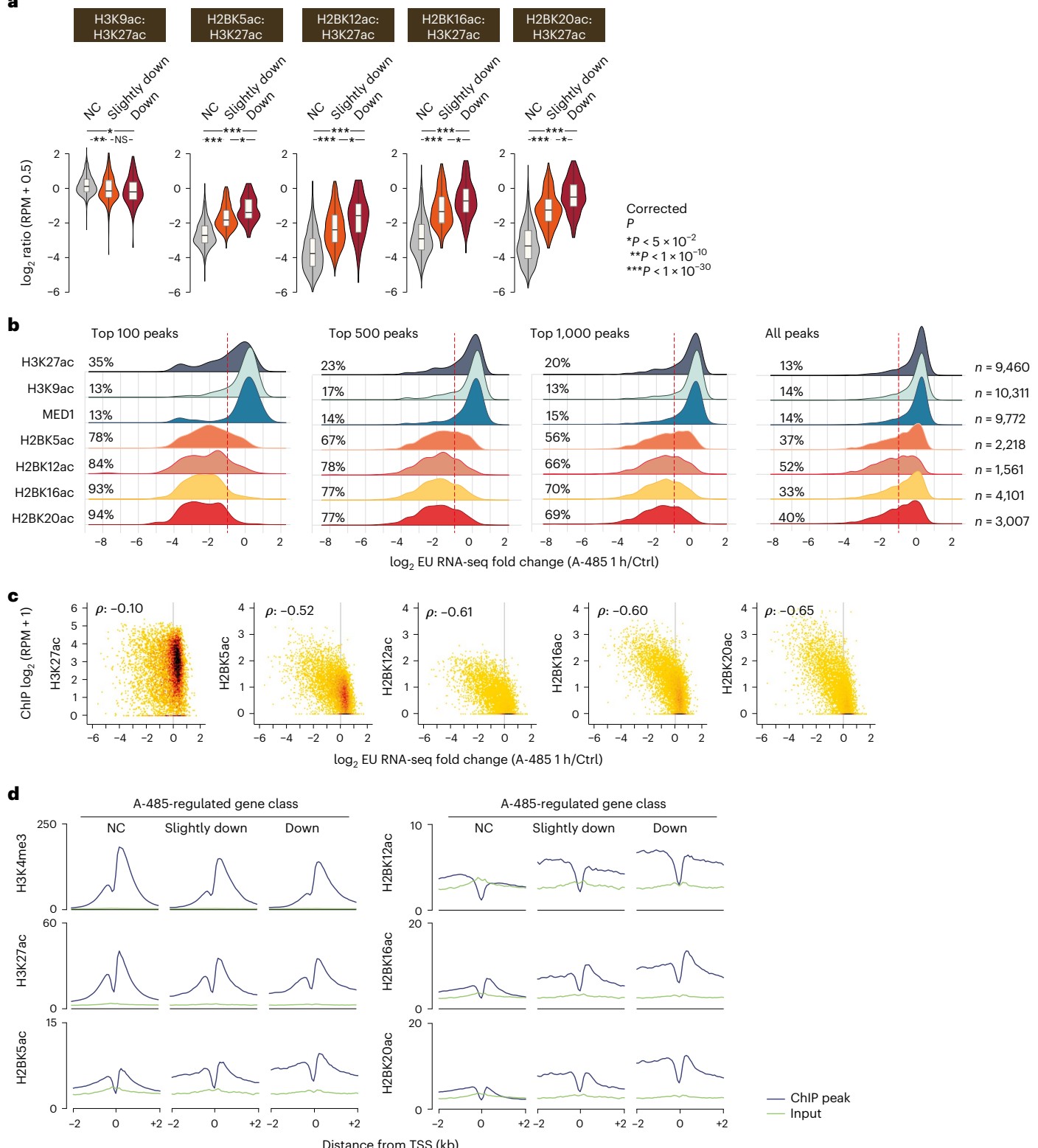

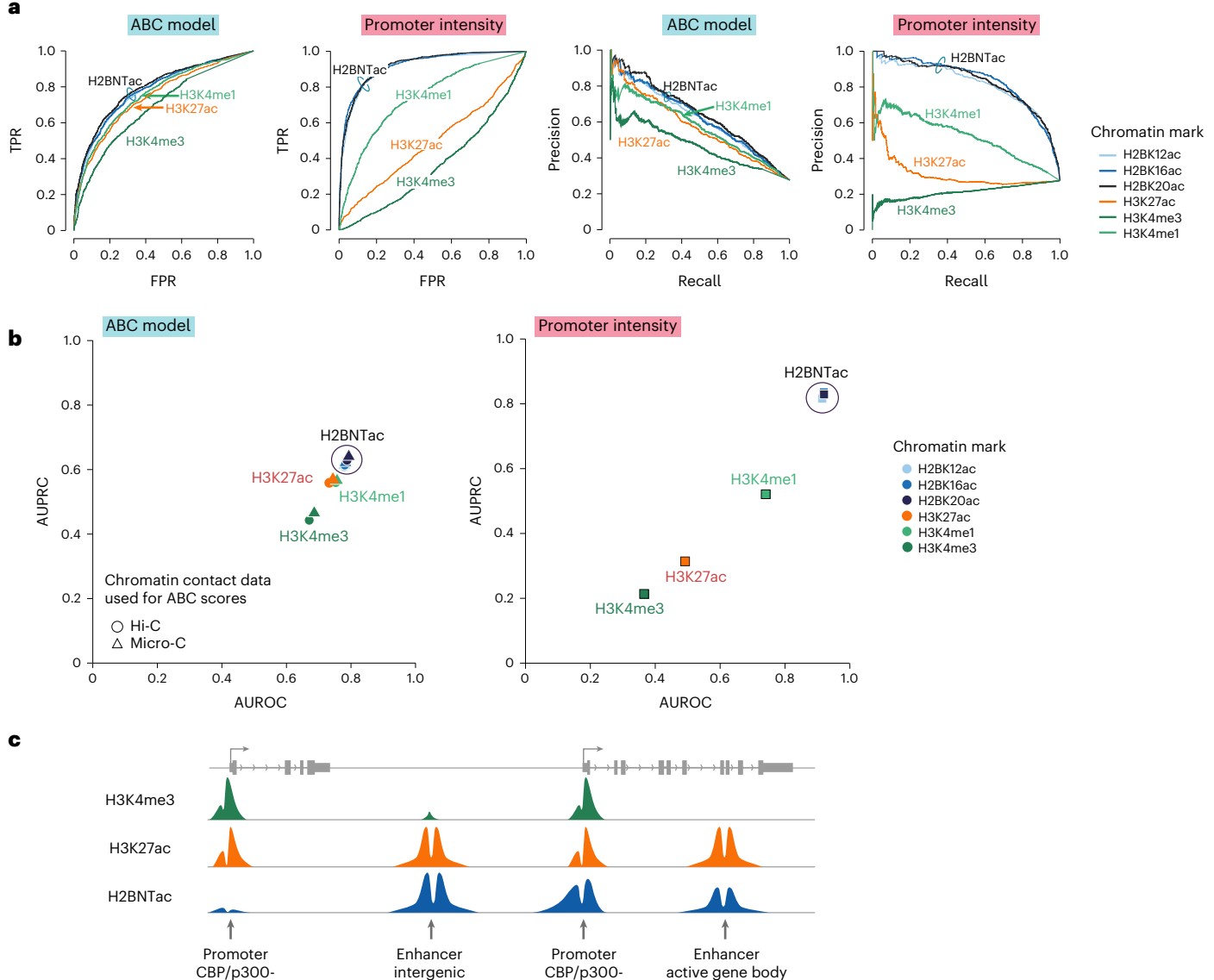

**Fig. 8 | H2BNTac promoter intensity predicts CBP/p300-regulated genes in mESCs. a**, ROC curve (left) and PRC (right) showing the performance of the ABC model and promoter ChIP–seq intensity of the indicated histone marks in discriminating CBP/p300-dependent and independent genes. CBP/p300-dependent (positive list) and CBP/p300-independent genes (negative list) were defined by nascent transcription analyses after acute CBP/p300 inhibition by A-485 (ref. 21). Genes downregulated (twofold or greater; average fold change of 30 min, 1 h and 2 h treatment; Methods) after A-485 treatment are considered CBP/p300-dependent; unaffected genes (<1.2-fold change) are considered CBP/ p300-independent. Performance was evaluated using cumulative nominal ABC scores or ChIP–seq enrichment of H3K4me3, H3K4me1, H3K27ac and H2BNTac in promoter regions (within ±1 kb from the TSS). The ABC score was calculated using ATAC–seq, H3K27ac and Hi-C contact frequency as reported by Fulco et al.[44]. **b**, H2BNTac promoter intensity outperforms the ABC model and other chromatin features in predicting CBP/p300 target genes. Left, AUPRC and AUROC for the ABC model. Right, comparative performance of the indicated chromatin marks. ABC scores, AUPRC and AUROC were determined as indicated in **a**. ABC scores were calculated using either Hi-C- or Micro-C-based contact frequency, as indicated. For the specified chromatin marks, AUPRC and AUROC were calculated using the ChIP signal in promoter regions, as defined in **a**. **c**, Schematic representation of the relative abundance of H3K4me3, H3K27ac and H2BNTac in the indicated genomic regions. H2BNTac and H3K4me3 positively discriminated active candidate enhancers and promoters, respectively.

A notable advantage of H2BNTac is that it does not require 3D contact information to predict CBP/p300 targets, which makes it attractive for use in diverse cell types and tissues. This, in no way, should be construed as enhancer–promoter contact frequency being irrelevant; rather, it highlights the difficulty in faithfully capturing enhancer–promoter contacts that may be weak and highly dynamic. We also acknowledge that the overall correlation ($P = -0.52$ to $-0.65$) between promoter H2BNTac and gene regulation is modest and there is room for future improvements. Globally, the functional impact of endogenous enhancers is impacted by the enhancer–promoter

distance and the autonomous activation strength of promoters[49]. Including the distance-calibrated distal H2BNTac signal, and factoring in autonomous promoter strength, could present future avenues for improving prediction accuracy.

While our work rigorously establishes the specificity of H2BNTac, the biological relevance of H2BNTac remains to be investigated. Hypothetically, a string of dynamically regulated H2BNTac has the potential to exert functional influence by affecting histone–DNA interaction[50], influencing the phase-separation property of chromatin[51], or serving as a ligand for bromodomain proteins, which

preferentially interact with multiply acetylated peptides[52]. These are mere ideas and further work is required to investigate them. We present no claim that H2BNTac is functionally more important than other histone acetylation marks. Because H3K27ac is dispensable for gene activation[53–57], we suggest that the CBP/p300 function in transcription activation should be considered beyond H3K27ac, and H2BNTac merits consideration.

Collectively, our findings provide a unified view of H2BNTac genomic occupancy and its locus-specific regulation. Identification of H2BNTac as a distinctive active enhancer signature will facilitate fine-grained mapping of CREs, better prediction of enhancer strength and function, thereby contributing to an improved understanding of gene regulation.

## Online content

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

## Methods

### Cell culture

E14TG2a Oct4-IRES-Puro mouse ESCs[58] were cultured in a custom-made (C.C.Pro GmbH) N2B27 medium consisting of a 1:1 mix of DMEM/F12 and neurobasal medium but lacking arginine and lysine. Before use, the medium was supplemented with 1,000 U ml$^{-1}$ leukemia inhibitory factor (Merck Millipore), 1 μM PD0325901, 3 μM CT-99021 (custom-made by ABCR GmbH), 100 μM 2-mercaptoethanol, 150 μM sodium pyruvate, 0.5× B27 supplement (Thermo Fisher Scientific), 0.5× N2 supplement (made in house or from Thermo Fisher Scientific) and, where relevant, with L-lysine and L-arginine. Where indicated, cells were treated with A-485 (10 μM). K562 and RPE-1 cells were cultured in DMEM supplemented with 10% FCS and 1% penicillin-streptomycin. All cells were cultured in 5% CO$_2$ at 37 °C.

### EU RNA-seq differential gene expression analysis

Previously published (GSE146328)[21] EU RNA-seq data of ESCs treated with DMSO or A-485 (10 μM, 30, 60 and 120 min) were reprocessed. Trimming of adapters and low-quality sequences (Phred quality score < 20) was performed using Cutadapt v.4.2 (https://doi.org/10.14806/ej.17.1.200). Read sequences were aligned to mm10 (mouse) using the Burrows–Wheeler Aligner (BWA) with default parameters (BWA v.0.7.10)[59]. Multi-mapped reads and reads with more than three mismatches were removed using SAMtools (v.1.4)[60]. Reads mapped to the ribosomal and transfer RNA region obtained from the UCSC genome browser were removed using BEDTools (v.2.23)[61]. For 30, 60 and 120 min time point data, based on the polymerase II elongation rates, maximum 30, 90 and 240 kb gene body regions from the TSS were used for differential gene expression analysis. The number of reads mapped to the defined regions was counted using HTSeq (v.0.11.1)[62]. log$_2$ fold change and $P$ values were calculated at individual time points using DEseq2 (v.1.32.0)[63] with the default scaling method (median of relative abundance). Expressed genes were determined as described in the reference genome annotation section; low-expressed genes were filtered further if the mean EU RNA-seq read count of control and A-485 treatment condition was less than 20.

### ChIP

Cells were cross-linked with 1% formaldehyde for 10 min. After 5 min neutralization with 0.2 M glycine, cells were collected, resuspended in SDS lysis buffer—composed of 10 mM Tris-HCl, pH 8.0, 150 mM NaCl, 1% SDS, 1 mM EDTA, pH 8.0, and cOmplete EDTA-free Protease Inhibitor Cocktail (Sigma-Aldrich)—and fragmented with a Bioruptor Pico (10 cycles, 30 s on and 30 s off, Diagenode). The sonicated solution was diluted (1:5 ratio) with ChIP dilution buffer (20 mM Tris-HCl, pH 8.0, 150 mM NaCl, 1 mM EDTA, 1% Triton X-100) and then used for ChIP; 2–5 μg of antibody was bound to preblocked (with 0.5% BSA) magnetic Dynabeads M-280 sheep anti-Mouse IgG or sheep anti-Rabbit IgG (Thermo Fisher Scientific) and applied to the diluted, sonicated solution for ChIP. Chromatin was added to the antibody–bead complex and incubated by rotating overnight at 4 °C. The beads were washed with ChIP dilution buffer, wash buffer high-salt (20 mM Tris-HCl, pH 8.0, 500 mM NaCl, 2 mM EDTA, 1% Triton X-100, 0.1% SDS), wash buffer low-salt (10 mM Tris-HCl, pH 8.0, 1 mM EDTA, 250 mM LiCl, 0.5% sodium deoxycholate, 0.5% NP-40) and Tris-EDTA buffer. Bound materials were eluted with elution buffer (50 mM Tris-HCl, pH 8.0, 10 mM EDTA, 1% SDS) overnight at 65 °C and treated with RNase A for 30 min at 37 °C and further incubation with proteinase K for 1 h at 55 °C. Then DNA was purified with phenol-chloroform extraction.

### ChIP–qPCR

To validate acetyl histone antibodies for ChIP–seq, we performed ChIP–qPCR to measure the fold enrichment of modified chromatin. Approximately equal amounts of chromatin were mixed with acetyl histone antibodies listed in Supplementary Table 1; then, the relative enrichment of positive (*Nanog* enhancer) and negative (*Hoxa13*) genomic regions was quantified by ChIP–qPCR. Antibodies giving fold enrichment (*Nanog* and *Hoxa13*) greater than five were used for ChIP–seq analysis. The primer sequences for ChIP–qPCR are provided in Supplementary Table 3.

### ChIP–seq library preparation

The ChIP–seq library was prepared using DNA sonicated to an average size of 0.5 kb. ChIP samples were processed for library preparation using the NEBNext Ultra II DNA Library Prep Kit for Illumina (New England Biolabs) according to the manufacturer's instructions and sequenced on a NextSeq 500 Sequencer (Illumina) as single-end 75-bp reads.

### Processing of ChIP–seq data

Mouse and human genome annotation and reference genomes were downloaded from the GENCODE website (mouse: GRCm38 release 25; human: Grch37 release 29)[64]. The transcript type 'protein_coding' was chosen as representative genes. For genes with multiple isoforms, the longest isoform was used for the analyses. Quality checks on sequencing reads were performed using fastqc v.0.11.5. Reads were mapped to the reference genome by using the BWA with default parameters[59]. Multi-mapped reads, duplicated reads or reads with more than three mismatches were removed with SAMtools (v.1.4)[60]. Reads mapped to the DAC Blacklisted Regions (https://www.encodeproject.org/annotations/ENCSR636HFF/) were also omitted from the downstream analysis. The following publicly available datasets were downloaded from the GSE repository and reprocessed for consistency. ESC H3K27ac (GSE135562, GSE160890), K562 H3K9ac (GSE29611), ESC H3K4me1 (GSE146324), ESC NANOG (GSE146324), ESC OCT4 (GSE146324), ESC p300 (GSE146324), ESC H3K36me3 (GSE118785), ESC H3K9me3 (GSE90895), ESC CTCF (GSE178982), K562 H3K4me3 (GSE163049) and K562 H3K4me1 (GSE29611, GSE31755).

Peak regions were called using LanceOtron (v.1.0.8)[29] with the default model (wide-and-deep_jan-2021)[29]. Proximally occurring peaks (within 2 kb) were merged using BEDTools[61] and poorly enriched peaks with a height of less than eight reads per kilobase per million mapped reads (RPKM) were omitted. Peak height was calculated with the bamCompare function (deepTools v.3.5.0)[65] with the following parameters: centerReads, minMappingQuality 10, 20-bp bin, smooth length 400 bp, extended reads 200 bp, RPKM normalization and input-subtracted. The region with the maximum peak height within each peak locus was defined as a peak summit region. Where indicated, peak regions were classified into the following classes: promoter (TSS ± 1 kb); gene body (exon, intron, 5′-UTR, and 3′-UTR, excluding promoter regions); and intergenic (outside promoter and gene body) by using ChIPseeker (v.1.28.3)[66]. The mESC superenhancer region BED file was downloaded from dbSuper[67] and converted from mm9 to mm10 using the UCSC LiftOver function[68]. For the visualization of the ChIP signal, BIGWIG files were generated by bamCompare with the following parameters: centerReads, minMappingQuality 10, 20-bp bin, smooth length 400 bp, extended reads 200 bp, RPKM normalization and input-subtracted. For the visualization of ATAC and the EU RNA-seq signal, BIGWIG files were generated by function (deepTools)[65] with the following parameters: centerReads, minMappingQuality 10, 20-bp bin, smooth length 400 bp, extended reads 200 bp, RPKM normalization). The integrated genome viewer v.2.10.3 was used to visualize gene tracks.

### ChIP–seq coverage analysis

In the ChIP–seq coverage analysis, genomes were binned by a 2-kb window and the ChIP–seq read count was calculated using HTSeq[62]. Read number was normalized using reads per million mapped reads (RPM) and input-subtracted values were used. For regions with negative values, the values were substituted with 0. Unless indicated otherwise, to compare peak enrichment in the defined peak regions, a 2-kb

window of an input-subtracted RPM value greater than 1 was chosen and a $\log_2$ ratio of RPM + 0.5 was used to calculate differential ChIP coverage ratios. For the analysis of promoters, promoter regions were defined as regions in ±1 kb around the H3K4me3-marked TSS of actively transcribed genes (TPM ≥ 2), instead of a genome-wide 2-kb bin. TPM values are calculated from EU RNA-seq data (for mESC, GSE146328)[21] or RNA-seq data (for the K562 cell line).

## Peak overlap

In counting the overlap between ATAC peak, H3K27ac and H2BNTac peak regions, the findOverlaps function (GenomicAlignments v.1.8.4)[69], which allowed maximum 2-kb gaps, was used. Peak numbers in the overlapping regions are not identical between two peak sets because several peaks in one set can overlap with one peak in another set. The overlapping peak numbers shown in the figures are the peak numbers of H2BNTac (when compared with H3K27ac or ATAC peaks) or the peak numbers of a set with more overlapping peaks (when compared with H2BNTac). A Mann–Whitney $U$-test was used for testing the significance of peak height difference between overlapping and nonoverlapping peaks. When annotating peaks using OCT4 and NANOG bindings (Supplementary Fig. 6), STARR-seq and ATAC peaks, PINTS region and DHS regions, no gaps were allowed to overlap.

## Enrichment of ChromHMM states in peak regions

mESC 18 ChromHMM states were downloaded from the ENCODE portal[70]. We assigned chromatin states at each peak summit region as representative peak states. In the enrichment analysis, the expected peak number in each state was calculated by assigning an equal number of randomly sampled genomic regions; the enrichment of chromatin states was defined by the ratio of the observed peak number to the expected peak number.

## Chromatin states with and without H2BNTac chromatin marks

We used a 15-state ChromHMM model to predict chromatin states in mESCs using ChromHMM (v.1.22)[31]. The following data were used as input: ATAC; H3K4me3; H3K4me1; H3K36me3 (GSE118785); H3K9ac; H3K9me3 (GSE90895); H3K27me3; CTCF (GSE178982); and H3K27ac, with and without including H2BNTac ChIP samples. The ChIP peak enrichment calculation included corresponding input samples. For consistency reasons, all ChIP data were processed using the same pipeline described above. We compared emission parameters between the predicted chromatin states in the presence and absence of H2BNTac peaks.

## Overlap of MPRA regions with peak regions

ESC MPRA regions were downloaded from the supplementary files of Peng et al.[37] and lifted over to mm10 using the UCSC LiftOver function[68]. We used union STARR regions identified in 2i or SL condition. Overlap between MPRA regions with a ChIP–seq peak is calculated using distanceToNearest function (GenomicRanges v.1.44.0)[69].

## Overlap of PINTS regions with peak regions

Annotations of proximal and distal PINTS elements for human cells were downloaded from the PINTS web portal (https://pints.yulab.org/summary_stats) and lifted over to hg19 using the UCSC LiftOver function[68]. Using these data, we generated a reference dataset of PINTS regions identified in 110 cell lines and tissues and merged proximally overlapping regions.

We classified the PINTS regions into active and inactive in K562 cells using the following criteria: (1) proximal active: proximal PINTS regions of K562 cells; (2) proximal inactive: proximal PINTS regions that are active in other cell types but not in K562 and occurred >2 kb away from any K562 proximal active PINTS regions; (3) distal active: distal PINTS regions that are active in K562 cells; (4) distal inactive: distal PINTS regions that are active in other cell types but not active

in K562. The following inactive regions were excluded: (1) inactive regions that overlapped with any of the proximal PINTS regions in K562; (2) inactive regions located within 5 kb of active TSS in K562; and (3) inactive regions located within 2 kb from distal active PINTS regions in K562 cells. The overlap between PINTS regions and ChIP–seq peak was calculated using the distanceToNearest function[69].

## Discriminative power assessment of PINTS regions

For each set of positive and negative PINTS regions, enrichment of the ChIP signal was calculated from input-subtracted RPM values. We computed the average ChIP signal by varying the window (±250, 500, 1,000 bp from the center of the PINTS region). The discriminatory potential of different histone marks was analyzed by comparing ChIP signal enrichment at different PINTS regions. ROCs and PRCs were generated under the different thresholds of peak enrichment or peak enrichment ratios. ROCs, PRCs and area under the curve (AUC) were computed using the PRROC R package (v.1.3)[71].

## Discriminative power assessment of CBP/p300-regulated genes

ABC scores were calculated using the ABC model pipeline (https://github.com/broadinstitute/ABC-Enhancer-Gene-Prediction). ATAC–seq data (GSE146328) were used to call candidate enhancer regions, and the geometric mean of ATAC–seq and H3K27ac intensities, or ATAC–seq and H2BNTac intensities, were used to quantify enhancer activity. For consistency reasons, we used in-house-generated H3K27ac and H2BKN-Tac ChIP–seq data to calculate the ABC scores. Hi-C (GSE118911) or Micro-C (GSE130275) were used to measure enhancer target accessibilities in mESCs. Hi-C data were reprocessed to map to the mm10 genome using JUICER (v.1.6)[72] with filtering of mapping quality greater than 30. Processed Micro-C data were downloaded from the Gene Expression Omnibus (GEO) repository (GSE130275_mESC_WT_combined_2.6B.hic). To calculate the ABC model-based enhancer contribution, nominal scores (powerlaw.Score.Numerator) were summed up for each target gene. For the genes with no ABC scores or a score less than 0.01, an ABC score of 0.01 was imputed. In calculating the enhancer contribution based on the histone mark intensities, the ChIP read enrichment at the promoter regions (±1 kb from the TSS) was used. As a positive and negative set of CBP/p300-regulated genes, greater than twofold down-regulated genes and less than 20% changes on A-485 treatment based on EU RNA-seq were selected. For a fair comparison, genes included in the processed data by Fulco et al.[44] were analyzed. ROC and PRC were generated by using different thresholds of peak enrichment or cumulative ABC score. ROC, PRC and AUC were computed using PRROC R package[71].

## Evaluation of H3K27ac and H2BK20ac performance for predicting enhancer targets using the ABC model

To evaluate H3K27ac and H2BK20ac performance in the context of the ABC model, we used two CRISPRi enhancer target datasets that were independently generated by Fulco et al.[44] and Gasperini et al.[47]. Both datasets were generated using K562 cells. To calculate the ABC scores in K562 cells, we used the same DHS and Hi-C data as Fulco et al.[44] and used in-house-generated H3K27ac or H2BK20ac ChIP–seq data. ABC scores were calculated using the ABC model pipeline. To test the ABC model in the Fulco et al.[44] dataset, we downloaded the processed data (41588_2019_538_MOESM3_ESM.xlsx) and used positive and negative gene lists as defined by Fulco et al.[44] For the Gasperini et al.[47] dataset, we downloaded the processed data from GSE120861_all_deg_results.at_scale.txt and defined positive and negative lists as follows: positive list: enhancer–gene interactions, which were defined by the authors as high-confidence interactions (that is, where CRISPRi-induced enhancer silencing caused significant gene downregulation); and negative list: enhancer–gene interactions where target genes were not significantly downregulated (remaining transcript abundance greater than 95%).

Because of chromosomal translocation, we excluded interactions on chromosome 9. When calculating enhancer contribution, a summed ABC score was used if multiple enhancers overlapped with targeted regions[44]. To calculate the AUROC and AUPRC, only interactions with valid ABC scores were considered. ROC and PRC were generated by using different thresholds of the ABC score. ROC, PRC and AUC were computed using the PRROC R package[71].

## Statistical analysis

Unless otherwise stated, *P* values were calculated using a Mann–Whitney *U*-test and corrected for multiple comparisons using the Benjamini–Hochberg method (R package stats v.3.6.2).

## Analysis of publicly available histone acetylation ChIP–seq projects in the GEO

The openly accessible ChIP–seq data was retrieved from the NCBI GEO using the R package reutils v.0.2.3 using the query 'peak'[All Fields] AND gse[Filter] AND Genome binding/occupancy profiling by high throughput sequencing[Filter]', where 'peak' was replaced with histone acetylation marks (that is, H3K27ac, H2BK5ac, H2BK11ac, H2BK12ac, H2BK15ac, H2BK16ac or H2BK20ac). Data deposited from the year 2009 to 2021 were retrieved. After a manual check of duplicates and availability of datasets, the number of projects belonging to each dataset was counted by the year when the data became public.

## Acetylation of histone peptides

Unmodified peptides corresponding to histones H2B and H3 N termini were synthesized (Schafer-N) with C-terminal biotinylated lysine. The peptide was dissolved in PBS to a concentration of 2 mg ml$^{-1}$. Fully acetylated forms of H2B and H3 N-terminal peptides were generated by in vitro chemical acetylation with Sulfo-NHS-acetate. Then, 200 µg peptide was mixed with 40 µl (20 µg µl$^{-1}$) Sulfo-NHS-acetate in acetonitrile and incubated for 30 min at room temperature. Excess Sulfo-NHS-acetate in the reaction was quenched by the addition of Tris-HCl, pH 7.5, to a final concentration of 100 mM. To check for completion of the reaction, 2 µl of the product was acidified and desalted with C18-Stage tips, eluted, dried and redissolved in water with 0.1% formic acid. Acetylation was confirmed by mass spectrometry (Orbitrap Exploris 480 mass spectrometer). Sequence information of the histone peptides used to analyze antibody specificity is provided in Supplementary Table 2.

## Immunostaining

hTERT RPE-1 cells (catalog no. CRL-4000, ATCC) were seeded onto 12-mm coverslips distributed in a 6-cm dish and allowed to attach overnight. The day after, cells were fixed with 4% formaldehyde in PBS for 15 min and washed once with PBS. Coverslips were placed into a 24-well plate. Cells were permeabilized with 0.5% Triton X-100 in PBS for 5 min followed by blocking in antibody diluent (sterile filtered DMEM with 10% FCS and 0.02% sodium azide) for 30 min. The primary antibody was diluted 1:1,000 in the antibody diluent and split into three 200-µl stocks. In one, no additive was included; in the second 0.2 µl of 0.67 mg ml$^{-1}$ unmodified histone peptide was added; and in the third 0.2 µl of 0.67 mg ml$^{-1}$ acetylated histone peptide was added. Two sets of coverslips for each antibody were then stained for 2 h with the antibody without or with peptide. The coverslips were washed three times with PBS and stained with a secondary anti-rabbit Alexa Fluor 488 antibody diluted (1:500) in the antibody diluent for 1 h. The coverslips were washed once and stained with Hoechst 33342 (1 µg ml$^{-1}$) in PBS for 10 min followed by two PBS washes, dipping into water and mounting on slides using 5 µl Fluoromount-G.

## Image-based cytometry

Images for image-based cytometry were acquired on an Olympus ScanR automated widefield screening microscope with 12-bit dynamics

on a 16-bit Hamamatsu ORCA-Flash4.0 2,048 × 2,048 pixel camera with 6.5 µm and pixel size using an UPLSAPO ×20 objective with 0.75 numerical aperture. Images were analyzed using Olympus ScanR Image Analysis software v.2.8. Nucleus segmentation was carried out with the integrated intensity-based object detection using the Hoechst signal; after background correction, the total and mean intensities of the two channels were calculated for each object. Further analysis was done on properly segmented cells gated for having a 2C–4C DNA content based on total and mean DAPI intensities. Data tables from the ScanR Analysis were further processed and visualized in R using ggplot2. Three thousand randomly sampled cells from each coverslip were included for each experiment and repetition. Example images from each coverslip with set contrast and brightness settings for each antibody are displayed with their corresponding quantifications.

## Generation of HDAC 1 and 2-GFP-FKBP12$^{F36V}$ cells

The GFP-FKBP12$^{F36V}$ tag was knocked in at the C terminus of the endogenous *Hdac1* and *Hdac2* using the CRISPR–Cas9 system. Mouse ESCs were cotransfected with the pX330 plasmid and a donor plasmid containing a resistance selection gene (*Hdac1*: Puro; *Hdac2*: Neo) flanked by homology arms (approximately 500 bp each side from the start of the stop codon of the target genes).

## Sequences of guide RNAs for targeting *Hdac1* and *Hdac2*

Puromycin or neomycin-resistant cell clones were screened by genomic PCR using KOD Xtreme Hot Start DNA polymerase (Merck Millipore). The sequences of the guide RNAs used, the homology arms and genotyping primers are provided in Supplementary Table 3 and the Source Data file.

## Confirmation of HDAC 1 and 2-GFP-FKBP12$^{F36V}$ depletion

Cells were seeded at 15,000 per well in PerkinElmer Cell Carrier Ultra PhenoPlate 96-well, black, optically clear flat-bottom plates coated with Thermo Fisher Geltrex LDEV-Free Reduced Growth Factor Basement Membrane Matrix (Thermo Fisher Scientific). After seeding, cells were grown for 2 d. Cells were treated with either d-TAG13 (100 nM) or DMSO as solvent control. Finally, cells were fixed using 4% PFA in 1× PBS and blocked with 5% BSA and 0.3% Triton X-100 in 1× PBS. Cells were counterstained with 1:1,000 Hoechst 33342. GFP-488 and Hoechst 350 fluorescence signals were acquired using an Opera Phenix Plus High-Content Screening System. The fluorescence signals were obtained from 15 fields of three wells and two plates. Fluorescence intensity levels were analyzed based on nuclear and area detection using the Hoechst 350 channel, and the GFP-488 mean intensity levels within this area as the principal read out. Data analysis was performed at a single-cell level. Images were analyzed using the Harmony High-Content Imaging and Analysis Software, v.4.9.

## Immunoblotting

Cells were lysed on the plate with radioimmunoprecipitation assay buffer (25 mM Tris, pH 7.6, 150 mM NaCl, 1% (w/v) NP-40, 0.1% (w/v) SDC, 1 mM EDTA) containing protease inhibitor (cOmplete Protease Inhibitor Cocktail, Roche). Lysates were collected and sonicated with a Bioruptor (Diagenode) for five cycles with 45 s on and 30 s off. The lysate was spun down for 10 min at 4 °C and 16,100*g*. The supernatant was kept and the pellet was discarded. Protein concentration was determined using the Bradford assay (Bio-Rad Laboratories) according to the manufacturer's instructions. From each sample, 25 µg of protein was mixed with LDS Sample Buffer (NuPAGE) to a 1× concentration and the sample was boiled at 95 °C for 10 min before separation on 5–12% Bis-Tris gel. Proteins were transferred onto a polyvinylidene fluoride membrane, the membrane was blocked in 5% (w/v) skimmed milk (Sigma-Aldrich) in TBS-Tween 20 (20 mM Tris, 150 mM NaCl, 0.1% (w/v) Tween 20) for 1 h. The membrane was subsequently washed three times with TBS-Tween 20 for 15 min. Then, the membrane was cut and

the pieces were either incubated with rabbit monoclonal antibody against acetylated H2BK20 (Acetyl-Histone H2B (Lys20) (D709W) Rabbit monoclonal antibody, catalog no. 34156, Cell Signaling Technology) or H3 (Histone H3 Rabbit monoclonal antibody no. 4499, Cell Signaling Technology) in 5% BSA (Sigma-Aldrich) in TBS-Tween 20 overnight at 4 °C. The antibody solution was removed and the membranes were washed three times in TBS-Tween 20 for 15 min and then incubated with peroxidase-conjugated anti-rabbit antibody (Peroxidase AffiniPure F(ab')₂ Fragment Goat Anti-Rabbit IgG (H+L), Jackson ImmunoResearch) in 5% skimmed milk in TBS-Tween 20 for 1 h at room temperature. The antibody solution was removed and the membranes were washed four times in TBS-Tween 20 for 1 h in total. Blots were developed by incubating the membranes with 2 ml enhanced chemiluminescence solution (SuperSignal West Pico PLUS Chemiluminescent Substrate, Thermo Fisher Scientific) at room temperature for subsequent detection with a chemiluminescence film (High Performance Chemiluminescence Film, GE Healthcare).

### Reporting summary

Further information on research design is available in the Nature Portfolio Reporting Summary linked to this article.

## Data availability

This project's sequencing raw data, processed peak regions, and gene tracks are available on the NCBI GEO under accession no. GSE186349. Additionally, the following datasets were downloaded and analyzed: gene annotation for human and mice (https://www.gencodegenes.org); gene expression profiles based on promoter transcripts, FANTOM5 CAGE dataset, array express (http://www.ebi.ac.uk/arrayexpress/); ENCODE DAC Blacklisted Regions (https://www.encodeproject.org/annotations/ENCSR636HFF/); super enhancer regions, dbSuper (http://bioinfo.au.tsinghua.edu.cn/dbsuper/); genome states annotation, ESC 18 chromHMM states (https://www.encodeproject.org/search/?searchTerm=ChromHMM+Zhiping+Weng); K562 25 ChromHMM states (http://hgdownload.cse.ucsc.edu/goldenpath/hg19/encodeDCC/wgEncodeAwgSegmentation); transcription-supported enhancers and promoters, PINTS elements (https://pints.yulab.org/summary_stats); MPRA-defined candidate enhancers, mESC STARR-seq peaks were obtained from the supplementary data section of Peng et al.[37] (13059_2020_2156_MOESM4_ESM.xlsx); regulatory interactions between enhancers and genes based on CRISPRi perturbations, K562 CRISPRi data from Fulco et al.[44] (supplementary data section 41588_2019_538_MOESM3_ESM.xlsx) and Gasperini et al.[47] (supplementary data GSE120861_all_deg_results.at_scale.txt). We reanalyzed the following publicly available sequencing datasets: ESC H3K27ac (nos. GSE135562, GSE160890); K562 H3K9ac (no. GSE29611); ESC H3K4me1 (no. GSE146324); ESC NANOG (no. GSE146324); ESC OCT4 (no. GSE146324); ESC p300 (no. GSE146324); ESC H3K36me3 (no. GSE118785); ESC H3K9me3 (no. GSE90895); ESC CTCF (no. GSE178982); K562 H3K4me3 (no. GSE163049); K562 H3K4me1 (nos. GSE29611 and GSE31755); ESC RNA-seq (no. GSE146324); ESC Hi-C (no. GSE118911); K562 Hi-C (no. GSE63525); ESC Micro-C (no. GSE130275); ESC DNase-seq (no. GSE37074); and K562 DNase-seq (no. GSE29692).

## Code availability

We did not use any unique code or algorithm in this study that is central to our conclusions. The methods used for the analyses are described in the main text and Methods.

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

## Acknowledgements

We thank the members of the Choudhary laboratory for their helpful discussions. We thank E. Maskey for her excellent technical assistance. The Novo Nordisk Foundation Center for Protein Research is financially supported by the Novo Nordisk Foundation (no. NNF14CC0001). C.C. was supported by the Novo Nordisk Foundation (nos. NNF14OC0008541 and NNF22OC0074677). This work was funded by the European Commission FP7 grant (no. SyBoSS FP7-242129). Y.H. was supported by a Grant-in-Aid for JSPS Overseas Postdoctoral Fellows. S.K. was supported by a Lundbeck Foundation Fellowship (no. R347-2020-2170). We thank the SyBoSS partners K. Anastassiadis, F. Steward, A. Smith and W. Skarnes for sharing E14TG2a Oct4-IRES-Puro mouse ESCs. We thank Abcam, Cell Signaling Technology, Thermo Fisher Scientific, Active Motif and RevMAb Biosciences for providing some of the antibody samples for testing. We thank the CPR Imaging Platform, the CPR Big Data Management Platform and the CPR and DanStem Genomics Platform for their assistance. We thank the ENCODE Consortium and the ENCODE production laboratory for generating and sharing the data. We also thank Fulco et al.[44] and Gasperini et al.[47] for providing unrestricted access to their data.

## Author contributions

C.C. and T.N. conceived the project. T.N., Y.H., S.K., T.L. and J.W. designed the research, performed the experiments, analyzed the data and interpreted the results. C.C. supervised the project. T.N. and C.C. wrote the manuscript with input from all coauthors.

## Competing interests

The authors declare no competing interests.

## Additional information

**Correspondence and requests for materials** should be addressed to Chunaram Choudhary.

| | |
|---|---|

# Reporting Summary

## Statistics

For all statistical analyses, confirm that the following items are present in the figure legend, table legend, main text, or Methods section.

| n/a | Confirmed | |
|---|---|---|
| ☐ | ☒ | The exact sample size (*n*) for each experimental group/condition, given as a discrete number and unit of measurement |
| ☐ | ☒ | A statement on whether measurements were taken from distinct samples or whether the same sample was measured repeatedly |
| ☐ | ☒ | The statistical test(s) used AND whether they are one- or two-sided<br>*Only common tests should be described solely by name; describe more complex techniques in the Methods section.* |
| ☐ | ☒ | A description of all covariates tested |
| ☐ | ☒ | A description of any assumptions or corrections, such as tests of normality and adjustment for multiple comparisons |
| ☐ | ☒ | A full description of the statistical parameters including central tendency (e.g. means) or other basic estimates (e.g. regression coefficient) AND variation (e.g. standard deviation) or associated estimates of uncertainty (e.g. confidence intervals) |
| ☐ | ☒ | For null hypothesis testing, the test statistic (e.g. *F*, *t*, *r*) with confidence intervals, effect sizes, degrees of freedom and *P* value noted<br>*Give P values as exact values whenever suitable.* |
| ☒ | ☐ | For Bayesian analysis, information on the choice of priors and Markov chain Monte Carlo settings |
| ☒ | ☐ | For hierarchical and complex designs, identification of the appropriate level for tests and full reporting of outcomes |
| ☐ | ☒ | Estimates of effect sizes (e.g. Cohen's *d*, Pearson's *r*), indicating how they were calculated |

*Our web collection on statistics for biologists contains articles on many of the points above.*

## Software and code

Policy information about availability of computer code

| | |
|---|---|
| Data collection | The statistics of ChIP-seq projects in GEO: reutils (version 0.2.3). |
| Data analysis | Sequencing analysis: Cutadapt(4.2), FastQC(0.11.5), Bedtools(2.23), HTseq(0.11.1), BWA(0.7.10), samtools(1.4), LanceOtron(1.0.8), R(4.1.1), PRROC (1.3), ChIPseeker (1.28.3), GenomicAlignments (1.8.4), GenomicRanges (1.44.0), stats (3.6.2), ggplot2(3.3.5),  DESeq(1.32.0), ChIPseeker(l.28.3),  msa(1.24.0), deeptools(3.5.0), JUICER (1.6), IGV(2.10.3), ChromHMM(1.22), ABC model pipeline (https://github.com/broadinstitute/ABC-Enhancer-Gene-Prediction).  PRROC (1.3), ChIPseeker (1.28.3), GenomicAlignments (1.8.4), GenomicRanges (1.44.0).<br><br>Microscopic analysis: Harmony High-Content Imaging and Analysis (4.9), Olympus ScanR Image Analysis(2.8). |

For manuscripts utilizing custom algorithms or software that are central to the research but not yet described in published literature, software must be made available to editors and reviewers. We strongly encourage code deposition in a community repository (e.g. GitHub). See the Nature Portfolio guidelines for submitting code & software for further information.

## Data

Policy information about availability of data

All manuscripts must include a data availability statement. This statement should provide the following information, where applicable:
- Accession codes, unique identifiers, or web links for publicly available datasets
- A description of any restrictions on data availability
- For clinical datasets or third party data, please ensure that the statement adheres to our policy

This project's sequencing raw data, processed peak regions, and gene tracks are available on the National Center for Biotechnology Information Gene Expression Omnibus (GEO) under accession number GSE186349.

Additionally, the following datasets are downloaded and analyzed.
Gene annotation for human and mice: (https://www.gencodegenes.org)
Gene expression profiles based on promoter transcripts: FANTOM5 CAGE dataset, array express (http://www.ebi.ac.uk/arrayexpress/)
Encode DAC Blacklisted Regions (https://www.encodeproject.org/annotations/ENCSR636HFF/)
Super enhancer regions: dbSuper (http://bioinfo.au.tsinghua.edu.cn/dbsuper/)
Genome states annotation: ESC 18 chromHMM states (https://www.encodeproject.org/search/?searchTerm=ChromHMM+Zhiping+Weng)
K562 25 chromHMM states (http://hgdownload.cse.ucsc.edu/goldenpath/hg19/encodeDCC/wgEncodeAwgSegmentation)
Transcription-supported enhancers and promoters: PINTS elements (https://pints.yulab.org/summary_stats)
Plasmid-based assay-supported enhancers: ESC STARR-seq peaks (Supplemental data section of Peng et.al (PMID: 32912294))
Plasmid-based assay-supported autonomous promoters: K562 SuRE peaks (Supplementary section of van Arensbergen et al. (PMID: 28024146))
Regulatory interactions between enhancers and genes based on CRISPRi perturbations: K562 CRISPRi screen (Supplemental data section of Fulco et.al (41588_2019_538_MOESM3_ESM.xlsx, PMID: 31784727), Supplemental data of Gasperini et al. (GSE120861_all_deg_results.at_scale.txt, PMID: 30612741)

We re-analyzed the following publicly available sequencing datasets.
ESC H3K27ac (GSE135562, GSE160890), K562 H3K9ac (GSE29611), ESC H3K4me1 (GSE146324), ESC NANOG (GSE146324), ESC OCT4 (GSE146324), ESC p300 (GSE146324), ESC H3K36me3 (GSE118785), ESC H3K9me3 (GSE90895), ESC CTCF(GSE178982), K562 H3K4me3 (GSE163049), and K562 H3K4me1 (GSE29611, GSE31755). ESC RNA-seq (GSE146324). ESC Hi-C (GSE118911), K562 Hi-C (GSE63525), ESC Micro-C (GSE130275), ESC DNase-Seq (GSE37074), K562 DNase-Seq (GSE29692)

# Field-specific reporting

Please select the one below that is the best fit for your research. If you are not sure, read the appropriate sections before making your selection.

☒ Life sciences      ☐ Behavioural & social sciences      ☐ Ecological, evolutionary & environmental sciences

For a reference copy of the document with all sections, see nature.com/documents/nr-reporting-summary-flat.pdf

# Life sciences study design

All studies must disclose on these points even when the disclosure is negative.

| | |
|---|---|
| Sample size | Image-based cytometry: 3000 randomly sampled cells from two independent experiments were analyzed. |
| Data exclusions | Image-based cytometry: Cells that did not show 2C-4C DNA based on total and mean DAPI intensities were excluded. |
| Replication | We generally performed at least two biologically independent experiments per antigen and conditions. For the following ChIP-seq experiments, single replicate was performed: H2BK5ac(ab40866), H2BK11ac, H3K9ac, H3K4me3, H3K27me3 ChIP in ESC ; H3K27ac ChIP in K562 cells on control condition, or A-485 5 min treatment, and H2BK20ac ChIP in ESC on NVP-2 120 min treatment condition. Immunoblotting, flow cytometry experiments were repeated at least twice independently. All attempts to replicate the results were successful. |
| Randomization | Samples were grouped according to antigen and experimental condition, and each group was analyzed individually.  A-485 or NVP-2 treated samples were paired with the corresponding controls. |
| Blinding | The investigators were not blinded because the study did not involve any clinical samples. All experiments were done in cell lines, and blinding was not necessary as the data were mainly generated by digital reading. For all samples, data analysis was performed using the same pipeline. |

# Reporting for specific materials, systems and methods

We require information from authors about some types of materials, experimental systems and methods used in many studies. Here, indicate whether each material, system or method listed is relevant to your study. If you are not sure if a list item applies to your research, read the appropriate section before selecting a response.

## Materials & experimental systems

| n/a | Involved in the study |
|---|---|
| ☐ | ☒ Antibodies |
| ☐ | ☒ Eukaryotic cell lines |
| ☒ | ☐ Palaeontology and archaeology |
| ☒ | ☐ Animals and other organisms |
| ☒ | ☐ Human research participants |
| ☒ | ☐ Clinical data |
| ☒ | ☐ Dual use research of concern |

## Methods

| n/a | Involved in the study |
|---|---|
| ☐ | ☒ ChIP-seq |
| ☒ | ☐ Flow cytometry |
| ☒ | ☐ MRI-based neuroimaging |

# Antibodies

| Antibodies used | ChIP-seq<br>Antibody Antibody supplier Cat# Clone ID/ Lot volume of antibody for each ChIP experiment (ug or ul)<br>anti-H2BK5ac Abcam ab40886 rabbit EP857Y 2ug<br>anti-H2BK5ac Cell Signaling Technology 12799S rabbit D5H1S 10ul<br>anti-H2BK11ac ReMAb Biosciences 31-1348-00 rabbit RM456 3ug<br>anti-H2BK12ac Abcam ab40883 rabbit EP858Y 2ug<br>anti-H2BK16ac Abcam ab177427 rabbit EPR17598 2ug<br>anti-H2BK20ac Abcam ab177430 rabbit EPR859 3.5ug<br>anti-H2BK20ac Cell Signaling Technology 34156 rabbit D7O9W 10ul<br>anti-H3K27ac Abcam ab4729 rabbit 2ug<br>anti-H3K9ac CST #9671 rabbit 10ul<br>anti-Med1 Millipore 17-10530 rabbit 3.45ug<br>anti-H3K4me3 Millipore 04-745 rabbit MC315  5ul<br>anti-H3K27me3 CST #9733 rabbit C36B11 5ul<br><br>Immunostaining<br>Antibody Antibody supplier Cat# Clone ID/ Lot Dilution<br>anti-H2BK5ac Active motif 39124 rabbit 229 1:1000<br>anti-H2BK5ac abcam ab40886 rabbit GR111987-5 1:1000<br>anti-H2BK5ac Cell Signaling Technologies 12799S rabbit  09/2020 lot 1 1:1000<br>anti-H2BK20ac Cell Signaling Technologies 34156 rabbit  04/2020 1:1000<br>anti-H2BK20ac abcam ab177430 rabbit  GR3196303-2 1:1000<br>anti-H2BK120ac Upstate 07-564 rabbit  26872 1:1000<br>anti-H2BK120ac Active motif 39119 rabbit 1008001 1:1000<br>anti-H3K27ac abcam ab177178 rabbit GR202987-5 1:1000<br>anti-GFP abcam ab290 rabbit  70828032ab 1:250<br>Alexa Fluor A488 anti-rabbit Thermo Fisher A11034 rabbit 2110499 1:500, or 1:1000<br><br>Western blotting<br>Antibody Antibody supplier Cat# Clone ID/ Lot Dilution<br>Peroxidase AffiniPure F(ab')₂ Fragment Goat anti-Rabbit IgG (H+L) Jackson Immuno Research 111-036-045 goat  107341 1:5000<br>anti-H2BK20ac Cell Signaling Technology 34156 rabbit D7O9W 1:1000<br>anti-H3 Cell Signaling Technologies 4499 rabbit 9 1:1000 |
| --- | --- |
| Validation | Anti-H2BK5ac Abcam ab40886: This antibody cross-react with H3K27ac as shown in this manuscript. The manufacturer has also added this information in the datasheet for this antibody (https://www.abcam.com/histone-h2b-acetyl-k5-antibody-ep857y-chip-grade-ab40886.html).<br><br>Anti-H2BK5ac Cell Signaling Technology 12799S: According to the manufacturer's website, Acetyl-Histone H2B (Lys5) (D5H1S) XP® Rabbit mAb recognizes endogenous levels of histone H2B only when acetylated at Lys5. There is no cross-reactivity between this antibody and other acetylated histones. (https://www.cellsignal.com/products/primary-antibodies/acetyl-histone-h2b-lys5-d5h1s-xp-rabbit-mab/12799)<br><br>Anti-H2BK5ac Active motif 39124: According to the antibody datasheet, this antibody was validated by modENCODE and NIH Roadmap Epigenomics Mapping Consortiums. Our results indicate that this antibody also cross-reacts with H3K27ac.<br><br>anti-H2BK11ac ReMAb Biosciences 31-1348-00: According to the manufacturer's page, this antibody reacts to Histone H2B acetylated at Lysine11 (K11ac). No cross-reactivity with other acetylated Lysines in histones. (https://www.revmab.com/index.php/product/anti-acetyl-histone-h2b-lys11-rabbit-monoclonal-antibody-clone-rm456/)<br><br>anti-H2BK12ac Abcam ab40883: According to the manufacturer's page, this antibody only detects Histone H2B acetylated on Lysine 12. (https://www.abcam.com/histone-h2b-acetyl-k12-antibody-ep858y-chip-grade-ab40883.html)<br><br>anti-H2BK16ac Abcam ab177427: According to the manufacturer, this antibody selectively recognize H2BK16ac. The specificity of ab177430 was tested by the manufacturer in a Peptide array against 501 different modified and unmodified histone peptides; each peptide is printed on the array at six concentrations (https://www.abcam.com/histone-h2b-acetyl-k16-antibody-epr17598-chip-grade-ab177427.html).<br><br>anti-H2BK20ac Abcam ab177430: According to the manufacturer, this antibody selectively recognize H2BK20ac. The specificity of ab177430 was tested in a Peptide array against 501 different modified and unmodified histone peptides; each peptide is printed on the array at six concentrations (https://www.abcam.com/histone-h2b-acetyl-k20-antibody-epr859-chip-grade-ab177430.html)<br><br>anti-H2BK20ac Cell Signaling Technology 34156: According to the manufacturer's page, Acetyl-Histone H2B (Lys20) (D7O9W) Rabbit mAb recognizes endogenous levels of histone H2B protein when acetylated at Lys20. This antibody shows very slight cross-reactivity with histone H2B acetylated at Lys12.(https://www.cellsignal.com/products/primary-antibodies/acetyl-histone-h2b-lys20-d7o9w-rabbit-mab/34156)<br><br>anti-H3K27ac Abcam ab4729: According to the manufacturer, all batches of ab4729 are tested in Peptide Array against peptides to different Histone H3 modifications. Results show strong binding to Histone H3K27ac peptide.<br><br>anti-H3K27ac Abcam ab177178: According to the manufacturer, ab177178 was tested in a Peptide array against 501 modified and |

unmodified histone peptides. Results show strong binding to Histone H3K27ac peptide.

anti-H2BK120ac Upstate 07-564: According to the manufacturer, this antibody recognizes H2BK120ac. Our results indicate that this antibody also cross-reacts with H2BK20ac.

anti-H2BK120ac Active motif 39119: According to the manufacturer, this antibody recognizes H2BK120ac. Our results indicate that this antibody also cross-reacts with H2BK20ac.

anti-H3 Cell Signaling Technologies 4499: The antibody is validated by the manufacturer, and according to the antibody datasheet, this antibody has been used in >1000 papers.

anti-GFP abcam ab290:  The antibody is validated by the manufacturer, and according to the antibody datasheet, this antibody has been used in >2500papers.

H3K27me3 CST #9733: According to the manufacturer's page,  the antibody does not cross-react with non-methylated, mono-methylated, or di-methylated Lys27. In addition, the antibody does not cross-react with mono-methylated, di-methylated, or tri-methylated histone H3 at Lys4, Lys9, Lys36, or Histone H4 at Lys20 (https://www.cellsignal.com/products/primary-antibodies/tri-methyl-histone-h3-lys27-c36b11-rabbit-mab/9733).

H3K9ac CST 9671:  According to the manufacturer's page, the acetyl-Histone H3 (Lys9) Antibody detects endogenous levels of Histone H3 only when acetylated at lysine 9. It does not cross-react with phosphorylated histone H3 (https://www.cellsignal.com/product/productDetail.jsp?productId=9671).

Med1 Millipore 17-10530: The manufacturer validates this antibody using ChIP-qPCR (https://www.merckmillipore.com/DK/en/product/ChIPAb-MED1-Antibody-ChIP-Validated-Antibody-and-Primer-Set,MM_NF-17-10530?ReferrerURL=https%3A%2F%2Fwww.google.com%2F#anchor_keySpecTable).

# Eukaryotic cell lines

Policy information about cell lines

| Cell line source(s) | K562: source ATCC, cat# CCL 243<br>hTERT RPE-1: source ATCC (Cat# CRL-4000)<br>Mouse ESC were generously provided by the SyBOSS consortium partners (Francis Stewart, Austin Smith, Bill Skarnes). |
|---|---|
| Authentication | All human cell lines were authenticated by Eurofins Genomics using PCR-single-locus-technology. Identity of ESC was confirmed by the expression of cell-type-specific markers by RNA-seq. |
| Mycoplasma contamination | Cell lines were tested for mycoplasma contamination every 2-3 months, and were confirmed mycoplasma negative. |
| Commonly misidentified lines<br>(See ICLAC register) | None of the used cell lines are listed in the commonly misidentified lines. |

# ChIP-seq

## Data deposition

☒ Confirm that both raw and final processed data have been deposited in a public database such as GEO.

☒ Confirm that you have deposited or provided access to graph files (e.g. BED files) for the called peaks.

| Data access links<br>*May remain private before publication.* | GEO:GSE186349 |
|---|---|
| Files in database submission | ChIP-seq:<br>ESC_Ctrl_TC0_H2BK5ac.ab40886_CC8_1.fastq.gz<br>ESC_A485_10uM_TC15_H2BK5ac.ab40886_CC8_1.fastq.gz<br>ESC_Ctrl_TC0_H2BK5ac.CST12799_CC1_1.fastq.gz<br>ESC_Ctrl_TC0_H2BK5ac.CST12799_CC1_2.fastq.gz<br>ESC_Ctrl_TC0_H2BK5ac.CST12799_CC15_1.fastq.gz<br>ESC_Ctrl_TC0_H2BK5ac.CST12799_CC11_1.fastq.gz<br>ESC_A485_10uM_TC15_H2BK5ac.CST12799_CC1_1.fastq.gz<br>ESC_A485_10uM_TC15_H2BK5ac.CST12799_CC1_2.fastq.gz<br>ESC_Ctrl_TC0_H2BK12ac.ab40883_CC8_1.fastq.gz<br>ESC_Ctrl_TC0_H2BK12ac.ab40883_CC15_1.fastq.gz<br>ESC_A485_10uM_TC15_H2BK12ac.ab40883_CC8_1.fastq.gz<br>ESC_A485_10uM_TC15_H2BK12ac.ab40883_CC15_1.fastq.gz<br>ESC_Ctrl_TC0_H2BK16ac.ab177427_CC8_1.fastq.gz<br>ESC_Ctrl_TC0_H2BK16ac.ab177427_CC15_1.fastq.gz<br>ESC_A485_10uM_TC15_H2BK16ac.ab177427_CC8_1.fastq.gz<br>ESC_A485_10uM_TC15_H2BK16ac.ab177427_CC15_1.fastq.gz<br>ESC_Ctrl_TC0_H2BK20ac.ab177430_CC8_1.fastq.gz |

ESC_Ctrl_TC0_H2BK20ac.ab177430_CC15_1.fastq.gz
ESC_Ctrl_TC0_H2BK20ac.ab177430_CC19_1.fastq.gz
ESC_A485_10uM_TC15_H2BK20ac.ab177430_CC8_1.fastq.gz
ESC_A485_10uM_TC15_H2BK20ac.ab177430_CC15_1.fastq.gz
ESC_Ctrl_TC0_H2BK20ac.CST34156_CC15_1.fastq.gz
ESC_A485_10uM_TC15_H2BK20ac.CST34156_CC15_1.fastq.gz
ESC_Ctrl_TC0_Med1_dfix_Spike.HEK293_CC10_1.fastq.gz
ESC_Ctrl_TC0_Med1_dfix_Spike.HEK293_CC12_1.fastq.gz
ESC_Ctrl_TC0_H2BK20ac.CST34156_CC27_1.fastq.gz
ESC_NVP2_100nM_TC120_H2BK20ac.CST34156_CC27_1.fastq.gz
ESC_Ctrl_TC0_H2BK11ac.RM456_CC20_1.fastq.gz
ESC_Ctrl_TC0_H3K4me3_CC10_1.fastq.gz
ESC_Ctrl_TC0_H3K9ac_CC14_1.fastq.gz
ESC_Ctrl_TC0_H3K27me3_CC8_1.fastq.gz
K562_Ctrl_TC0_H2BK20ac.ab177430_CC16_1.fastq.gz
K562_Ctrl_TC0_H2BK20ac.CST34156_CC20_1.fastq.gz
K562_Ctrl_TC0_H3K27ac.ab4729_CC10_1.fastq.gz

**Genome browser session**
(e.g. UCSC)

NA

## Methodology

**Replicates**

Single ChIP-seq replicate was performed in the following samples: H2BK5ac(ab40866), H2BK11ac, H3K9ac, H3K4me3, H3K27me3 in ESC and to map H3K27ac in K562 cells on control condition. A-485 5 min treatment and H2BK20ac ChIP in ESC on NVP-2 120 min treatment condition. Other ChIP-seq experiments were in two or more biological replicates.

**Sequencing depth**

ChIP-seq:
ESC_Ctrl_TC0_H2BK5ac.ab40886_CC8_1.fastq.gz  total: 34589169 unique: 25018109 75bp single-end
ESC_A485_10uM_TC15_H2BK5ac.ab40886_CC8_1.fastq.gz  total: 34730315 unique: 25082513 75bp single-end
ESC_Ctrl_TC0_H2BK5ac.CST12799_CC1_1.fastq.gz total: 54917433 unique: 38503438 75bp single-end
ESC_Ctrl_TC0_H2BK5ac.CST12799_CC1_2.fastq.gz total: 56102173 unique: 39752879 75bp single-end
ESC_Ctrl_TC0_H2BK5ac.CST12799_CC15_1.fastq.gz  total: 36220680 unique: 25711848 75bp single-end
ESC_Ctrl_TC0_H2BK5ac.CST12799_CC11_1.fastq.gz  total: 31431198 unique: 20844959 75bp single-end
ESC_A485_10uM_TC15_H2BK5ac.CST12799_CC1_1.fastq.gz  total: 59354741 unique: 41082159 75bp single-end
ESC_A485_10uM_TC15_H2BK5ac.CST12799_CC1_2.fastq.gz  total: 56323325 unique: 39203285 75bp single-end
ESC_Ctrl_TC0_H2BK12ac.ab40883_CC8_1.fastq.gz  total: 34085582 unique: 25128054 75bp single-end
ESC_Ctrl_TC0_H2BK12ac.ab40883_CC15_1.fastq.gz  total: 31747589 unique: 23745357 75bp single-end
ESC_A485_10uM_TC15_H2BK12ac.ab40883_CC8_1.fastq.gz total: 31094097 unique: 22239318 75bp single-end
ESC_A485_10uM_TC15_H2BK12ac.ab40883_CC15_1.fastq.gz  total: 33282776 unique: 24517700 75bp single-end
ESC_Ctrl_TC0_H2BK16ac.ab177427_CC8_1.fastq.gz total: 33306788 unique: 24418077 75bp single-end
ESC_Ctrl_TC0_H2BK16ac.ab177427_CC15_1.fastq.gz total: 28793790 unique: 21965837 75bp single-end
ESC_A485_10uM_TC15_H2BK16ac.ab177427_CC8_1.fastq.gz total: 32313984 unique: 23562769 75bp single-end
ESC_A485_10uM_TC15_H2BK16ac.ab177427_CC15_1.fastq.gz  total: 33637842 unique: 25096858 75bp single-end
ESC_Ctrl_TC0_H2BK20ac.ab177430_CC8_1.fastq.gz  total: 33346076 unique: 24429920 75bp single-end
ESC_Ctrl_TC0_H2BK20ac.ab177430_CC15_1.fastq.gz  total: 31915477 unique: 24135271 75bp single-end
ESC_Ctrl_TC0_H2BK20ac.ab177430_CC19_1.fastq.gz  total: 36479905 unique: 24698633 75bp single-end
ESC_A485_10uM_TC15_H2BK20ac.ab177430_CC8_1.fastq.gz  total: 31152039 unique: 22319165 75bp single-end
ESC_A485_10uM_TC15_H2BK20ac.ab177430_CC15_1.fastq.gz  total: 28526891 unique: 21228677 75bp single-end
ESC_Ctrl_TC0_H2BK20ac.CST34156_CC15_1.fastq.gz total: 22737115 unique: 17146650 75bp single-end
ESC_A485_10uM_TC15_H2BK20ac.CST34156_CC15_1.fastq.gz  total: 30444009 unique: 22533354 75bp single-end
ESC_Ctrl_TC0_Med1_dfix_Spike.HEK293_CC10_1.fastq.gz total: 28787747 unique: 14300032 75bp single-end
ESC_Ctrl_TC0_Med1_dfix_Spike.HEK293_CC12_1.fastq.gz total: 34914638 unique: 10244526 75bp single-end
ESC_Ctrl_TC0_H2BK20ac.CST34156_CC27_1.fastq.gz total: 37810933 unique: 30406203 75bp single-end
ESC_NVP2_100nM_TC120_H2BK20ac.CST34156_CC27_1.fastq.gz total: 40411831 unique: 32516846 75bp single-end
ESC_Ctrl_TC0_H2BK11ac.RM456_CC20_1.fastq.gz total: 30338672 unique: 23964425 75bp single-end
ESC_Ctrl_TC0_H3K9ac_CC14_1.fastq.gz  total: 33695764 unique: 25289334 75bp single-end
ESC_Ctrl_TC0_H3K4me3_CC10_1.fastq.gz  total: 28251742 unique: 19797653 75bp single-end
ESC_Ctrl_TC0_H3K27me3_CC8_1.fastq.gz  total: 34075541 unique: 26440139 75bp single-end
K562_Ctrl_TC0_H2BK20ac.ab177430_CC16_1.fastq.gz total: 32428667 unique: 26093021 75bp single-end
K562_Ctrl_TC0_H2BK20ac.CST34156_CC20_1.fastq.gz total: 42894132 unique: 35626866 75bp single-end
K562_Ctrl_TC0_H3K27ac.ab4729_CC10_1.fastq.gz total: 27374349 unique: 21414657 75bp single-end

**Antibodies**

anti-H2BK5ac: abcam, ab40886, rabbit EP857Y
anti-H2BK5ac: Cell Signaling Technology, cat# CST12799, clone# rabbit D5H1S
anti-H2BK12ac: abcam cat# ab40883, clone# rabbit EP858Y
anti-H2BK16ac: abcam cat# ab177427, clone# rabbit EPR17598
anti-H2BK20ac: abcam cat# ab177430, clone# rabbit EPR859
anti-H2BK20ac: Cell Signaling Technology cat# 34156, clone# rabbit D7O9W
anti-H3K27ac: abcam cat# ab4729, clone# Rabbit polyclonal
anti-H3K9ac: Cell Signaling Technology, cat# 9671, clone# rabbit polyclonal
anti-H2BK11ac: RevMAb Bioscience, cat# 31-1348-00, clone# rabbit RM456

anti-H3K27me3: CST #9733, rabbit C36B11
anti-H3K4me3: Millipore 04-745, rabbit MC315
anti-Med1: Millipore 17-10530, clone# rabbit polyclonal

**Peak calling parameters**

Reads were mapped to the reference genome by using bwa aln with default parameters (BWA version 0.7.10). Multi-mapped reads, duplicated reads, or reads with more than three mismatches were removed by samtools. Reads mapped to the DAC Blacklisted Regions (https://www.encodeproject.org/annotations/ENCSR636HFF/) were omitted from the downstream analysis. Peak regions were called using LanceOtron(1.0.8) with default model (wide-and-deep_jan-2021) (doi: https://doi.org/10.1101/2021.01.25.428108 ). The peaks proximal within 2Kbp are merged using Bedtools, and poorly enriched peaks of maximum peak height < 8 reads mapped per million (rpm) were omitted.

**Data quality**

FastQC was used for quality check of sequencing reads.

**Software**

Sequencing analysis: Cutadapt(4.2), FastQC(0.11.5), Bedtools(2.23), HTseq(0.11.1), BWA(0.7.10), samtools(1.4), LanceOtron(1.0.8), R(4.1.1), PRROC (1.3), ChIPseeker (1.28.3), GenomicAlignments (1.8.4), GenomicRanges (1.44.0), stats (3.6.2), ggplot2(3.3.5), DESeq(1.32.0), ChIPseeker(l.28.3),  msa(1.24.0), deeptools(3.5.0), JUICER (1.6), IGV(2.10.3), ChromHMM(1.22), ABC model pipeline (https://github.com/broadinstitute/ABC-Enhancer-Gene-Prediction).  PRROC (1.3), ChIPseeker (1.28.3), GenomicAlignments (1.8.4), GenomicRanges (1.44.0).

