## [Peer Review File · Nature Genetics]

Peer Review Information

Manuscript Title: Acetylation of histone H2B marks active enhancers and predicts CBP/p300 target genes

Corresponding author name(s): Professor Chunaram Choudhary

Reviewer Comments & Decisions:

Decision Letter, Initial Version:

23rd Sep 2022

Dear Chuna,

Your Article entitled "A unique H2B acetylation signature marks active enhancers and predicts their target genes" has now been seen by 4 referees, whose comments are attached. I apologize for the slow review process.

While the reviewers find your work of potential interest, they have raised serious concerns which in our view are sufficiently important that they preclude publication of the work in Nature Genetics, at least in its present form.

Reviewer #1 seems generally positive about this paper but is not convinced that H2BNTac is a better mark for enhancers than H3K27ac. They want to know whether all the ChIP-seq datasets are truly comparable (i.e. prepared using the same pipeline). The reviewer finds the ABC section misleading; this needs to be carefully addressed.

Reviewer #2 is convinced that H2BNTac is a robust enhancer mark but raises a lot of concerns about

least in its present form.

Reviewer #1 seems generally positive about this paper but is not convinced that H2BNTac is a better mark for enhancers than H3K27ac. They want to know whether all the ChIP-seq datasets are truly comparable (i.e. prepared using the same pipeline). The reviewer finds the ABC section misleading; this needs to be carefully addressed.

Reviewer #2 is convinced that H2BNTac is a robust enhancer mark but raises a lot of concerns about the data interpretation, including the ABC modeling.

Reviewer #3 highlights that since H2BK20ac has already been shown to mark enhancers (ref. 9), the novelty is limited. They think that you should do further antibody testing and remove all data using non-specific antibodies.

Reviewer #4 is enthusiastic about this work overall and would support publication after a suitable revision.

Should further experimental data allow you to fully address these criticisms we would be willing to consider an appeal of our decision (unless, of course, something similar has by then been accepted at Nature Genetics or appeared elsewhere). This includes submission or publication of a portion of this work someplace else. We hope you understand that until we have read the revised manuscript in its entirety we cannot promise that it will be sent back for peer review.

If you are interested in attempting to revise this manuscript for submission to Nature Genetics in the future, please contact me to discuss a potential appeal. Otherwise, we hope that you find our referees' comments helpful when preparing your manuscript for resubmission elsewhere.

Sincerely,

Tiago

Tiago Faial, PhD
Chief Editor
Nature Genetics
<https://orcid.org/0000-0003-0864-1200>

Reviewers' Comments:

Reviewer #1:

Remarks to the Author:

This manuscript by the Choudhary lab explores N-terminal acetylation of histone H2B as a general marker for enhancers. The authors generate ChIP-seq data using a variety of antibodies recognizing different H2BNTac residues in mESCs and K562 cells. The authors argue that H2BNTac forms a distinct signature from H3K27ac and other chromatin marks that are in wide use. The H2BNTac signature is most prominent at distal intergenic sequence and a handful of promoters, suggesting that it may have

some specificity for specific targets. There is a lot that I like about this paper. The distribution of H2BNTac marks does appear to be distinct from K27ac (pending the resolution of several technical comments, below). I am also fairly convinced that H2BNTac marks genes that are more sensitive to CBP/p300 based on Fig. 7. Lastly, I admire the author's care in interpreting antibody specificity of two H2BK5ac antibodies in the text. However the premise that H2BNTac is a better marker for enhancers in general is not convincing to me: First, data comparing recovery of MPRA peaks using H2BNTac and H3K27ac in Fig. 5 and 6 does not make H2BNTac stand out as unequivocally better than H3K27ac; and Second, I have some major reservations about the analysis comparing promoter H2BNTac to ABC scores as noted in detail below. Broadly, my takeaway is that H2BNTac is an interesting mark that warrants further study, but evidence that it is indistinguishable from the function of an enhancer is not sufficient at this time.

Major comments:

* Throughout the manuscript the authors rely on comparisons between H3K27ac and H2BNTac ChIP-seq data. ChIP-seq data has a ton of assay and biological noise that can make two experimental datasets for the same mark quite discrepant from each other. Therefore the origin of H3K27ac datasets and its relationship to H2BNTac datasets needs to be spelled out very clearly: Are these datasets both prepared in the same person's hands at the same time from the same cell passage (i.e., what we would consider technical replicates if the ChIP-seq was performed on the same mark)? Different datasets generated by the same person in the Choudhary lab at different times (i.e., biological replicates)? Or different datasets generated by different labs at a different time (these are likely to have a higher degree of both biological and technical noise)? Reading the Methods section it is clear that at least some of the H3K27ac data from both mESCs and perhaps all of the data from K562 was scraped from existing datasets from other labs. Although it is great to reuse data, differences observed between these datasets need to be interpreted with extreme caution. Which dataset used in each comparison should be spelled out. Using a dataset from a different lab could result in clustering datasets across labs as observed in Fig. 1 b-c, overlaps between peaks, and other comparisons made by the authors.

* The authors primarily make use of background-subtracted signals in peak calls in their analysis. While there are substantial advantages of this approach, this strategy makes a non-linear transformation that could impact correlations between the marks. I would welcome the addition of analyses comparing raw counts, both in peaks and in genomic windows, in the extended data as a complement to the correlations. As Spearman's correlation is a rank order that is often best for genomic analyses, it does have more difficulty if large numbers of the observations are at or below background, and therefore have a random order - and I would also therefore encourage the authors to add log-transformed Pearson's correlations.

Along the same lines, the authors should spell out whether correlations presented in Fig. 1b are between the universe of all peaks for any of the marks (or all promoters in and outside of peak calls)? Or between peaks that overlap between each pair? This type of detail matters a great deal when interpreting peaks.

* I find the section comparing promoter H2BNTac with the ABC model to be misleading. The purpose of the ABC model is to predict whether a specific enhancer will regulate the expression from a specific target promoter. The setup in this paper appears to be slightly different: the authors treat with the CBP/p300 inhibitor, A-485, and attempt to classify which genes are sensitive to down-regulation. The

assumption made by the authors is that genes regulated by A-485 require enhancers for their up-regulation and all other genes do not. I am not at all convinced that this assumption is appropriate. Rather, as shown elsewhere in the paper, the authors are able to predict which genes are sensitive to A-485 inhibition using H2BNTac based on ChIP-seq signal in the promoter. Equating this to the sensitivity of the gene to any enhancer is not appropriate.

Reviewer #2:

Remarks to the Author:

In their manuscript, Narita et al report that histone H2B N-terminus multisite lysine acetylation (H2BNTac) marks active enhancers and promoters of enhancer-regulated genes. The correlation between H2BNTac and active enhancers as well as dependency of H2BNTac on p300/CBP is solid and interesting. From the presented data I am quite convinced that H2BNTac represents a novel active enhancer-associated mark. Unfortunately, beyond this observation, most of the other claims in the manuscript are poorly supported by the data. In addition, presentation of the results is obfuscated to the point that it is difficult to understand what has been done. Altogether, we cannot support publication of the manuscript in its current form.

Major points:

1. To make their point that H2BNTac is more closely correlated with active enhancers than H3K27ac (a histone modification mark that is most commonly used as an indicator of active cis-regulatory elements), authors compare the defined set of peaks associated with each mark against an orthogonal assay, such as MPRA. However, it is difficult to assess whether the comparison was fairly made. Some of the H2BNTac marks have fewer significant peaks than H3K27ac and therefore may perform better (e.g. in proportion of peaks with MPRA signal) but only because they represent more robustly active enhancers, with likely also higher H3K27ac (in fact, authors themselves point to that problem in relation to other data). To make fair comparisons, authors should compare equivalent numbers of peaks matched for H3K27ac enrichment level, with or without H2BNTac enrichment. Or at the very least, compare the equivalent numbers of top peaks for each mark.
2. On a related note, the authors' overview of the relationship between the various H2BNTac marks and H3K27ac (Fig 1a) is presented in a confusing and unintuitive manner. A much simpler and more easily interpretable visualization would be to use heatmaps of ChIP-seq signal (along with meta plots if desired) for each mark across the superset of H3K27ac + H2BNTac peaks, segregated by proximal vs distal and centered at DNase or ATAC peak summits.
3. H2BNTac at promoters seems to predict A-485-dependent gene transcription quite well, and the authors claim this represents the dependency of these genes on enhancers. However, the authors have not established whether A-485 in fact inhibits activity of all (or even most) enhancers, nor have they excluded a possibility that a subset of promoters is simply more dependent on p300/CBP and largely responsible for the observed expression changes. The observation that TSS H2BNTac signal outperforms a modified ABC score (Fig 8) in fact suggests that A-485 may selectively impact the transcription of genes that have certain promoter features rather than certain enhancer landscapes, and thus the claim that 'promoter H2BNTac signal predicts enhancer target genes' is grossly overblown. Interestingly, previous work on p300/CBP and H3K27ac also reports much better correlation of gene expression changes with acetylation changes at promoters rather than enhancers,

but with a different interpretation (quote from Martire et al., 2020: Transcriptional dysregulation is generally correlated with dysregulation of promoter acetylation upon depletion of p300 (but not CBP) and appears to be relatively independent of dysregulated enhancer acetylation; PMID: 32690000). Fig. 3a also shows that changes in H2BNTac upon A-485 are concentrated at promoters (while H3K27ac changes are minimal), raising the question of whether A-485 is truly inhibiting enhancer function. This is confusing given the authors' claims that H2BNTac is specific to enhancers rather than promoters.

4. To establish that H2BNTac outperforms H3K27ac in predicting enhancers regulating a given gene in the context of the ABC model, the authors should reanalyze data from the original ABC paper (Fulco et al, Nature Genetics 2019) and others that have specifically identified functional enhancer-promoter pairs (Gasperini et al, Cell 2019) by performing CRISPRi of individual enhancers and assessed resulting changes on gene expression. If the authors' claim is true, using H2BNTac instead of H3K27ac would improve the power of the ABC model in predicting specific enhancers regulating a given gene. Indeed, the authors already have H2BNTac ChIP-seq in K562 cells (the cell line in which the enhancer CRISPRi experiments were performed). Promoter acetylation should be excluded from the ABC score analysis if the comparison is to assess predictive value of these marks in enhancer activity.

Even if the authors believe that their heuristic of enhancers regulating the closest TSS with H2BNTac is the best strategy for identifying enhancer-promoter pairs, they should systematically evaluate how well this strategy performs in predicting the functionally validated enhancer-promoter pairs from Fulco et al and Gasperini et al.

Reviewer #3:

Remarks to the Author:

Takeo et al have systematically studied the genomic distributions of histone H2B N terminus multisite acetylation (H2BNTac) and establish H2BNTac as a potential signature for cell-type-specific active enhancers.

Their main findings include:

1. Using rabbit monoclonal antibodies recognizing specific acetylation sites on H2B, and through ChIP-seq experiment they find that H2BNTac is predominantly enriched at cell-type-specific active enhancers.
2. H2BNTac has a more pronounced enhancer-specificity than H3K27ac, which is also catalysed by CBP/P300 and widely used in defining enhancers but also occupies active promoters. Based on the observation that transcription inhibition could lead to increased H2BNTac at promoter and gene body regions, but not at enhancers, they conclude that transcription-coupled histone H2A-H2B exchange at promoter and gene body regions contributes to the relative depletion of H2BNTac at gene proximal regions but enrichment at distal regulatory regions.
3. Furthermore, they find H2BNTac by itself exhibits the highest specificity in decorating active enhancers and outperform other histone modifications like H3K27me3 and H3K4me1 in differentiating active enhancer from active promoters.
4. They suggest that incorporating the promoter H2BNTac information could bring in more accuracy when using chromatin confirmations to identify enhancer and target gene pairs, suggesting that the

promoter H2BNTac signal reflects the activity of enhancer-promoter interaction.

The major novelty in this manuscript is to improve the Chromatin-enrichment method by which enhancers are mapped. However, because H2BK20ac already has been shown to provide a distinctive enhancer signature (reference 9), the submitted work is rather incremental. Moreover, because the manuscript does not provide any novel biological or mechanistic insights into enhancer regulation, we are unable to recommend the manuscript for publication in Nature Genetics.

Despite we cannot recommend the manuscript for publication in Nature Genetics, we have provided a few specific comments below, which may be of help for the authors.

1. The authors start their analysis by 'quality controlling' the antibodies by ChIP-qPCR (Supplementary Table 1). The best performing antibody based on this assay is H2BK5ac (EP857Y), which the authors later choose to discard as being non-specific – mainly because it is an outlier in their comparative assays, and they find it can cross-react to H3K acetylation. Obviously, this raises the question regarding the assay presented in Supplementary Table 1 and how it can be used by the authors. Moreover, it is rather confusing to read all the comparisons in the manuscript, in which the authors use an antibody that they later show is not specific. Therefore, we suggest that the authors present their data by first performing robust assays for the specificity of the used antibodies. Here, the authors should also try to use peptide arrays derived from histones with a large number of acetylated peptides. Moreover, the authors should consider using modified nucleosomes to address specificity (see for instance PMID 30244833 for how to carefully characterize antibodies for ChIP).

2. In Figure 5, the authors attempted to validate the enhancer activity of the H2B acetylated enhancers by overlapping the regions enriched by the various H2B lysine acetylated antibodies with what has been mapped as active enhancers. The degree of overlap was between 31-53%, suggesting a false negative rate of 47-69%. Interestingly, the best H2B acetylation antibodies performed better than H3K27ac and e-RNA based methods, although H2BK20ac used throughout the manuscript (and the accompanying) only maps 45% of active enhancers. The authors should discuss the implication of these results for the use of H2BK20ac to map active enhancers.

3. Using a combination of H2BK20ac and/or H3K27ac the authors show that 88% of all distal ATAC+MPRA+ regions (active enhancers) overlap with these modifications (Figure 6). Based on this result it would be logical to suggest that it would be more appropriate to use H2BK20ac and H3K27ac in combination to map active enhancers, however, this is not the take home message of the manuscript. Again, it would be useful if the authors discuss this in the manuscript.

Reviewer #4:

Remarks to the Author:

Summary

The authors show the remarkable importance of H2B acetylation in discriminating between levels of gene activity in a way that outperforms standard H3K27ac approaches. The work has nice functional data using inhibitors of transcription/acetylation. I suspect this work will greatly increase the use of H2B acetylation among epigenetic researchers in many fields, and will enhance the precision of future research directions for those using histone marks to delineate functional roles of chromatin machinery. Overall comments

This paper represents a major advance in understanding the modifications of histone and their relationship to enhancer activity and gene control. The work is of high interest to the field, and is technically robust.

Major points

1. Only minor criticisms were noted. Overall, I recommend reducing the number of figures to reduce redundancy, adding additional types of figure panels to increase the diversity within each figure, and I also recommend adding one or more cartoon representations of the main finding.

Novelty/notability claims

2. The paper's novelty is strong. A literature survey indicated that, indeed, the genomic locations of H2B acetylation have not been (to my surprise!) elsewhere shown.

Minor points

3. Minor grammatical errors have been noted in an attached PDF of the manuscript via comments/notes.

4. The authors discuss the role of H2B acetylation in relationship to chromHMM defined chromatin. It would be really nice to see the "emission" data and defined chromatin states with and without H2B acetylation. Here is an example:

5. There is a glaring lack of genome browser tracks that are need in the main figures to reveal the quality of the ChIP-seq data, and to assist in giving readers an intuition of the patterns being claimed through boxplots and correlation matrices. To fix this, the authors should promote Extended Figure 8b to the main figure set, or make a new genome browser figure that shows the differences in H3 acetylation and H2B acetylation at diverse regulatory elements. Another example is in the CDK inhibition data – the increase of H2B acetylation after inhibition of transcription is beautifully shown in Ext Data Figure 10C – perhaps 1 of these gene can be moved to the main figure 3, and some of the intellectually redundant violin plots can be moved to supplementary.

6. The role of H3K v H2B acetylation in predicting response to A485 is striking. The authors should add a metagene/average plot of the histone makes in Figure 7b, centered around the TSS of genes that are strongly down regulated, and compared to the same data plotting at the TSS of genes that are non-responsive.

Figure comments

7. Figure 5 and Figure 6 were swapped, causing some confusion for this reviewer.

8. Figure 6a should have H3K27ac as an additional row of plots with all the same measurements calculated.

Author Rebuttal to Initial comments

A point-by-point response to Reviewers' comments

We thank the reviewers for evaluating our work, and for providing constructive feedback, which we believe has helped us in improving the manuscript. Below, we provide a point-by-point response to the reviewers' comments.

Reviewer #1:

Remarks to the Author:

This manuscript by the Choudhary lab explores N-terminal acetylation of histone H2B as a general marker for enhancers. The authors generate ChIP-seq data using a variety of antibodies recognizing different H2BNTac residues in mESCs and K562 cells. The authors argue that H2BNTac forms a distinct signature from H3K27ac and other chromatin marks that are in wide use. The H2BNTac signature is most prominent at distal intergenic sequence and a handful of promoters, suggesting that it may have some specificity for specific targets. There is a lot that I like about this paper. The distribution of H2BNTac marks does appear to be distinct from K27ac (pending the resolution of several technical comments, below). I am also fairly convinced that H2BNTac marks genes that are more sensitive to CBP/p300 based on Fig. 7. Lastly, I admire the author's care in interpreting antibody specificity of two H2BK5ac antibodies in the text. However the premise that H2BNTac is a better marker for enhancers in general is not convincing to me: First, data comparing recovery of MPRA peaks using H2BNTac and H3K27ac in Fig. 5 and 6 does not make H2BNTac stand out as unequivocally better than H3K27ac; and Second, I have some major reservations about the analysis comparing promoter H2BNTac to ABC scores as noted in detail below. Broadly, my takeaway is that H2BNTac is an interesting mark that warrants further study, but evidence that it is indistinguishable from the function of an enhancer is not sufficient at this time.

We thank the reviewer for his/her efforts in evaluating our work and offering insightful comments. We are encouraged to read his/her feedback.

While noting the strengths of our work, the Reviewer commented that *"recovery of MPRA peaks using H2BNTac and H3K27ac in Fig. 5 and 6 does not make H2BNTac stand out as unequivocally better than H3K27ac"*.

We wish to clarify that the data presented in these figures are not the basis of our claim that H2BNTac is a better enhancer marker than H3K27ac. What these data show is that H2BNTac and H3K27ac are similarly sensitive in detecting active enhancers (defined by STARR-seq or eRNA expression). Thus, we agree with the Reviewer's observation that, from the data presented in Figs. 5 and 6, H2BNTac does not stand out as a much better marker of enhancers than H3K27ac.

We consider that sensitivity and specificity are two distinct attributes of an enhancer marker. A marker can be more sensitive but less specific or vice versa. For example, DHS/ATAC-seq is perhaps even more sensitive than H3K27ac in marking enhancer regions, but H3K27ac has greater specificity in marking active enhancers than DHS. Thus, if judged based on the sensitivity to detecting enhancers, H3K27ac is not a better marker of enhancers than DHS. However, if judged based on specificity for active enhancers, then H3K27ac is a better marker than DHS. Our results show that H2BNTac has even greater specificity than H3K27ac for active enhancers, and therefore, we suggest that it is a better marker of enhancers. The claim, that H2BNTac is a better marker of active enhancers, CBP/p300 activity, and

CBP/p300-dependent gene regulation than H3K27ac, is supported by the data presented in Fig. 1b, Fig. 2, Fig. 4, Fig. 7 and 8, and Extended Data Fig. 6.

To appreciate the importance of specificity, it is worth comparing the abundance of H3K27ac and other histone acetylation marks in enhancers. To a greater or lesser extent, almost all the widely studied histone acetylation marks can be detected at highly active enhancers and promoters. However, H3K27ac is considered a more specific marker of enhancers because of its relatively higher abundance in enhancers than in other regions. However, as shown in Fig. 2e, the relative level of H3K27ac/H3K9ac is only modestly higher in enhancers than promoters. In the same analyses, if we compare the relative level of H2BNTac/H3K9ac, or H2BNTac/H3K27ac, the relative level of H2BNTac/H3K27ac in enhancers is much higher than in promoters (Fig. 2e). Therefore, H2BNTac affords much greater distinction than H3K27ac in discriminating these regulatory elements. Furthermore, the correlation between H3K27ac and H2BNTac show striking differences in correlation with H3K4me3 in promoter regions (Fig. 1b). Unlike H3K27ac, H2BNTac in promoter regions shows a positive correlation with H3K4me1 (Fig. 1b), a well-known marker of enhancers. These data support our conclusion that H2BNTac is a superior marker for defining active enhancers and discriminating them from constitutively active promoters of housekeeping genes.

H3K27ac is considered as the best marker of CBP/p300 activity, which is required for enhancer function. Thus, H3K27ac has been used as a proxy for enhancer activity/strength (for example, in the ABC model). CBP/p300 catalyzes the enhancer associated H3K27ac, but a large portion of H3K27ac in promoter regions is not catalyzed by CBP/p300. Thus, H3K27ac is not a very good marker for predicting locus-specific CBP/p300 activity. Our analyses show that H2BNTac more accurately reflects CBP/p300 activity, especially in promoter-proximal regions. This makes H2BNTac an attractive marker of CBP/p300 and offers a possible explanation for why H2BNTac intensity is a better predictor of CBP/p300-dependent gene activation. Because of the above reasons, we believe that H2BNTac is a better marker of active enhancers and CBP/p300 activity than H3K27ac.

Major comments:

* Throughout the manuscript the authors rely on comparisons between H3K27ac and H2BNTac ChIP-seq data. ChIP-seq data has a ton of assay and biological noise that can make two experimental datasets for the same mark quite discrepant from each other. Therefore the origin of H3K27ac datasets and its relationship to H2BNTac datasets needs to be spelled out very clearly: Are these datasets both prepared in the same person's hands at the same time from the same cell passage (i.e., what we would consider technical replicates if the ChIP-seq was performed on the same mark)? Different datasets generated by the same person in the Choudhary lab at different times (i.e., biological replicates)? Or different

datasets generated by different labs at a different time (these are likely to have a higher degree of both biological and technical noise)? Reading the Methods section it is clear that at least some of the H3K27ac data from both mESCs and perhaps all of the data from K562 was scraped from existing datasets from other labs. Although it is great to reuse data, differences observed between these datasets need to be interpreted with extreme caution. Which dataset used in each comparison should be spelled out. Using a dataset from a different lab could result in clustering datasets across labs as observed in Fig. 1 b-c, overlaps between peaks, and other comparisons made by the authors.

We fully agree with the reviewer's point and apologize for the lack of these details in the original manuscript. As the reviewer points out, there are many datasets available for H3K27ac, H3K9ac, H3K27me3, H3K4me3, H3K4me1, and MED1 in mouse ESC. Nonetheless, to avoid the confounding issue raised by the reviewer, we used datasets that were generated in-house, by the same person, using the same protocol. Because the differences in H2BNTac and H3K27ac occupancy in the promoter regions are very striking, some colleagues in the field suggested to us that, to rule out the possibility that the observed differences are not due to H3K27ac being an outlier in our dataset, we should include previously published H3K27ac ChIP-seq data, ideally generated using different antibodies, in our comparison. Therefore, we included two previously published H3K27ac ChIP-seq datasets for comparison, but this is in addition to the H3K27ac data generated in our lab. In short, the different clustering between H3K27ac and H2BNTac, and between these marks and other marks (such as H3K4me3, MED1) is not due to the production of data using a different pipeline.

* The authors primarily make use of background-subtracted signals in peak calls in their analysis. While there are substantial advantages of this approach, this strategy makes a non-linear transformation that could impact correlations between the marks. I would welcome the addition of analyses comparing raw counts, both in peaks and in genomic windows, in the extended data as a complement to the correlations. As Spearman's correlation is a rank order that is often best for genomic analyses, it does have more difficulty if large numbers of the observations are at or below background, and therefore have a random order - and I would also therefore encourage the authors to add log-transformed Pearson's correlations.

Along the same lines, the authors should spell out whether correlations presented in Fig. 1b are between the universe of all peaks for any of the marks (or all promoters in and outside of peak calls)? Or between peaks that overlap between each pair? This type of detail matters a great deal when interpreting peaks.

We thank the reviewer for this helpful suggestion. We will perform the suggested analyses and include them in the revised manuscript.

* I find the section comparing promoter H2BNTac with the ABC model to be misleading. The purpose of the ABC model is to predict whether a specific enhancer will regulate the expression from a specific target promoter. The setup in this paper appears to be slightly different: the authors treat with the CBP/p300 inhibitor, A-485, and attempt to classify which genes are sensitive to down-regulation. The assumption made by the authors is that genes regulated by A-485 require enhancers for their up-regulation and all other genes do not. I am not at all convinced that this assumption is appropriate. Rather, as shown elsewhere in the paper, the authors are able to predict which genes are sensitive to A-485 inhibition using H2BNTac based on ChIP-seq signal in the promoter. Equating this to the sensitivity of the gene to any enhancer is not appropriate.

We fully agree with the Reviewers that premise of the ABC model is to predict whether a specific enhancer will regulate expression of a specific gene. This is useful for predicting enhancer targets, but until now, the field has lacked the tools for predicting genes regulated by CBP/p300 activity. CBP/p300-catalyzed H3K27ac is widely used for defining candidate enhancers, most functionally characterized native enhancers require CBP/p300 activity for target gene activation, and the ABC model also uses H3K27ac for predicting enhancer targets. Thus, we hypothesized that the sum of ABC scores of all enhancers should predict the cumulative impact of all enhancers on a gene, and hence, predict a gene's overall dependence on enhancers. Therefore, we tested if the ABC model can be adapted for predicting CBP/p300 regulated genes.

We acknowledge that we cannot rule out an enhancer-independent role of CBP/p300 in gene activation. Thus, we agree with the Reviewers that we should be cautious in our interpretations and not equate the functions of CBP/p300 and enhancers in gene activation. Nonetheless, we believe that it is useful to find out if the model shows similar or different accuracy for predicting genes impacted by enhancer perturbation (by CRISPRi) or A-485 treatment. If the model shows similar accuracy as for predicting genes regulated by enhancer perturbation and by A-485 treatment, it will at least provide an indication that the genes affected by A-485 are regulated by enhancers.

Based on the Reviewer's feedback, we will revise the text to acknowledge the limitations pointed out by the reviewers, we will include additional analyses to present the fairest possible comparison.

Reviewer #2:

Remarks to the Author:

In their manuscript, Narita et al report that histone H2B N-terminus multisite lysine acetylation (H2BNTac) marks active enhancers and promoters of enhancer-regulated genes. The correlation between H2BNTac and active enhancers as well as dependency of H2BNTac on p300/CBP is solid and interesting. From the presented data I am quite convinced that H2BNTac represents a novel active enhancer-associated mark. Unfortunately, beyond this observation, most of the other claims in the manuscript are poorly supported by the data. In addition, presentation of the results is obfuscated to

the point that it is difficult to understand what has been done. Altogether, we cannot support publication of the manuscript in its current form.

We thank the reviewer for carefully evaluating the work and offering very helpful comments. Below, we offer a point-by-point response to his/her comments.

Major points:

1. To make their point that H2BNTac is more closely correlated with active enhancers than H3K27ac (a histone modification mark that is most commonly used as an indicator of active cis-regulatory elements), authors compare the defined set of peaks associated with each mark against an orthogonal assay, such as MPRA. However, it is difficult to assess whether the comparison was fairly made. Some of the H2BNTac marks have fewer significant peaks than H3K27ac and therefore may perform better (e.g. in proportion of peaks with MPRA signal) but only because they represent more robustly active enhancers, with likely also higher H3K27ac (in fact, authors themselves point to that problem in relation to other data). To make fair comparisons, authors should compare equivalent numbers of peaks matched for H3K27ac enrichment level, with or without H2BNTac enrichment. Or at the very least, compare the equivalent numbers of top peaks for each mark.

We fully agree with the reviewer's concern and thank him/her for the excellent suggestion. We have included the suggested analyses in the revised manuscript. These results confirm our original conclusions.

2. On a related note, the authors' overview of the relationship between the various H2BNTac marks and H3K27ac (Fig 1a) is presented in a confusing and unintuitive manner. A much simpler and more easily interpretable visualization would be to use heatmaps of ChIP-seq signal (along with meta plots if desired) for each mark across the superset of H3K27ac + H2BNTac peaks, segregated by proximal vs distal and centered at DNase or ATAC peak summits.

We apologize for the confusion and thank the reviewer for the suggestion. We now additionally present the ChIP-seq data in a heatmap format as suggested by the reviewer.

3. H2BNTac at promoters seems to predict A-485-dependent gene transcription quite well, and the authors claim this represents the dependency of these genes on enhancers. However, the authors have not established whether A-485 in fact inhibits activity of all (or even most) enhancers, nor have they excluded a possibility that a subset of promoters is simply more dependent on p300/CBP and largely responsible for the observed expression changes. The observation that TSS H2BNTac signal outperforms a modified ABC score (Fig 8) in fact suggests that A-485 may selectively impact the transcription of genes that have certain promoter features rather than certain enhancer landscapes, and thus the claim that

'promoter H2BNTac signal predicts enhancer target genes' is grossly overblown. Interestingly, previous work on p300/CBP and H3K27ac also reports much better correlation of gene expression changes with acetylation changes at promoters rather than enhancers, but with a different interpretation (quote from Martire et al., 2020: Transcriptional dysregulation is generally correlated with dysregulation of promoter acetylation upon depletion of p300 (but not CBP) and appears to be relatively independent of dysregulated enhancer acetylation; PMID: 32690000). Fig. 3a also shows that changes in H2BNTac upon A-485 are concentrated at promoters (while H3K27ac changes are minimal), raising the question of whether A-485 is truly inhibiting enhancer function. This is confusing given the authors' claims that H2BNTac is specific to enhancers rather than promoters.

This point encompasses several questions.

Does A-485 inhibit the activity of all (or most) enhancers?

We will present analyses that show that CBP/p300 indeed inhibits acetylation, and thus likely, the activity of all distal enhancers.

Does the H2BNTac signal in promoters reflect the activity of CBP/p300 bound to enhancers, or if it reflects the enhancer-independent CBP/p300 activity at specific groups of promoters?

Consistent with the literature, our data show that CBP/p300 activity catalyzes acetylation in enhancer regions. We will present analyses that suggest that promoter H2BNTac reflects activity of proximal enhancers, rather than independent activity of CBP/p300 activity on promoters. CBP/p300 is a highly proficient enzyme, and our results imply that promoter acetylation reflects spread of acetylation from proximal enhancers. Indeed, it is already known that CBP/p300-catalyzed histone acetylation can easily spread several kilobases away from the enhancer centers, and in some instances acetylation spreads to 10s of kilobases.

It is true that CBP/p300-dependent gene expression shows better correlation with change in promoter acetylation, but we sincerely believe that this is not because CBP/p300 act independently of CBP/p300. CBP/p300 inhibition reduces acetylation at virtually all genes that are known to be activated by distal enhancers (such as Sox2, Myc, Klf4 etc.), but this does not mean that CBP/p300 activate these genes by directly activating promoters, independently of distal enhancers.

While our results strongly suggest that promoter H2BNTac signal reflects CBP/p300 activity at nearby enhancers, we will revise the text to acknowledge that in some instances CBP/p300 may acetylate promoters and we cannot rule out this possibility. Technically, demonstrating that CBP/p300 acetylate promoters, independently of enhancers is extremely difficult, if not impossible. This is because there are many enhancers that occur almost overlappingly with promoters, making it difficult to disentangle CBP/p300 function in promoter and enhancer.

Regarding the comment “Fig. 3a also shows that changes in H2BNTac upon A-485 are concentrated at promoters (while H3K27ac changes are minimal), raising the question of whether A-485 is truly inhibiting enhancer function. This is confusing given the authors’ claims that H2BNTac is specific to enhancers rather than promoters.”

The question here is- why does A-485 causes more rapid reduction in H2BNTac in promoters than in enhancers? Could this be an indication that CBP/p300 act on promoters independently of enhancers?

Firstly, we wish to clarify that the presented analyses were done at relatively early time (15 min) point after A-485 treatment. While acetylation is not fully abrogated at this early time point, H2BK20ac is globally reduced (log2 median ratio is significantly lower than 0). The vast majority of acetylated H2B is present in enhancer regions, and mass spectrometry analyses show that A-485 treatment reduced H2B acetylation by 95-98%. Thus, we believe that CBP/p300 acetylates virtually all candidate enhancers.

As the Reviewer notes, H2BNTac is indeed more strongly reduced at promoters and actively transcribed regions, but this should not be interpreted that A-485 more prominently inhibits promoter-associated CBP/p300 than enhancer-bound CBP/p300. The reason for more rapid reduction in H2BNTac is the transcription-coupled rapid exchange of H2A-H2B dimers.

4. To establish that H2BNTac outperforms H3K27ac in predicting enhancers regulating a given gene in the context of the ABC model, the authors should reanalyze data from the original ABC paper (Fulco et al, Nature Genetics 2019) and others that have specifically identified functional enhancer-promoter pairs (Gasperini et al, Cell 2019) by performing CRISPRi of individual enhancers and assessed resulting changes on gene expression. If the authors’ claim is true, using H2BNTac instead of H3K27ac would improve the power of the ABC model in predicting specific enhancers regulating a given gene. Indeed, the authors already have H2BNTac CHIP-seq in K562 cells (the cell line in which the enhancer CRISPRi experiments were performed). Promoter acetylation should be excluded from the ABC score analysis if the comparison is to assess predictive value of these marks in enhancer activity. Even if the authors believe that their heuristic of enhancers regulating the closest TSS with H2BNTac is the best strategy for identifying enhancer-promoter pairs, they should systematically evaluate how well this strategy performs in predicting the functionally validated enhancer-promoter pairs from Fulco et al and Gasperini et al.

After reading the Reviewer’s comment (and comment 4 from Reviewer #2), we recognize that the way we compared the ABC model in our analyses was confusing to the extent that it appeared misleading. We apologize for not having presented these data in a better manner. We assure the Reviewers that this was not our intention. We will thoroughly revise the text to better clarify this point. We will also perform the suggested analyses.

Of note, even though we cannot equate gene regulation by CBP/p300 inhibition and enhancer perturbation, it is worth noting that the ABC model predicts A-485 regulated genes a good accuracy. Substituting H3K27ac with H2BNTac marginally improves the ABC model's predictive power for separating A-485 regulated genes from not regulated genes. This is not surprising because H3K27ac and H2BNTac are similarly abundant in the distal regions. The main difference in H3K27ac and H2BNTac is in the promoter proximal regions. Indeed, the predictive power of the ABC model decreases if we include promoter H3K27ac signal. In contrast, if we substitute H3K27ac by H2BNTac, inclusion of promoter H2BNTac signal improves the ABC model's prediction.

The question is whether promoter H2BNTac should be included for predicting enhancer targets or not? The ABC model removes H3K27ac and DHS intensity in promoter proximal regions (\pm 500bp from TSS). There is no evidence to show that enhancers do not localize within 500bp of TSS. Indeed, excluding promoter-proximal H3K27ac is counterintuitive because proximally located enhancers have the greatest impact on gene regulation and activity of enhancer decreases non-linearly from with their distance from the TSS. We believe that including acetylation signal from promoter regions is essential for accurate prediction of enhancer influence and will elaborate on the reasons why this is the case. Others have also started to recognize this, and the recent ENCODE (ENCODE Consortium, Nature 2020) classification include enhancers that localize much closer to TSS and only define promoter as \pm 200bp from TSS.

Reviewer #3:

Remarks to the Author:

Takeo et al have systematically studied the genomic distributions of histone H2B N terminus multisite acetylation (H2BNTac) and establish H2BNTac as a potential signature for cell-type-specific active enhancers.

Their main findings include:

1. Using rabbit monoclonal antibodies recognizing specific acetylation sites on H2B, and through ChIP-seq experiment they find that H2BNTac is predominantly enriched at cell-type-specific active enhancers.
2. H2BNTac has a more pronounced enhancer-specificity than H3K27ac, which is also catalysed by CBP/P300 and widely used in defining enhancers but also occupies active promoters. Based on the observation that transcription inhibition could lead to increased H2BNTac at promoter and gene body regions, but not at enhancers, they conclude that transcription-coupled histone H2A-H2B exchange at promoter and gene body regions contributes to the relative depletion of H2BNTac at gene proximal regions but enrichment at distal regulatory regions.
3. Furthermore, they find H2BNTac by itself exhibits the highest specificity in decorating active enhancers and outperform other histone modifications like H3K27me3 and H3K4me1 in differentiating active

enhancer from active promoters.

4. They suggest that incorporating the promoter H2BNTac information could bring in more accuracy when using chromatin confirmations to identify enhancer and target gene pairs, suggesting that the promoter H2BNTac signal reflects the activity of enhancer-promoter interaction.

The major novelty in this manuscript is to improve the Chromatin-enrichment method by which enhancers are mapped. However, because H2BK20ac already has been shown to provide a distinctive enhancer signature (reference 9), the submitted work is rather incremental. Moreover, because the manuscript does not provide any novel biological or mechanistic insights into enhancer regulation, we are unable to recommend the manuscript for publication in Nature Genetics.

We thank the reviewer for evaluating our manuscript.

Firstly, we wish to clarify a misunderstanding. The reviewer mentions that "*The major novelty in this manuscript is to improve the chromatin-enrichment method*". We wish to clarify that we used standard chromatin enrichment and sequencing methods for all our analyses, and we claim no improvements in the methods used for chromatin enrichment.

Secondly, we request the reviewer to not equate H2BNTac with H2BK20ac (H2BNTac \neq H2BK20ac).

We appropriately cite the prior work on H2BK20ac (ref. 9) and discuss the discrepancies in the available literature on H2BK20ac and other H2BNTac marks. It is worth noting that in the past >6 years, since the H2BK20ac ref. 9 was published, no new ChIP-seq data has been deposited in the GEO repository. This is also noted by Reviewer #4. The main reason for the historical lack of focus on H2B acetylation is that there are many discrepancies in the literature (See Supplemental Note 1), and most crucially, there has been a lack of mechanistic details that can explain the differences in H2B acetylation and H3K27ac.

We feel that major findings of our work may have escaped the reviewer's attention. The major findings of our work are that it provides the first unified picture of the H2BNTac in mammalian cells and reveals that all of the H2BNTac sites tested here indistinguishably mark the same genomic regions. Thus, unlike site-specific regulation and non-identical genomic occupancy of acetylation in the histones H3 and H4, the sites H2B N-terminus are similarly regulated and occupy the same genomic regions. Contrary to prior claims, we found no evidence of a specific group of enhancers that lack H3K27ac and are marked by H2BK20ac. Importantly, our work goes well beyond showing that H2BNTac is an enhancer marker. We reveal the mechanistic basis of differences in H2BNTac and H3K27ac, show that a vast majority of active enhancers (defined by MPRA and eRNA) are marked by CBP/p300-catalyzed marks, and H2BNTac is a superior predictor of CBP/p300-regulated genes than H3K27ac. We sincerely hope that the reviewer will

recognize the strengths of these findings, and the usefulness of H2BNTac in better defining enhancers and investigating their role in gene regulation.

Despite we cannot recommend the manuscript for publication in Nature Genetics, we have provided a few specific comments below, which may be of help for the authors.

1. The authors start their analysis by ‘quality controlling’ the antibodies by ChIP-qPCR (Supplementary Table 1). The best performing antibody based on this assay is H2BK5ac (EP857Y), which the authors later choose to discard as being non-specific – mainly because it is an outlier in their comparative assays, and they find it can cross-react to H3K acetylation. Obviously, this raises the question regarding the assay presented in Supplementary Table 1 and how it can be used by the authors. Moreover, it is rather confusing to read all the comparisons in the manuscript, in which the authors use an antibody that they later show is not specific. Therefore, we suggest that the authors present their data by first performing robust assays for the specificity of the used antibodies. Here, the authors should also try to use peptide arrays derived from histones with a large number of acetylated peptides. Moreover, the authors should consider using modified nucleosomes to address specificity (see for instance PMID 30244833 for how to carefully characterize antibodies for ChIP).

In general, we share the reviewers’ concern about antibody specificity. We wish to clarify that the specificity of the antibodies used here has already been tested for a set of acetylated histone peptides, as requested by the Reviewer (these validation data will be provided with the rebuttal). The antibodies have additionally been tested the manufacturers in immunofluorescence and/or immunoblotting with and without HDAC inhibitors. These assays have been used by other colleagues in the field for characterizing new histone modification ChIP-seq antibodies for and the results have been published in well-known journals ((H3K122ac; *Nature Genetics* 2016, PMID: 27089178), (H3K79suc; *Nature* 2017, PMID: 29211711), (H3K18la; *Nature* 2019, PMID: 31645732), (H3K4me3Q5ser; *Nature* 2019, PMID: 30867594).

Testing antibodies using site-specifically acetylated nucleosomes is a good idea, but comprehensive testing will require the production of hundreds of site-specifically modified recombinant nucleosomes, harboring PTMs in different combinations. Histone acetylation sites are widely mapped using ChIP-seq, but we are not aware of any study, including the ENCODE project, that has comprehensively validated specificity of histone acetylation antibodies using a complete set of recombinant nucleosomes bearing all acetylation sites. The use of modified nucleosomes can provide additional support for specificity, but arguably, even the use of recombinant nucleosomes will not be sufficient to completely rule out the possibility that antibodies may cross-react with histones bearing acetylation with another PTM or recognize cross-linked chromatin differently than purified nucleosomes. Also, there may be hundreds, if not thousands, of acetylation sites occurring in chromatin-associated proteins. In native chromatin ChIP,

it would be virtually impossible to rule out that the antibodies do not enrich a non-histone acetylated target in cross-linked chromatin.

We would like to clarify that what we mean by ‘quality controlling’ is that we tested whether the antibodies are suitable for doing CHIP or not. This was done because several of the antibodies used in our work have never been used previously for CHIP-seq. Antibodies that gave at least 5-fold enrichment over the background signal were used for CHIP-seq.

The Reviewer mentions that including cross-reactive antibodies caused confusion and “*The best performing antibody based on this assay is H2BK5ac (EP857Y), which the authors later choose to discard as being non-specific...*” We would like to clarify that did not discard data generated from any antibody. We simply excluded the H2BK5ac (EP857Y) CHIP-seq data from defining the H2BNTac signature because of its cross-reactivity with H3K27ac. It is unclear to us why this should cause concern for the data presented in Supplemental Table 1.

We wish to mention that we did not include cross-reactive antibodies in our comparisons by purpose. We had no prior knowledge about the antibody specificity, and the issue was only realized after noticing that the CHIP-seq profile of H2BK5ac (EP857Y) was an outlier. Alerted by this, we checked the amino acid similarity of H2BNT with other histones, which helped to uncover and explain the basis of cross-reactivities. There are two reasons why we did not entirely remove the data generated from the cross-reactivity antibody. (1) We felt that readers should be informed about it because most of previously published CHIP-seq data were generated these apparently cross-reactive antibodies. (2) Selectively excluding the data of cross-reactive antibodies would have made things appear clean, but we felt that, for transparency reasons, it is inappropriate to selectively remove data. To avoid any confusion, we explicitly state the antibodies that were excluded from defining the H2B acetylation signature.

As mentioned in the Discussion section, beyond their expected specificity in the standard antibody characterization assays, the following observations further corroborate the on-target specificity of the used H2BNTac antibodies, even if indirectly.

(1) Many histone acetylation marks have been profiled using CHIP-seq. To our knowledge, the observed genomic occupancy profile of H2BNTac sites is distinct from the known profiles of other histone acetylation marks. This indicates that the used H2BNTac antibodies are unlikely cross-react with any of the well-known histone acetylation sites.

(2) A-485-dependent H2BNTac downregulation in our CHIP-seq is consistent with mass spectrometry-based analyses of H2BNTac sites. The only other histone sites that show similarly strong, global downregulation after CBP/p300 inhibition are a subset of sites present in histone H3 N-terminus, whose amino acid sequences and genome occupancy profiles are dissimilar from H2BNTac.

(3) Importantly, all H2BNTac antibodies consistently show differences in the relative enrichment at actively transcribed regions and distal enhancers. Transcription inhibition preferentially increased H2BNTac in actively transcribed genes, consistent with the transcription-induced exchange of H2A-H2B, but not of H3-H4. This strongly supports that antibodies must recognize acetylated residues within histones H2A or H2B, not in H3 or H4. Because there is no sequence similarity between acetylation sites in H2A and H2B, the antibodies likely recognize acetylation sites within H2B.

(4) Because histone termini are unstructured, antibody specificity is likely primarily dictated by linear sequences, not 3D conformation. Cross-reactivity can arise if amino acids flanking modified residues are identical. Within the H2BNT, amino acids flanking lysine are highly dissimilar, making it unlikely that all monoclonal antibodies cross-react within H2BNT. Even if there is some cross-reactivity within H2BNTac sites, it will still reflect the H2BNTac signature.

(5) Guided by the sequence similarity, and the unique H2BNTac profile, we found one of the H2BK5ac antibodies as an outlier, explained its cross-reactivity with H3K27ac, and excluded it from defining the H2BNTac signature.

(6) CREs marked with H2BNTac bear all the canonical features of active enhancers, including DHS, H3K4me1, H3K27ac, and MED1 binding.

(7) H2BNTac marked regions show a high validation rate in orthogonal enhancer activity assays, supporting that this marks candidate active enhancers.

In brief, it is exceedingly difficult, if not impossible, to completely rule out off-target reactivities of antibodies. The question here is whether the unique ChIP-seq signature observed in our analyses is a consequence of a lack of antibody specificity and if this impacts our conclusions. The issue of antibody specificity can be a much more serious if the conclusions of a study rely on the use of a single antibody, especially if it is polyclonal. The idea that six different monoclonal antibodies, raised against 5 different H2BNTac sites, manufactured by three different companies, displaying expected on-target specificities in standard antibody characterization assays, produce an almost identical ChIP-seq signature by cross-reacting with an unknown mark appears highly unlikely to us. We sincerely believe that the distinct H2BNTac signature established here is not an artifact of antibody specificity and reflects a genuinely different occupancy profile of H2B acetylation sites.

2. In Figure 5, the authors attempted to validate the enhancer activity of the H2B acetylated enhancers by overlapping the regions enriched by the various H2B lysine acetylated antibodies with what has been mapped as active enhancers. The degree of overlap was between 31-53%, suggesting a false negative rate of 47-69%. Interestingly, the best H2B acetylation antibodies performed better than H3K27ac and e-

RNA based methods, although H2BK20ac used throughout the manuscript (and the accompanying) only maps 45% of active enhancers. The authors should discuss the implication of these results for the use of H2BK20ac to map active enhancers.

This appears a misunderstanding.

The extent of overlap in this analysis depends on the size of the dataset used for comparison. As the Reviewer notes, only ~45% of H2BK20ac (9000/22000) peaks overlap with active enhancers defined by MPRA. The reason for this is that the MPRA dataset only contains 10,000 active enhancers, but H2BK20ac has ~23K peaks, hence a complete overlap with H2BK20ac is not expected because the number of H2BK20ac peaks is greater. An incomplete overlap likely results from the fact that many weak enhancers did not score significantly in MPRA assays. This is supported by the finding that the overlap of H2B acetylation ChIP-seq peaks with MPRA+ enhancers is related to the intensity of H2B acetylation. In the bottom quartile, only a small fraction of H2BK20ac peaks overlap with MPRA regions, but in the top quartile, 72% of H2BK20ac peaks overlap with MPRA+ enhancers. The overlap of the top 25% peaks is remarkable given that the library used for the MPRA analyses only covered 83% of the genome with modest (9x) coverage (i.e. the maximum expected overlap is 83%). To our knowledge, this represents one of the highest validation rates for chromatin mark-defined enhancers in genome-scale MPRA assay.

3. Using a combination of H2BK20ac and/or H3K27ac the authors show that 88% of all distal ATAC+MPRA+ regions (active enhancers) overlap with these modifications (Figure 6). Based on this result it would be logical to suggest that it would be more appropriate to use H2BK20ac and H3K27ac in combination to map active enhancers, however, this is not the take home message of the manuscript. Again, it would be useful if the authors discuss this in the manuscript.

There appears a misunderstanding.

We agree with the reviewer that the use of H2BNTac and H3K27ac can offer complementary information in some comparisons. We make no statement that H3K27ac should not be used in combination with H2BNTac for mapping enhancers. Rather, the data shown in Fig. 4b illustrates the usefulness of H2BNTac with other chromatin mark in discriminating active and inactive enhancers and promoters. We stated that *“These results demonstrate that genomic occupancy profiles of H2BNTac, H3K27ac, H3K4me3, and H3K4me1 are non-identical. While none of the marks display absolute specificity, H2BNTac complements other marks and affords valuable information in confidently discerning candidate active enhancers.”*

Regarding the cited Fig. 6a, combination of H2BK20ac and H3K27ac provides little added value because either of the marks is sufficient to map most of the active candidate enhancers. Nonetheless, we agree

that, depending on the research question, combining both marks can be useful. If the goal was to simply map active regulatory regions, and not to discriminate active enhancers from promoters, we believe that H3K27ac is a good choice. But if the goal was to discriminate active enhancers from promoters, or to determine the dependency of gene activation of CBP/p300 activity, then H2BNTac presents a better choice. If H2BNTac and H3K27ac are used in combination, they can help in mapping both active enhancers and promoters and in combination with H3K4me3, these marks can be used to discriminate candidate enhancers from promoters. There are also several other useful chromatin marks, including H3K4me1, DHS, MED1, etc. If the resources were unlimited, we agree with the reviewer that using multiple markers can aid in more confidently mapping enhancers and discriminating them from other CREs.

Reviewer #4:

Remarks to the Author:

Summary

The authors show the remarkable importance of H2B acetylation in discriminating between levels of gene activity in a way that outperforms standard H3K27ac approaches. The work has nice functional data using inhibitors of transcription/acetylation. I suspect this work will greatly increase the use of H2B acetylation among epigenetic researchers in many fields, and will enhance the precision of future research directions for those using histone marks to delineate functional roles of chromatin machinery.

Overall comments

This paper represents a major advance in understanding the modifications of histone and their relationship to enhancer activity and gene control. The work is of high interest to the field, and is technically robust.

We thank the reviewer for evaluating our work. We are pleased that the reviewer found our work of high interest. Below, we provide a point-by-point response to Reviewer's comments.

Major points

1. Only minor criticisms were noted. Overall, I recommend reducing the number of figures to reduce redundancy, adding additional types of figure panels to increase the diversity within each figure, and I also recommend adding one or more cartoon representations of the main finding.

Following the reviewer's recommendation, we removed panels that were somewhat redundant. For example, several of the violin plots conveyed the same message, and were thus removed to minimize the redundancy. We will also add a cartoon diagram showing the relative specificities of chromatin marks for enhancers versus promoters.

Novelty/notability claims

2. The paper's novelty is strong. A literature survey indicated that, indeed, the genomic locations of H2B acetylation have not been (to my surprise!) elsewhere shown.

We fully concur with the reviewer assessment, and sincerely believe that our work will change this and encourage further research into H2B acetylation.

Minor points

3. Minor grammatical errors have been noted in an attached PDF of the manuscript via comments/notes.

We thank the reviewer for pointing out the errors, which are corrected in the revised version.

4. The authors discuss the role of H2B acetylation in relationship to chromHMM defined chromatin. It would be really nice to see the "emission" data and defined chromatin states with and without H2B acetylation. Here is an example:

We will perform these analyses.

5. There is a glaring lack of genome browser tracks that are need in the main figures to reveal the quality of the ChIP-seq data, and to assist in giving readers an intuition of the patterns being claimed through boxplots and correlation matrices. To fix this, the authors should promote Extended Figure 8b to the main figure set, or make a new genome browser figure that shows the differences in H3 acetylation and H2B acetylation at diverse regulatory elements. Another example is in the CDK inhibition data – the increase of H2B acetylation after inhibition of transcription is beautifully shown in Ext Data Figure 10C – perhaps 1 of these gene can be moved to the main figure 3, and some of the intellectually redundant violin plots can be moved to supplementary.

Following the reviewer's suggestion, we include example gene tracks to show the differences in H3K27ac and H2B acetylation.

6. The role of H3K v H2B acetylation in predicting response to A485 is striking. The authors should add a metagene/average plot of the histone marks in Figure 7b, centered around the TSS of genes that are strongly down regulated, and compared to the same data plotting at the TSS of genes that are non-responsive.

The suggested analyses are included in the revised figure.

Figure comments

7. Figure 5 and Figure 6 were swapped, causing some confusion for this reviewer.

We apologize for this mistake. It is rectified in the revised version.

8. Figure 6a should have H3K27ac as an additional row of plots with all the same measurements calculated.

The suggested analyses are included in the revised figure.

Decision Letter, Appeal:

7th Oct 2022

Dear Chuna,

Thank you for asking us to reconsider our decision on your manuscript entitled "A unique H2B acetylation signature marks active enhancers and predicts their target genes". I have now discussed the points of your letter with my colleagues, and we think that you have some valid points and that the revision plan sounds encouraging. We therefore invite you to revise your manuscript along the lines that you propose.

When preparing a revision, please ensure that it generally complies with our editorial requirements for format and style; details can be found in the Guide to Authors on our website (<http://www.nature.com/ng/>).

Please be sure that your manuscript is accompanied by a separate letter detailing the changes you have made and your response to the points raised. At this stage we will need you to upload:

1) a copy of the manuscript in MS Word .docx format.

2) The Editorial Policy Checklist:

<https://www.nature.com/documents/nr-editorial-policy-checklist.pdf>

3) The Reporting Summary:

(Here you can read about the role of the Reporting Summary in reproducible science:

<https://www.nature.com/news/announcement-towards-greater-reproducibility-for-life-sciences-research-in-nature-1.22062>)

Please use the link below to be taken directly to the site and view and revise your manuscript:

[Redacted link]

With kind wishes,

Tiago

Tiago Faial, PhD
Chief Editor
Nature Genetics
<https://orcid.org/0000-0003-0864-1200>

Author Rebuttal, Appeal:

See inserted PDF

A point-by-point response to Reviewers' comments

We thank the Reviewers for evaluating our work and for providing constructive feedback, which we believe has helped greatly in improving the manuscript. We thoroughly address all the points raised, and the results of new analyses further support and strengthen our original conclusions. The main changes are summarized below.

1. We compared an equal number of top-ranked H2BNTac and H3K27ac for their overlap with MPRA-defined active enhancers. The results confirm a greater overlap of H2BNTac peaks with MPRA-defined enhancers.

2. Following the Reviewer's feedback, we revised the title, abstract, and manuscript text to clarify that H2BNTac predicts genes regulated by CBP/p300 activity. We no longer equate CBP/p300 activity with enhancer function and do not refer to A-485 regulated genes as enhancer targets. We believe that this change more accurately describes our results and avoids any confusion.

3. Based on the comments from Reviewer #3, and in consultation with the handling editor, the data generated using the cross-reactive H2BK5ac (EP857Y) antibody was moved to supplementary figures, which we hope helps in minimizing the confusion.

4. We extensively revised the text to better explain and clarify the performance of the ABC model and promoter H2BNTac in predicting A-485 regulated genes. Following the Reviewers' suggestion, we evaluated the usefulness of H3K27ac and H2BNTac in the context of the ABC model for predicting enhancer-gene pairs in large-scale CRISPRi data (Gasperini et al., PMID: 30612741; Fulco et al., PMID: 31784727). The outcome of these analyses is briefly presented in the manuscript and explained in detail in our response to comment 4 from Reviewer #2.

Reviewer #1:

Remarks to the Author:

This manuscript by the Choudhary lab explores N-terminal acetylation of histone H2B as a general marker for enhancers. The authors generate ChIP-seq data using a variety of antibodies recognizing different H2BNTac residues in mESCs and K562 cells. The authors argue that H2BNTac forms a distinct signature from H3K27ac and other chromatin marks that are in wide use. The H2BNTac signature is most prominent at distal intergenic sequence and a handful of promoters, suggesting that it may have some specificity for specific targets. There is a lot that I like about this paper. The distribution of H2BNTac marks does appear to be distinct from K27ac (pending the resolution of several technical comments, below). I am also fairly convinced that H2BNTac marks genes that are more sensitive to CBP/p300 based on Fig. 7. Lastly, I admire the author's care in interpreting antibody specificity of two H2BK5ac antibodies in the text. However the premise that H2BNTac is a better marker for enhancers in general is not convincing to me: First, data comparing recovery of MPRA peaks using H2BNTac and H3K27ac in Fig. 5 and 6 does not make H2BNTac stand out as unequivocally better than H3K27ac; and Second, I have some major reservations about the analysis comparing promoter H2BNTac to ABC scores as noted in detail below. Broadly, my takeaway is that H2BNTac is an interesting mark that warrants further study, but evidence that it is indistinguishable from the function of an enhancer is not sufficient at this time.

We thank the Reviewer for carefully reviewing our work and offering insightful comments. We are pleased to read his/her remark that "*There is a lot that I like about this paper.*" We performed the suggested analyses and revised the manuscript to thoroughly address all points raised by the Reviewer.

The Reviewer noted that “*recovery of MPRA peaks using H2BNTac and H3K27ac in Fig. 5 and 6 does not make H2BNTac stand out as unequivocally better than H3K27ac*”.

We agree that based only on the data presented in Figs. 5 and 6, H2BNTac does not stand out as a much better marker of enhancers than H3K27ac. We would like to clarify that the data presented in Figs. 5 and 6 were meant to show that H2BNTac and H3K27ac are similarly sensitive in detecting active enhancers (defined by STARR-seq or eRNA expression). The conclusion, that H2BNTac is a better marker of active enhancers, and a better predictor of CBP/p300 activity and CBP/p300-dependent gene regulation than H3K27ac, is supported by the data presented in Fig. 1b, Fig. 2, Fig. 4, Fig. 7 and 8.

We consider sensitivity and specificity as two distinct attributes of an enhancer marker. A marker can be more sensitive but less specific or vice versa. For example, DHS/ATAC-seq is similarly (or perhaps more) sensitive than H3K27ac in marking enhancers, but H3K27ac has greater specificity in marking active enhancers than DHS/ATAC-seq. Thus, if judged based on sensitivity, H3K27ac is not a better marker than DHS/ATAC-seq. However, if judged based on specificity, H3K27ac is a better marker of enhancers than DHS/ATAC-seq. Our results show that H2BNTac has even greater specificity than H3K27ac for enhancers, and therefore, we suggest that it is a better marker of active enhancers.

To highlight the importance of specificity, it is worth noting that, to a greater or lesser extent, almost all the widely studied histone acetylation marks can be detected at highly active enhancers and promoters. H3K27ac abundance in enhancers is relatively higher than other marks, but, as shown in **Fig. 2e**, the ratio of H3K27ac/H3K9ac is only modestly higher in enhancers than in promoters. In the same analyses, the ratio of H2BNTac/H3K9ac and H2BNTac/H3K27ac in enhancers is much higher than in promoters. Therefore, H2BNTac affords greater distinction than H3K27ac. In promoter regions, H3K27ac and H2BNTac show strikingly different correlations with H3K4me3 (**Fig. 1b**). Unlike H3K27ac, in promoter regions, H2BNTac shows a positive correlation with H3K4me1 (**Fig. 1b**), a well-known marker of enhancers.

We acknowledge that the difference in H3K27ac and H2BNTac is not always qualitative, but the quantitative differences in H3K27ac and H2BNTac at promoters are notable. This is reminiscent of H3K4me3 specificity. The differences in H3K4me3 at promoters and enhancers are not always qualitative, but H3K4me3 abundance shows large quantitative differences in enhancer and promoter regions, and hence, it is a useful marker of promoters. Conversely to H3K4me3, H2BNTac/H3K27ac ratio is much higher in enhancers than in promoters, and therefore, we believe that H2BNTac is a good marker for defining active enhancers and discriminating them from constitutively active promoters.

Beyond mapping enhancers, H3K27ac has been used as a proxy for defining locus-specific CBP/p300 activity and enhancer strength (for example, in the ABC model). Our findings suggest that H2BNTac intensity is a better predictor of CBP/p300-dependent gene activation.

Because of the above-mentioned attributes, we believe that H2BNTac nicely complements the currently used chromatin features and will be broadly useful for improved genome annotation and for predicting CBP/p300-regulated genes.

Below, we provide a detailed point-by-point response to specific comments.

Major comments:

1. Throughout the manuscript the authors rely on comparisons between H3K27ac and H2BNTac ChIP-seq data. ChIP-seq data has a ton of assay and biological noise that can make two experimental datasets for the same mark quite discrepant from each other. Therefore the origin of H3K27ac datasets and its relationship to H2BNTac datasets needs to be spelled out very clearly: Are these datasets both prepared in the same person's hands at the same time from the same cell passage (i.e., what we would consider technical replicates if the ChIP-seq was performed on the same mark)? Different datasets generated by the same person in the Choudhary lab at different times (i.e., biological replicates)? Or different datasets generated by different labs at a different time (these are likely to have a higher degree of both biological and technical noise)? Reading the Methods section it is clear that at least some of the H3K27ac data from both mESCs and perhaps all of the data from K562 was scraped from existing datasets from other labs. Although it is great to reuse data, differences observed between these datasets need to be interpreted with extreme caution. Which dataset used in each comparison should be spelled out. Using a dataset from a different lab could result in clustering datasets across labs as observed in Fig. 1 b-c, overlaps between peaks, and other comparisons made by the authors.

We fully agree with the Reviewer's point. The key ChIP-seq data used for our comparison (H3K27ac, H3K9ac, H3K27me3, H3K4me3, and MED1) were generated in-house, by the same person, using the same protocol. Also, for K562 H3K27ac, and H2BK20ac data were generated in-house, by the same person, using the same protocol. The H3K4me1, p300, NANOG, and OCT4 ChIP-seq data are from our recently published work (Narita et al.), and were generated using the same pipeline. Some of the ChIP-seq was performed from the same batch of cells, but the experiments were performed over a long period, and ChIP was performed from multiple batches of the same cells, and the results were highly consistent. Because of the striking differences in H2BNTac and H3K27ac occupancy in the promoter regions, some colleagues in the field suggested that we should also include published H3K27ac ChIP-seq data in our comparison. Therefore, we included two previously published H3K27ac ChIP-seq datasets from mESC, but this is in addition to the H3K27ac data generated in our lab. Reassuringly, three different H3K27ac data, from different labs, show similar occupancy, and their occupancy is distinct from H2BNTac. We did include some publicly available data in our comparisons (i.e. CTCF, H3K9me3, H3K36ac, etc.), but most of those marks are not particularly enriched in enhancers and do not form the basis of the main claims of the work. In summary, the different clustering between H3K27ac and H2BNTac, and between these marks and other marks, is not due to technical differences in ChIP-seq analyses.

As a note, we may mention that in our ongoing work we use H2BNTac as an enhancer marker and have performed ChIP-seq analyses of H3K27ac, H2BNTac, and H3K4me3 in seven different mammalian cells, and the results are highly consistent with the data shown in this manuscript (unpublished data).

2. The authors primarily make use of background-subtracted signals in peak calls in their analysis. While there are substantial advantages of this approach, this strategy makes a non-linear transformation that could impact correlations between the marks. I would welcome the addition of analyses comparing raw counts, both in peaks and in genomic windows, in the extended data as a complement to the correlations. As Spearman's correlation is a rank order that is often best for genomic analyses, it does have more difficulty if large numbers of the observations are at or below background, and therefore have a random order - and I would also therefore encourage the authors to add log-transformed Pearson's correlations. Along the same lines, the authors should spell out whether correlations presented in Fig. 1b are between the universe of all peaks for any of the marks

(or all promoters in and outside of peak calls)? Or between peaks that overlap between each pair? This type of detail matters a great deal when interpreting peaks.

We thank the Reviewer for this helpful suggestion. Pearson's and Spearman's correlations in promoter regions, calculated using peaks or genomic windows, with or without background subtraction, appear virtually identical (**Extended Data Fig. 3**). However, in distal regions, correlations based on genomic windows do not appear to reflect the known differences in the occupancy of H3K27ac, H3K4me3, and H3K27me3. This is not entirely surprising because, in intergenic regions, genomic windows include large swathes of non-peak regions, and correlations reflect similar background signals. Because genomic windows for promoters are fixed near the known promoters (+/-1kb from TSS), correlation in promoters is not influenced by the noise in non-peak regions. Correlations presented in Fig. 1b, we used the universe of all peaks, and this is specified in the figure legend.

To avoid confusion to readers, and to minimize the number of redundant figures (as recommended by Reviewer #4), these analyses were not included in the manuscript. For the Reviewer's information, these correlations are shown at the end of this rebuttal (**Reviewer Fig. 1**). If the Reviewer recommends, we will happily include this figure in the manuscript.

3. I find the section comparing promoter H2BNTac with the ABC model to be misleading. The purpose of the ABC model is to predict whether a specific enhancer will regulate the expression from a specific target promoter. The setup in this paper appears to be slightly different: the authors treat with the CBP/p300 inhibitor, A-485, and attempt to classify which genes are sensitive to down-regulation. The assumption made by the authors is that genes regulated by A-485 require enhancers for their up-regulation and all other genes do not. I am not at all convinced that this assumption is appropriate. Rather, as shown elsewhere in the paper, the authors are able to predict which genes are sensitive to A-485 inhibition using H2BNTac based on ChIP-seq signal in the promoter. Equating this to the sensitivity of the gene to any enhancer is not appropriate.

We thank the Reviewer for raising this point. We admit that how we presented the results of the ABC model was confusing. We sincerely apologize for this and assure the Reviewers that this was unintentional; we have no reason to undermine the performance of the ABC model.

To our knowledge, virtually all of the functionally characterized native enhancers require CBP/p300 activity for target gene activation. Nonetheless, we agree with the Reviewer that we should not assume that all A-485-regulated genes require enhancers and all other genes do not. We also acknowledge that we cannot rule out an enhancer-independent role of CBP/p300 in gene activation. Therefore, we agree with the Reviewers that we must be cautious in our interpretations and not equate the functions of CBP/p300 activity and enhancers in gene activation. We revised the text to clarify that what we study here is gene regulation by CBP/p300 activity, not the function of enhancers per se.

We fully agree with the Reviewers that the premise of the original ABC model was to predict whether a specific enhancer will regulate the expression of a specific gene. Until now, the field has lacked the tools for predicting genes regulated by CBP/p300 activity. Because the ABC model uses CBP/p300-catalyzed H3K27ac, we asked if the model could be deployed for predicting A-485-regulated genes. Indeed, the model appears to perform similarly, if not better, at predicting A-485-regulated genes than predicting enhancer-gene pairs from unbiased CRISPRi screens (for further details on it, please see our response to comment 4, Reviewer #2). To avoid any confusion, we present the results of the ABC model and promoter H2BNTac data in separate figure panels (**Fig. 8**).

We hope that the Reviewer will find this comparison fair and accurate. We are fully committed to presenting the findings most accurately and fairly and are open to any further suggestions.

Reviewer #2:

Remarks to the Author:

In their manuscript, Narita et al report that histone H2B N-terminus multisite lysine acetylation (H2BNTac) marks active enhancers and promoters of enhancer-regulated genes. The correlation between H2BNTac and active enhancers as well as dependency of H2BNTac on p300/CBP is solid and interesting. From the presented data I am quite convinced that H2BNTac represents a novel active enhancer-associated mark. Unfortunately, beyond this observation, most of the other claims in the manuscript are poorly supported by the data. In addition, presentation of the results is obfuscated to the point that it is difficult to understand what has been done. Altogether, we cannot support publication of the manuscript in its current form.

We thank the reviewer for diligently reviewing our work and offering very helpful comments. We take the Reviewer's criticism wholeheartedly and have made our best efforts to address the raised points. Below, we offer a point-by-point response to his/her comments.

Major points:

1. To make their point that H2BNTac is more closely correlated with active enhancers than H3K27ac (a histone modification mark that is most commonly used as an indicator of active cis-regulatory elements), authors compare the defined set of peaks associated with each mark against an orthogonal assay, such as MPRA. However, it is difficult to assess whether the comparison was fairly made. Some of the H2BNTac marks have fewer significant peaks than H3K27ac and therefore may perform better (e.g. in proportion of peaks with MPRA signal) but only because they represent more robustly active enhancers, with likely also higher H3K27ac (in fact, authors themselves point to that problem in relation to other data). To make fair comparisons, authors should compare equivalent numbers of peaks matched for H3K27ac enrichment level, with or without H2BNTac enrichment. Or at the very least, compare the equivalent numbers of top peaks for each mark.

We fully agree with the Reviewer's concern and thank him/her for the excellent suggestion. As suggested by the Reviewer, we compared an equivalent number of top n peaks for each mark (**Fig. 5a**). In this comparison, H2BNTac ranks first, H3K27ac ranks second, H3K4me1 ranks third, and MED1 ranks last. We also analyzed an equal number of top-ranked H3K27ac peaks, with or without H2BNTac enrichment (**Extended Data Fig. 14b-c**). A greater fraction of MPRA+ATAC+ peaks overlaps with H3K27ac⁺H2BNTac⁺ peaks as compared to H3K27ac⁺H2BNTac⁻ peaks, indicating that it is mainly the H3K27ac⁺H2BNTac⁺ regions that act as enhancers. Together, these results are fully consistent with our original conclusions.

2. On a related note, the authors' overview of the relationship between the various H2BNTac marks and H3K27ac (Fig 1a) is presented in a confusing and unintuitive manner. A much simpler and more easily interpretable visualization would be to use heatmaps of ChIP-seq signal (along with meta plots if desired) for each mark across the superset of H3K27ac + H2BNTac peaks, segregated by proximal vs distal and centered at DNase or ATAC peak summits.

We apologize for the confusion. We now additionally present the ChIP-seq data in a heatmap format as suggested by the reviewer (**Extended Data Fig. 4**). We retained Fig. 1a because based on an incomplete overlap of H3K27ac and other histone acetylation marks peaks, previous studies

suggested the existence of different classes of enhancers lacking H3K27ac (PMID: 23990607; PMID: 27089178; PMID: 26957309). In our analyses, we also found non-overlapping H2BNTac peaks, but those peaks are low abundant, suggesting that an incomplete overlap likely occurred due to technical difficulty in detecting low abundant peaks. We felt that this message did not come across as clearly in heatmaps. We wished to clarify the reasons for how the figures were organized. If the Reviewer has a different view on it, we will happily re-organize the figures as he/she suggests.

3. H2BNTac at promoters seems to predict A-485-dependent gene transcription quite well, and the authors claim this represents the dependency of these genes on enhancers. However, the authors have not established whether A-485 in fact inhibits activity of all (or even most) enhancers, nor have they excluded a possibility that a subset of promoters is simply more dependent on p300/CBP and largely responsible for the observed expression changes. The observation that TSS H2BNTac signal outperforms a modified ABC score (Fig 8) in fact suggests that A-485 may selectively impact the transcription of genes that have certain promoter features rather than certain enhancer landscapes, and thus the claim that ‘promoter H2BNTac signal predicts enhancer target genes’ is grossly overblown. Interestingly, previous work on p300/CBP and H3K27ac also reports much better correlation of gene expression changes with acetylation changes at promoters rather than enhancers, but with a different interpretation (quote from Martire et al., 2020: Transcriptional dysregulation is generally correlated with dysregulation of promoter acetylation upon depletion of p300 (but not CBP) and appears to be relatively independent of dysregulated enhancer acetylation; PMID: 32690000). Fig. 3a also shows that changes in H2BNTac upon A-485 are concentrated at promoters (while H3K27ac changes are minimal), raising the question of whether A-485 is truly inhibiting enhancer function. This is confusing given the authors’ claims that H2BNTac is specific to enhancers rather than promoters.

Our analyses showed that promoter H2BNTac correlates well with A-485-induced gene downregulation. This result can be interpreted in two different ways. (1) Promoter H2BNTac reflects a direct function of CBP/p300 at promoters, as the Reviewer appears to suggest. (2) Promoter H2BNTac may reflect, at least partially, the function of enhancer associated CBP/p300, as we implied in our original submission. After carefully considering the Reviewer’s comment, we acknowledge that these two possibilities are non-mutually exclusive.

Because this is quite a complex and important point, we provide a concise and detailed response to it.

Concise response

We agree with the Reviewer’s point and concede that we cannot rule out the possibility that promoter H2BNTac reflects a direct function of CBP/p300 at promoters. As such the distinction between enhancers and promoters is blurred (PMID: 31605096). A sizeable fraction of mammalian enhancers has autonomous promoter activity (PMID: 28024146), and several high-profile studies have reported that promoters can act as enhancers (PMID: 27783602; PMID: 28581502; PMID: 28417999; PMID: 29673772). Furthermore, the recent analyses by the ENCODE consortium (PMID: 32728249) reported that a vast majority of actively transcribed genes have TSS proximal enhancers (termed as proximal enhancer-like sequences, pELS) (Suppl. Fig. 3b in PMID: 32728249, and our observations from re-analysis of the ENCODE data, not shown here). Given the overlapping features of promoters and enhancers, it is extremely difficult, if not impossible, to conclusively separate enhancers and promoters, and to delineate CBP/p300 function at promoters and enhancers. Therefore, in response to the Reviewer’s point, we revised our interpretation. We conclude that promoter H2BNTac correlates with gene regulation by CBP/p300 activity (without making a direct claim that this reflects enhancer function). We added that: *“Although CBP/p300 is thought to*

predominantly act through binding to enhancers, we do not rule out the possibility that it also acts at promoters." We hope that our revised interpretation reconciles with the Reviewer's interpretation.

Detailed response

Because of the complexity of the point, we separately address different parts of it.

"However, the authors have not established whether A-485 in fact inhibits activity of all (or even most) enhancers, nor have they excluded a possibility that a subset of promoters is simply more dependent on p300/CBP and largely responsible for the observed expression changes. The observation that TSS H2BNTac signal outperforms a modified ABC score (Fig 8) in fact suggests that A-485 may selectively impact the transcription of genes that have certain promoter features rather than certain enhancer landscapes..."

CBP/p300-catalyzed H3K27ac and H2BNTac mark virtually all distal enhancers, and A-485 treatment globally reduces acetylation in enhancer regions. Furthermore, acetylation of H2B sites is reduced by 95%–99% after A-485 treatment or genetic deletion of CBP/p300 (PMID: 29804834). Therefore, we believe that A-485 inhibits acetylation at most, if not all, active enhancers.

We cannot rule out the possibility that A-485 may selectively impact the transcription of genes that have certain promoter features rather than certain enhancer landscapes. However, it is worth noting that A-485 acts through CBP/p300, and >90% of p300-bound strong peaks occur in >1kb from TSS (Nature 2009, PMID: 19212405). CBP/p300 binding can be detected in promoters, as well as in Polycomb-repressed regions, but p300 binding is not a good predictor of its activity (PNAS 2010, PMID: 21106759, Nature 2011, PMID: 21160473). We detect p300 binding in promoters, but p300 binding between A-485 downregulated and not regulated gene promoters is not appreciably different (**Extended Data Fig. 17b**).

One feature that does appear to distinguish A-485 regulated promoters is the weak autonomous promoter activity. A-485-regulated promoters have lower autonomous promoter activity in ectopic reporter assays, yet, in their native context, A-485-regulated and not-regulated promoters drive similar level of gene expression (measured by nascent transcription output) (<https://www.biorxiv.org/content/10.1101/2022.07.18.500456v2>). To us, this implies that, in the native context, A-485-regulated promoters are activated by distal enhancers. If CBP/p300 directly activated A-485-regulated promoters, they will not be expected to have weak promoter activity in a reporter assay. Interestingly, promoter H2BNTac shows a much stronger correlation with H3K4me1, another enhancer marker, and H3K4me1 predicts A-485 downregulated genes with a much better accuracy than H3K27ac, nearly at the same accuracy as H3K27ac-based ABC model (**Fig. 8a, b**).

We wish to avoid excessive speculation in the manuscript, but we note the reports that enhancers can form phase-separated condensates (although this remains a matter of active debate). Examination of individual gene tracks show that histone acetylation can spread away from the enhancer centers (**Figs. 2f, 3d; Extended Data Figs. 11b, 12c**), and in some instances, acetylation can spread to 10s of kilobases (i.e. at super-enhancers). Thus, it remains possible that H2BNTac in A-485 regulated promoters may occur through the promiscuous activity of enhancer-associated CBP/p300. Regardless of whether this speculation is valid or not, our data demonstrate that H2BNTac is a good predictor of genes impacted by A-485 and this is the main claim of the revised manuscript; we no longer claim that A-485 exclusively impacts enhancer function.

"Interestingly, previous work on p300/CBP and H3K27ac also reports much better correlation of gene expression changes with acetylation changes at promoters rather than enhancers, but with a different interpretation (quote from Martire et al., 2020: Transcriptional dysregulation is generally

correlated with dysregulation of promoter acetylation upon depletion of p300 (but not CBP) and appears to be relatively independent of dysregulated enhancer acetylation; PMID: 32690000).”

With interest, we read the work by Martire et al.. The quoted statement is based on the observation that, upon p300 knockdown, the change in gene regulation is weakly correlated with change in the H3K27ac at distal enhancers ($r= 0.35$, $p=2.2e-16$) than at promoters ($r= 0.54$, $p=2.2e-16$). Despite this difference, please note that change in gene expression is highly significantly correlated with change in H3K27ac at enhancers.

We agree with the Martire et al. conclusion, that gene dysregulation correlates better with the decrease in H3K27ac at promoters than at enhancers, but in our view, this is insufficient to establish that p300 activates genes independently of enhancers. The functional impact of enhancers depends on both the enhancer strength and the distance from the regulated TSS, and the regulatory impact of enhancers decreases non-linearly with increasing distance from TSS (PMID: 35418676). H3K27ac ChIP signal only accounts for enhancer strength, and not for enhancer distance. Because there is not a direct 1:1 quantitative relationship between enhancer H3K27ac and activation of their target promoters, a weaker correlation between enhancer H3K27ac and gene regulation is to be expected.

We do not consider CBP/p300 function in enhancer and promoter to be mutually exclusive. It appears reasonable to consider that both mechanisms may operate in parallel and in some instances, promoter acetylation may reflect the activity of enhancer-associated CBP/p300 and in other instances, it may reflect the direct function of CBP/p300 in promoters. However, until more evidence emerges, a weaker correlation between enhancer-associated H3K27ac and gene regulation cannot be construed as solid evidence for an enhancer-independent function of p300 in gene activation. If the Reviewer is aware of strong evidence showing a direct role of CBP/p300 in activating native promoters, or CBP/p300-independent gene activation by endogenous enhancers, we will be highly grateful if the Reviewers could share the relevant literature with us. We strive to present our findings as accurately as possible and will be happy to revise the text and include the suggested references. Because we cannot rule out a direct CBP/p300 function in promoter activation, we have revised the text and we no longer equate CBP/p300 function with the enhancer function.

“Fig. 3a also shows that changes in H2BNTac upon A-485 are concentrated at promoters (while H3K27ac changes are minimal), raising the question of whether A-485 is truly inhibiting enhancer function. This is confusing given the authors’ claims that H2BNTac is specific to enhancers rather than promoters.”

We wish to draw the Reviewer’s attention that these analyses were done at a relatively early time point (15 min) after the A-485 treatment. Although acetylation is not fully abrogated at this early time point, H3K27ac and H2BK20ac are globally reduced (log2 median ratio is significantly lower than 0, $p=2.2e-16$). Unlike H2BNTac, H3K27ac is similarly reduced at promoters and enhancers.

As the Reviewer notes, H2BNTac is indeed more strongly reduced at promoters and actively transcribed regions, but from this, it should not be interpreted that A-485 more prominently inhibits promoter-associated CBP/p300 than enhancer-bound CBP/p300. As our result shows, the reason for the more rapid reduction in promoter H2BNTac is the transcription-coupled rapid exchange of H2A-H2B dimers. This is also the reason why enhancers localizing in the actively transcribed gene body and intergenic regions show differences in A-485-induced decrease in acetylation, and why the rate of deacetylation in promoter and gene body enhancers is associated with the level of transcription.

We apologize for the confusion caused by the fact that H2BNTac is being presented as an enhancer marker, but also shown to mark a subset of promoters. We kindly request the Reviewer to judge the specificity in the general context of the specificity of other chromatin marks. For example, H3K4me3 is widely mentioned as a marker of promoters. But, as shown in **Fig. 2a**, H3K4me3 is detectable at a quarter of distal enhancers, and >80% of super-enhancers. Thus, neither H3K4me3 nor H2BNTac has absolute specificity for promoters and enhancers, respectively, but their relative levels in these regions are sufficiently large to provide useful practical information for mapping cis-regulatory regions. We do not wish to claim that H2BNTac is exclusively present in enhancers.

4. To establish that H2BNTac outperforms H3K27ac in predicting enhancers regulating a given gene in the context of the ABC model, the authors should reanalyze data from the original ABC paper (Fulco et al, Nature Genetics 2019) and others that have specifically identified functional enhancer-promoter pairs (Gasperini et al, Cell 2019) by performing CRISPRi of individual enhancers and assessed resulting changes on gene expression. If the authors' claim is true, using H2BNTac instead of H3K27ac would improve the power of the ABC model in predicting specific enhancers regulating a given gene. Indeed, the authors already have H2BNTac ChIP-seq in K562 cells (the cell line in which the enhancer CRISPRi experiments were performed). Promoter acetylation should be excluded from the ABC score analysis if the comparison is to assess predictive value of these marks in enhancer activity. Even if the authors believe that their heuristic of enhancers regulating the closest TSS with H2BNTac is the best strategy for identifying enhancer-promoter pairs, they should systematically evaluate how well this strategy performs in predicting the functionally validated enhancer-promoter pairs from Fulco et al and Gasperini et al.

We thank the Reviewer for raising this excellent point. After reading the thoughtful comments from Reviewers, we recognize that how we compared the ABC model in our analyses was unclear and confusing. We sincerely apologize for not having presented these data in a better manner. We extensively revised the manuscript text to better clarify this point.

Following the Reviewer's recommendation, we also evaluated H3K27ac/H2BNTac performance in the context of the ABC model using large-scale CRISPRi datasets (**Extended Data Fig. 17c**). We used the Fulco et al. dataset, and calculated ABC scores using in-house generated H3K27ac or H2BK20ac ChIP-seq data (while keeping other parameters unchanged). The ABC model performed slightly better when using H2BK20ac (AUPRC of 0.65 for H3K27ac versus AUPRC of 0.69 for H2BK20ac) (**Extended Data Fig. 17c**). This result is consistent with the modest improvement in the ABC model's performance by H2BK20ac for predicting A-485 regulated genes in mESC (**Fig. 8**), and supports the idea, even if indirectly, that at least a portion of A-485-regulated genes is regulated by enhancers. Unfortunately, in the Gasperini et al., the ABC model performed poorly (AUPRC 0.25- 0.27) (**Extended Data Fig. 17c**). If the performance of the H3K27ac-based ABC model is so poor, it is not surprising that we see no improvement by substituting H3K27ac by H2BK20ac.

We also wish to point out that, even if the ABC model performed as expected, we would not expect a large difference in the performance of H3K27ac and H2BNTac. This is because H3K27ac and H2BNTac are similarly abundant in the distal regions, and the ABC model mainly uses H3K27ac intensity in distal regions. Thus, in the context of the ABC model, we would not expect a big difference in their performance. The main difference between H3K27ac and H2BNTac lies in the promoter-proximal regions, but the promoter H3K27ac signal is excluded in the ABC model for making predictions. This provides a likely explanation why, in the context of predicting A-485 regulated genes and for predicting enhancer targets in the Fulco et al. dataset, substituting H3K27ac with H2BNTac modestly increase the ABC model's performance.

We apologize for the very long answer to points 3 and 4, but we felt that it is important to thoroughly clarify these points to avoid any misunderstandings.

Reviewer #3:

Remarks to the Author:

Takeo et al have systematically studied the genomic distributions of histone H2B N terminus multisite acetylation (H2BNTac) and establish H2BNTac as a potential signature for cell-type-specific active enhancers.

Their main findings include:

1. Using rabbit monoclonal antibodies recognizing specific acetylation sites on H2B, and through ChIP-seq experiment they find that H2BNTac is predominantly enriched at cell-type-specific active enhancers.
2. H2BNTac has a more pronounced enhancer-specificity than H3K27ac, which is also catalysed by CBP/P300 and widely used in defining enhancers but also occupies active promoters. Based on the observation that transcription inhibition could lead to increased H2BNTac at promoter and gene body regions, but not at enhancers, they conclude that transcription-coupled histone H2A-H2B exchange at promoter and gene body regions contributes to the relative depletion of H2BNTac at gene proximal regions but enrichment at distal regulatory regions.
3. Furthermore, they find H2BNTac by itself exhibits the highest specificity in decorating active enhancers and outperform other histone modifications like H3K27me3 and H3K4me1 in differentiating active enhancer from active promoters.
4. They suggest that incorporating the promoter H2BNTac information could bring in more accuracy when using chromatin confirmations to identify enhancer and target gene pairs, suggesting that the promoter H2BNTac signal reflects the activity of enhancer-promoter interaction.

The major novelty in this manuscript is to improve the Chromatin-enrichment method by which enhancers are mapped. However, because H2BK20ac already has been shown to provide a distinctive enhancer signature (reference 9), the submitted work is rather incremental. Moreover, because the manuscript does not provide any novel biological or mechanistic insights into enhancer regulation, we are unable to recommend the manuscript for publication in Nature Genetics. Despite we cannot recommend the manuscript for publication in Nature Genetics, we have provided a few specific comments below, which may be of help for the authors.

We thank the Reviewer for evaluating our manuscript. Unfortunately, there appear to be several misunderstandings, which may have dampened the Reviewer's enthusiasm for our work.

First a small clarification- the reviewer mentions that "*The major novelty in this manuscript is to improve the chromatin-enrichment method*". We wish to clarify that we used standard chromatin enrichment and sequencing methods for all our analyses, and we do not claim any improvements in the methods used for chromatin enrichment.

We appropriately cite the prior work on H2BK20ac (ref. 9) and discuss the discrepancies in the available literature on H2BK20ac and other H2BNTac marks. We wish to point out that in the past >6 years, since the H2BK20ac ref. 9 was published, hardly any study has used this as an enhancer marker. We believe that the data presented in the Extended Data Fig. 18 provides an objective

assessment of the extent to which H3K27ac and H2BNTac marks are used for genome annotation. Independently of our assessment, Reviewer #4 also notes that H2BNTac is rarely used as an enhancer marker. We believe that the main reason for the historical lack of focus on H2B acetylation is that there are many discrepancies in the literature (for details on it, see **Supplemental Note 1**), and most crucially, there was no mechanistic basis known that can explain the differences in H2B acetylation and H3K27ac. The H2BNTac signature includes H2BK20ac, but this work is not just about H2BK20ac. Conceivably, singly acetylated and multiply acetylated histone tails can have very different consequences for acetyl-lysine binders and/or histone DNA interactions. We kindly request the reviewer to not equate H2BNTac with H2BK20ac (H2BNTac \neq H2BK20ac).

We feel that major novel aspects of our work may have escaped the Reviewer's attention. The major findings of our work are that it provides the first unified picture of H2BNTac in mammalian cells and reveals that all of the H2BNTac sites tested here indistinguishably mark the same genomic regions. Thus, unlike site-specific regulation and non-identical genomic occupancy of acetylation in the histones H3 and H4 tails, the sites acetylated in the H2B N-terminus are similarly regulated and occupy the same genomic regions. Contrary to prior claims, we found no solid evidence of a specific group of enhancers that lack H3K27ac and are marked by H2BK20ac. Importantly, our work goes well beyond showing that H2BNTac is an enhancer marker. We reveal the mechanistic basis of differences in H2BNTac and H3K27ac, show that a vast majority of active enhancers (defined by MPRA and eRNA) are marked by CBP/p300-catalyzed marks, and H2BNTac is a superior predictor of CBP/p300-regulated genes compared to H3K27ac. The usefulness of a good enhancer marker in the field is unequivocally demonstrated by the widespread use of H3K27ac. We sincerely hope that the reviewer will recognize the strengths of these findings, and the usefulness of H2BNTac in better defining enhancers and investigating their role in gene regulation.

Below, we provide a point-by-point response to his/her specific comments.

1. The authors start their analysis by 'quality controlling' the antibodies by ChIP-qPCR (Supplementary Table 1). The best performing antibody based on this assay is H2BK5ac (EP857Y), which the authors later choose to discard as being non-specific – mainly because it is an outlier in their comparative assays, and they find it can cross-react to H3K acetylation. Obviously, this raises the question regarding the assay presented in Supplementary Table 1 and how it can be used by the authors. Moreover, it is rather confusing to read all the comparisons in the manuscript, in which the authors use an antibody that they later show is not specific. Therefore, we suggest that the authors present their data by first performing robust assays for the specificity of the used antibodies. Here, the authors should also try to use peptide arrays derived from histones with a large number of acetylated peptides. Moreover, the authors should consider using modified nucleosomes to address specificity (see for instance PMID 30244833 for how to carefully characterize antibodies for ChIP).

In general, we would share the Reviewers' concerns about antibody specificity. We would like to clarify that what we mean by 'quality controlling' is that we tested whether the antibodies are suitable for doing ChIP or not. This was done because several of the antibodies used in our work have never been used previously for ChIP-seq. Antibodies that gave at least 5-fold enrichment over the background signal were used for ChIP-seq. The 5-fold enrichment criterion is based on the guidelines from a widely cited work (*Nat Immunol* 2011, PMID: 21934668).

We also wish to mention that the specificity of most of the antibodies used here has already been tested for a set of acetylated histone peptides, as requested by the Reviewer. Because the antibodies have already been tested using peptide arrays, we feel that repeating these analyses will not yield any further useful information. The specificity profiles of H3K11ac, H2BK16ac, and H2BK20ac (RevMab, and Abcam) are available on the manufacturer's webpage.

<https://www.abcam.com/histone-h2b-acetyl-k16-antibody-epr17598-chip-grade-ab177427.html#lb>
<https://www.abcam.com/histone-h2b-acetyl-k20-antibody-epr859-chip-grade-ab177430.html>
<https://www.revmab.com/index.php/product/anti-acetyl-histone-h2b-lys11-rabbit-monoclonal-antibody-clone-rm456/>

The specificity of H2BK16ac and H2BK20ac antibodies have also been independently confirmed in an academic lab using an acetylated histone peptide array (PMID: 26212453). These data can be viewed here:

<http://www.histoneantibodies.com/FinalArrayData/H2BK16ac/>

<http://www.histoneantibodies.com/FinalArrayData/H2BK20ac/>

For the H2BK5ac antibody (Cell Signaling Technology) specificity profiles are available on the manufacturer's website but provided in **Reviewer Fig. 2**.

The antibodies have additionally been tested by the manufacturers in immunofluorescence and/or immunoblotting with and without HDAC inhibitors. These assays have been used by other colleagues in the field for characterizing new histone modification ChIP-seq antibodies and the results have been published in well-known journals ((H3K122ac; *Nature Genetics* 2016, PMID: 27089178), (H3K79suc; *Nature* 2017, PMID: 29211711), (H3K18la; *Nature* 2019, PMID: 31645732), (H3K4me3Q5ser; *Nature* 2019, PMID: 30867594). We feel that, for the same journals, the bar for characterizing antibodies should be the same for all authors.

Testing antibodies using site-specifically acetylated nucleosomes is a good idea, but comprehensive testing will require the production of hundreds of site-specifically modified recombinant nucleosomes, harboring various PTM combinations. We are not aware of any study, including the ENCODE project, that has comprehensively validated the specificity of histone acetylation antibodies using a complete set of recombinant nucleosomes bearing all acetylation sites. Arguably, even the use of recombinant nucleosomes will not be sufficient to completely rule out the possibility that antibodies may cross-react with other related acylations or recognize acetylation differently in the context of another histone PTM or recognize cross-linked chromatin differently than purified nucleosomes. Also, there may be hundreds, if not thousands, of acetylation sites occurring in chromatin-associated proteins. In native chromatin ChIP, it would be virtually impossible to rule out that the antibodies do not enrich a non-histone acetylated target in cross-linked chromatin.

The Reviewer mentions that including cross-reactive antibodies caused confusion and "*The best performing antibody based on this assay is H2BK5ac (EP857Y), which the authors later choose to discard as being non-specific...*" We would like to clarify that we did not discard data generated from any antibody. We simply excluded the H2BK5ac (EP857Y) ChIP-seq data from defining the H2BNTac signature because of its cross-reactivity with H3K27ac.

We wish to clarify that we did not include cross-reactive antibodies in our comparisons for the purpose to diminish the prior work of our colleagues in the field. We had no prior knowledge about the antibody specificity, and the issue was only realized after noticing that the ChIP-seq profile of H2BK5ac (EP857Y) was an outlier. Alerted by this, we checked the amino acid similarity of H2BNT with other histones, which helped to uncover and explain the basis of reported cross-reactivities. There are two reasons why we did not entirely remove the data generated from the cross-reactivity antibody. (1) We felt that readers should be informed about it because most of the previously published ChIP-seq data were generated using these apparently cross-reactive antibodies. (2) Selectively excluding the data of cross-reactive antibodies would have made things appear clean, but we felt that, for transparency reasons, it is inappropriate to selectively remove data. To avoid any confusion, we explicitly state the antibodies that were excluded from defining the H2B acetylation signature.

As mentioned in the Discussion section, beyond their expected specificity in the standard antibody characterization assays, the following observations further corroborate the on-target specificity of the used H2BNTac antibodies, even if indirectly.

(1) Many histone acetylation marks have been profiled using ChIP-seq. To our knowledge, the observed genomic occupancy profile of H2BNTac sites is distinct from the known profiles of other histone acetylation marks. This indicates that the used H2BNTac antibodies are unlikely to cross-react with other well-known histone acetylation sites.

(2) A-485-induced H2BNTac reduction in our ChIP-seq is consistent with mass spectrometry-based analyses of H2BNTac sites. The only other histone sites that show similarly strong, global downregulation after CBP/p300 inhibition are a subset of sites present in histone H3 N-terminus, whose amino acid sequences and genome occupancy profiles are dissimilar from H2BNTac.

(3) Importantly, all H2BNTac antibodies consistently show differences in the relative enrichment at actively transcribed regions and distal enhancers. Transcription inhibition preferentially increased H2BNTac in actively transcribed genes, consistent with the transcription-induced exchange of H2A-H2B, but not of H3-H4. This strongly supports that antibodies likely recognize acetylated residues within histones H2A or H2B, not in H3 or H4. Because there is no sequence similarity between acetylation sites in H2A and H2B, the antibodies likely recognize acetylation sites within H2B.

(4) Because histone termini are unstructured, antibody specificity is likely primarily dictated by linear sequences, not 3D conformations. Cross-reactivity can arise if amino acids flanking modified residues are identical. Within the H2BNT, amino acids flanking lysine are highly dissimilar, making it unlikely that all monoclonal antibodies cross-react within H2BNT. Even if there is some cross-reactivity within H2BNTac sites, it will still reflect the H2BNTac signature, and not affect our conclusions.

(5) CREs marked with H2BNTac bear all the canonical features of active enhancers, including DHS, H3K4me1, H3K27ac, and MED1 binding.

(6) H2BNTac marked regions show a high validation rate in orthogonal enhancer activity assays, supporting that this marks candidate active enhancers.

In brief, it is exceedingly difficult, if not impossible, to completely rule out off-target reactivities of antibodies. The question here is whether the unique ChIP-seq signature observed in our analyses is a consequence of a lack of antibody specificity and if this impacts our conclusions. The issue of antibody specificity can be much more serious if the conclusions of a study rely on the use of a single antibody, especially if it is polyclonal. The idea that six different monoclonal antibodies, raised against 5 different H2BNTac sites, manufactured by three different companies, displaying expected on-target specificities in standard antibody characterization assays, produce an almost identical ChIP-seq signature by cross-reacting with an unknown mark appears highly unlikely to us. We sincerely believe that the distinct H2BNTac signature established here is not an artifact of antibody specificity and reflects a genuinely different occupancy profile of H2B acetylation sites.

Furthermore, the issue of antibody specificity is discussed transparently. We revised the text to acknowledge that “We cannot entirely rule out the possibility that antibodies used for establishing the H2BNTac signature may recognize some off targets.”

2. In Figure 5, the authors attempted to validate the enhancer activity of the H2B acetylated enhancers by overlapping the regions enriched by the various H2B lysine acetylated antibodies with what has been mapped as active enhancers. The degree of overlap was between 31-53%, suggesting a false negative rate of 47-69%. Interestingly, the best H2B acetylation antibodies performed better

than H3K27ac and e-RNA based methods, although H2BK20ac used throughout the manuscript (and the accompanying) only maps 45% of active enhancers. The authors should discuss the implication of these results for the use of H2BK20ac to map active enhancers.

This appears to be a misunderstanding.

The extent of overlap in this analysis depends on the size of the dataset used for comparison. The Reviewer correctly notes that less than half of H2BK20ac peaks overlap with active enhancers defined by MPRA. However, from this, it should not be interpreted that H2BNTac has a false negative rate of 47-69%. The reason for this is that the number of H2BK20ac peaks is greater than the number of MPRA⁺ATAC⁺-defined active enhancers, hence a complete overlap of H2BK20ac is not expected. An incomplete overlap likely results from the fact that many weak enhancers did not score significantly in MPRA assays. This is supported by the finding that the overlap of H2B acetylation ChIP-seq peaks with MPRA⁺ATAC⁺ enhancers is related to the intensity of H2B acetylation. Of the top 25% most abundant H2BNTac peaks, 74% overlap with MPRA⁺ATAC⁺ peaks. This is remarkable given that the library used for the MPRA analyses only covered 83% of the genome with modest (9x) coverage (i.e. the maximum expected overlap is ~83%). To our knowledge, this represents one of the highest validation rates for chromatin mark-defined enhancers in genome-scale MPRA assay.

The Reviewer may have missed noting it, but the reason for an incomplete overlap was mentioned in the text *“Of note, the total number of ATAC⁺MPRA⁺ regions is smaller than the number of H2BNTac⁺ regions, and hence, ATAC⁺MPRA⁺ is not expected to validate all H2BNTac⁺ regions.”*

3. Using a combination of H2BK20ac and/or H3K27ac the authors show that 88% of all distal ATAC+MPRA+ regions (active enhancers) overlap with these modifications (Figure 6). Based on this result it would be logical to suggest that it would be more appropriate to use H2BK20ac and H3K27ac in combination to map active enhancers, however, this is not the take home message of the manuscript. Again, it would be useful if the authors discuss this in the manuscript.

There appears to be a misunderstanding.

We agree with the reviewer that the use of H2BNTac and H3K27ac can offer complementary information in some comparisons, and we do not suggest that H3K27ac should not be used in combination with H2BNTac for mapping enhancers. Rather, Fig. 4b illustrates the usefulness of combining H2BNTac with other chromatin marks in discriminating active and inactive enhancers and promoters. This is also stated in the text *“These results demonstrate that genomic occupancy profiles of H2BNTac, H3K27ac, H3K4me3, and H3K4me1 are non-identical. While none of the marks display absolute specificity, H2BNTac complements other marks and affords valuable information in confidently discerning candidate active enhancers.”*

In the context of Fig. 6a, the combination of H2BK20ac and H3K27ac provides little added value because either of the marks is sufficient to map most of the active candidate enhancers. Nonetheless, we agree that, depending on the research question, combining both marks can be useful. If the goal was to simply map active regulatory regions, and not to discriminate active enhancers from promoters, we believe that H3K27ac is a good choice. But if the goal was to map active enhancers and discriminate from ubiquitously active promoters, or to determine the dependency of gene activation of CBP/p300 activity, then H2BNTac presents a better choice. There are also several other useful chromatin marks, including H3K4me1, H3K4me3, DHS, MED1, etc. If the resources were plentiful, we agree with the Reviewer that using multiple markers can aid in more confidently mapping enhancers and discriminating them from other CREs. Because the choice of chromatin marks depends on the research question, we feel that it is not useful to make specific

recommendations.

Reviewer #4:

Remarks to the Author:

Summary

The authors show the remarkable importance of H2B acetylation in discriminating between levels of gene activity in a way that outperforms standard H3K27ac approaches. The work has nice functional data using inhibitors of transcription/acetylation. I suspect this work will greatly increase the use of H2B acetylation among epigenetic researchers in many fields, and will enhance the precision of future research directions for those using histone marks to delineate functional roles of chromatin machinery.

Overall comments

This paper represents a major advance in understanding the modifications of histone and their relationship to enhancer activity and gene control. The work is of high interest to the field, and is technically robust.

We thank the reviewer for evaluating our work. We are pleased that the reviewer found our work of high interest. Below, we provide a point-by-point response to the Reviewer's comments.

Major points

1. Only minor criticisms were noted. Overall, I recommend reducing the number of figures to reduce redundancy, adding additional types of figure panels to increase the diversity within each figure, and I also recommend adding one or more cartoon representations of the main finding.

We agree with the Reviewer that the manuscript included considerable redundancy in figures. Following the Reviewer's recommendation, we consolidated some of the figures and removed three of the Extended Data figures (Figs. 10, 12, and 15), which we found largely redundant. In response to the Reviewers' comments, we accommodated several new figures; the total number of figures is reduced by 2). We kindly request the Reviewer for his/her understanding if some of the new figures may appear somewhat redundant. We added a cartoon diagram (**Fig. 8c**) showing the relative specificities of chromatin marks for enhancers versus promoters.

Novelty/notability claims

2. The paper's novelty is strong. A literature survey indicated that, indeed, the genomic locations of H2B acetylation have not been (to my surprise!) elsewhere shown.

We fully concur with the Reviewer's assessment, and sincerely believe that our work will encourage other researchers to include H2B acetylation sites for mapping enhancers and will spur further research into these relatively understudied modifications.

Minor points

3. Minor grammatical errors have been noted in an attached PDF of the manuscript via comments/notes.

We thank the Reviewer for pointing out the errors, which are corrected in the revised version.

4. The authors discuss the role of H2B acetylation in relationship to chromHMM defined chromatin. It would be really nice to see the “emission” data and defined chromatin states with and without H2B acetylation. Here is an example:

Unfortunately, the suggested example did not come through the system. Regardless, we used ChromHMM to predict chromatin states, with or without including H2BNTac (**Extended Data Fig. 10**). ChromHMM states are predicted computationally, but annotation of the predicted states is done manually, and we acknowledge that this can be somewhat subjective. Overall, the predicted ChromHMM states, with or without H2BNTac, appear very similar to ours. The most notable difference is that, if H2BNTac is included in the model, ChromHMM predicts a single state with strong H3K4me3. However, without H2BNTac, ChromHMM predicts two different states with almost equally strong H3K4me3, and these two states are distinguished by H3K4me1 (H3K4me3⁺H3K4me1⁻; H3K4me3⁺H3K4me1⁻). This confirms our observation that H2BNTac and H3K27ac differ most notably in the promoter regions.

5. There is a glaring lack of genome browser tracks that are need in the main figures to reveal the quality of the ChIP-seq data, and to assist in giving readers an intuition of the patterns being claimed through boxplots and correlation matrices. To fix this, the authors should promote Extended Figure 8b to the main figure set, or make a new genome browser figure that shows the differences in H3 acetylation and H2B acetylation at diverse regulatory elements. Another example is in the CDK inhibition data – the increase of H2B acetylation after inhibition of transcription is beautifully shown in Ext Data Figure 10C – perhaps 1 of these gene can be moved to the main figure 3, and some of the intellectually redundant violin plots can be moved to supplementary.

Following the reviewer’s suggestion, we include example gene tracks to show the differences in H3K27ac and H2B acetylation (**Figs. 1c, 3d**).

6. The role of H3K v H2B acetylation in predicting response to A485 is striking. The authors should add a metagene/average plot of the histone marks in Figure 7b, centered around the TSS of genes that are strongly downregulated, and compared to the same data plotting at the TSS of genes that are non-responsive.

We included an average plot of H3K27ac, H2BNTac, and H3K4me3 at promoters of genes that are downregulated by A-485, and genes that are non-response to A-485 (**Fig. 7c**).

Figure comments

7. Figure 5 and Figure 6 were swapped, causing some confusion for this reviewer.

We regret this error and apologize for it. It is rectified in the revised version.

8. Figure 6a should have H3K27ac as an additional row of plots with all the same measurements calculated.

In this figure, the sum of “H3K27ac” and “H3K27ac+H2BNTac” peaks present the total fraction of H3K27ac peaks overlapping with MPRA peaks. If the Reviewer finds it unclear, we will add a separate panel for H3K27ac only.

Figures for Reviewers

Reviewer Fig. 1. H3K27ac and H2BNTac distinctly correlate with other chromatin marks in promoter regions. a-b, Genome-wide correlation among the indicated chromatin marks.

Correlations (Spearman's ρ , panel **a**; and Pearson's r , panel **b**) are calculated between the indicated marks using 2kb-sized genomic windows. In promoter regions, H2BNTac poorly correlates with H3K27ac, H3K9ac, H3K27me3, and MED1. However, in intergenic regions, the correlation patterns appear not as expected, and non-peak regions are included in determining correlation in intergenic regions.

Reviewer Fig. 2. Specificity profile of H2BK5ac antibody.

Acknowledgment: This figure was shared by Cell Signaling Technology.

a

Spearman correlation

b

Pearson's correlation

Decision Letter, first revision:

30th Nov 2022

Dear Chuna,

Thank you for submitting your revised manuscript entitled "A unique H2B acetylation signature marks active enhancers and predicts CBP/p300 target genes" (NG-A60574R1). It has now been seen by the original referees and their comments are below. Overall, the reviewers find that the paper has improved in revision, and therefore we'll be happy in principle to publish it in Nature Genetics, pending minor revisions to satisfy the referees' final requests (textually/analytically, not experimentally) and to comply with our editorial and formatting guidelines.

The current version of your manuscript is in a PDF format. Please email us (natgen@us.nature.com) a copy of the file in an editable format (Microsoft Word) - we can not proceed with PDFs at this stage.

We will then be performing detailed checks on your paper and will send you a checklist detailing our editorial and formatting requirements afterwards. Please do not upload the final materials and make any revisions until you receive this additional information from us.

Thank you again for your interest in Nature Genetics. Please do not hesitate to contact me if you have any questions.

Congratulations!

Sincerely,

Tiago

Tiago Faial, PhD
Chief Editor
Nature Genetics
<https://orcid.org/0000-0003-0864-1200>

Reviewer #1 (Remarks to the Author):

I am reasonably satisfied with the authors' changes.

Reviewer #2 (Remarks to the Author):

The revised manuscript is significantly improved, especially by removing or rephrasing some of the most highly overblown claims. I have to admit that I am not entirely convinced by certain arguments made by authors in the response to my previous comments. I agree that the distinctions between enhancers and promoters are sometimes blurred and that the effects of p300/CBP inhibition on enhancers vs promoters cannot be easily disentangled. But... this gives even more reason to be cautious with interpretation of the data at hand. Contrary to authors claims, however, this is not entirely untestable hypothesis: one experiment that could at least formally demonstrate that activity of most enhancers is directly affected by A-485 (and would not be confounded by the genomic distance effects) would be to perform enhancer MPRA assays such as STARR-seq with and without the inhibitor, using a promoter whose basal activity is broad and not sensitive to p300/CBP (and such promoters have been identified). This would at least show that in principle, autonomous activity of most enhancers is dependent on p300/CBP and one could even test in such setup if the enhancers with relatively higher level of H2BNTac are more affected. Conversely, the activity of different promoters could be tested in a similar assay in the presence or absence of the inhibitor, in the context of presence or absence of an enhancer that is independent of p300/CBP. This could address to what extent basal and enhancer-activated transcription from these diverse promoters is dependent on p300/CBP. In the absence of such direct functional experiments, the authors need to stick to the claims that are supported by the data and emphasize the relevant caveats. Furthermore, while I completely agree with the authors that the observation that 'gene dysregulation correlates better with the decrease in H3K27ac at promoters than at enhancers, is insufficient to establish that p300 activates genes independently of enhancers', neither is authors' own data sufficient to establish the opposite conclusion.

Similarly, some of the statements on the ABC model in response to my comments are somewhat of a stretch. For example: 'as expected, we would not expect a large difference in the performance of H3K27ac and H2BNTac. This is because H3K27ac and H2BNTac are similarly abundant in the distal regions, and the ABC model mainly uses H3K27ac intensity in distal regions.' First, the original description of the ABC model in fact did not exclude promoter H3K27ac (although this was pointed out as a possible weakness, as promoters might provide an undue input into the score). With regards to authors' work, it is easily testable if the difference in the performance of the two marks in the ABC model increases in the manner dependent on the inclusion or exclusion of the promoter from the ABC calculations. Second, the model is quantitative, in that it does not just rely on the mark being or not being there at most distal elements, but on its relative levels, which is incorporated to the model as 'activity'. Third, if there would be a large difference in enhancer prediction abilities between H3K27ac and H2BNTac, this should translate into a substantially better performance in the ABC model. The marginally better performance of H2BNTac suggest that the predictive difference is not large. Nonetheless, this does not detract from the authors observations that H2BNTac is a novel mark associated with enhancers, and as such this is interesting for the field.

Reviewer #3 (Remarks to the Author):

In the revised version of the manuscript, the authors have provided additional experimental controls for the specificity of the H2BNTac antibodies used and more detailed analysis of the included data.

The demonstration that multiple H2BNTac modifications are more precise markers than H3K27ac for

active enhancers will be useful for the scientific community, and it should lead to a shift from using H3K27ac in mapping active enhancers to using H2BNTac. The authors have also demonstrated the benefits of employing H2BK20ac, probably the best H2BNTac site for predicting active enhancers. Could the authors elaborate more whether H2BK20ac by itself would be sufficient for active enhancer mapping or the entire panel of H2BNTac should be used?

Reviewer #4 (Remarks to the Author):

The authors have completely addressed all the concerns I raised, as well as concerns from the other referees. I am in full support of publishing these exciting findings!

Author Rebuttal, first revision:

See inserted PDF

A point-by-point response to Reviewer's comments

Reviewer #1

I am reasonably satisfied with the authors' changes.

We thank the reviewer for evaluating the revised manuscript. We are pleased to know that the reviewer is satisfied with our responses.

Reviewer #2:

The revised manuscript is significantly improved, especially by removing or rephrasing some of the most highly overblown claims. I have to admit that I am not entirely convinced by certain arguments made by authors in the response to my previous comments. I agree that the distinctions between enhancers and promoters are sometimes blurred and that the effects of p300/CBP inhibition on enhancers vs promoters cannot be easily disentangled. But... this gives even more reason to be cautious with interpretation of the data at hand. Contrary to authors claims, however, this is not entirely untestable hypothesis: one experiment that could at least formally demonstrate that activity of most enhancers is directly affected by A-485 (and would not be confounded by the genomic distance effects) would be to perform enhancer MPRA assays such as STARR-seq with and without the inhibitor, using a promoter whose basal activity is broad and not sensitive to p300/CBP (and such promoters have been identified). This would at least show that in principle, autonomous activity of most enhancers is dependent on p300/CBP and one could even test in such setup if the enhancers with relatively higher level of H2BNTac are more affected. Conversely, the activity of different promoters could be tested in a similar assay in the presence or absence of the inhibitor, in the context of presence or absence of an enhancer that is independent of p300/CBP. This could address to what extent basal and enhancer-activated transcription from these diverse promoters is dependent on p300/CBP. In the absence of such direct functional experiments, the authors need to stick to the claims that are supported by the data and emphasize the relevant caveats. Furthermore, while I completely agree with the authors that the observation that 'gene dysregulation correlates better with the decrease in H3K27ac at promoters than at enhancers, is insufficient to establish that p300 activates genes independently of enhancers', neither is authors' own data sufficient to establish the opposite conclusion.

We thank the reviewer for evaluating the revised manuscript and for providing further feedback.

In the revised manuscript, we do not claim that CBP/p300 activates genes through enhancers or promoters. We do not rule out the possibility that CBP/p300 can directly activate certain types of promoters, without involving enhancers. In future work, it would be interesting to test the requirement of CBP/p300 for different types of enhancers and promoters in MPRA, but we feel that this goes beyond the scope of this work. The focus of this work is to demonstrate the specificity of H2BNTac.

Similarly, some of the statements on the ABC model in response to my comments are somewhat of a stretch. For example: 'as expected, we would not expect a large difference in the performance of H3K27ac and H2BNTac. This is because H3K27ac and H2BNTac are similarly abundant in the distal regions, and the ABC model mainly uses H3K27ac intensity in distal regions.' First, the original

description of the ABC model in fact did not exclude promoter H3K27ac (although this was pointed out as a possible weakness, as promoters might provide an undue input into the score). With regards to authors' work, it is easily testable if the difference in the performance of the two marks in the ABC model increases in the manner dependent on the inclusion or exclusion of the promoter from the ABC calculations. Second, the model is quantitative, in that it does not just rely on the mark being or not being there at most distal elements, but on its relative levels, which is incorporated to the model as 'activity'. Third, if there would be a large difference in enhancer prediction abilities between H3K27ac and H2BNTac, this should translate into a substantially better performance in the ABC model. The marginally better performance of H2BNTac suggest that the predictive difference is not large. Nonetheless, this does not detract from the authors observations that H2BNTac is a novel mark associated with enhancers, and as such this is interesting for the field.

We thank the reviewer for his/her comment.

Firstly, the ABC model description in the Github states that "*EnhancerPredictions.txt: all element-gene pairs with scores above the provided threshold. Only includes expressed genes and does not include promoter elements. This file defines the set of 'positive' predictions of the ABC model.*" <https://github.com/broadinstitute/ABC-Enhancer-Gene-Prediction>

From this, we understand that promoter H3K27ac signal is not excluded for calculating the ABC score, but promoter elements are filtered from further analyses (i.e., promoter elements are not considered enhancers).

Secondly, H3K27ac is not the sole determinant of the ABC score, and according to a recent pre-print, removing H3K27ac only modestly decreases the performance of the ABC model (<https://www.biorxiv.org/content/10.1101/2022.01.28.478202v4.full>). If removing H3K27ac only modestly reduces the ABC model's performance, it indicates that the relative importance of H3K27ac to the ABC model's prediction performance is modest. Because more than one parameter contributes to the ABC score, just substituting H3K27ac with another mark will not be expected to dramatically improve the model's performance.

Thirdly, in our analyses, the performance of the ABC model was affected much more strongly by the type of CRISPRi dataset used than the use of H3K27ac or H2BNTac. We agree that it would be interesting to further compare the performance of H3K27ac and H2BNTac in the context of predicting enhancer targets (with or without the ABC model), but this will require the generation of additional enhancer-gene target datasets.

Finally, we do not claim that H2BNTac performs much better than H3K27ac in the context of the ABC model. As the Reviewer notes, even if H2BNTac does not perform better than H3K27ac in the context of the ABC model, it does not diminish the novelty of our main finding – that H2BNTac is a useful marker for identifying active enhancers and predicting CBP/p300 regulated genes.

Reviewer #3

In the revised version of the manuscript, the authors have provided additional experimental controls for the specificity of the H2BNTac antibodies used and more detailed analysis of the included data.

The demonstration that multiple H2BNTac modifications are more precise markers than H3K27ac for active enhancers will be useful for the scientific community, and it should lead to a shift from using H3K27ac in mapping active enhancers to using H2BNTac. The authors have also demonstrated the

benefits of employing H2BK20ac, probably the best H2BNTac site for predicting active enhancers. Could the authors elaborate more whether H2BK20ac by itself would be sufficient for active enhancer mapping or the entire panel of H2BNTac should be used?

We thank the reviewer for evaluating the revised manuscript, and for his/her positive comments.

The analyses presented in this work show that different H2BNTac sites similarly mark enhancers, and previous work indicates that these sites are targeted by CBP/p300 (Weinert, Narita et al., *Cell* 2018. PMID: 30837475). From this, we conclude that one of the H2BNTac sites is sufficient for mapping candidate enhancers, and including multiple H2BNTac sites provides little additional value. The reason we chose H2BK20ac as a representative H2BNTac mark in some of our analyses are (1) there are two commercially available ChIP quality monoclonal antibodies for this mark, (2) in proteomic analyses, H2BK20ac appears to be a bit more abundant than some of the other H2BNTac sites (Hansen et al., *Nat Commun* 2019. PMID: 30837475), possibly because the sites occurring at the end of H2B N-terminus are deacetylated with slower kinetics (Extended Data Fig. 1, and Weinert, Narita et al., *Cell* 2018. PMID: 30837475). For mapping active candidate enhancers, in our view, it is not necessary to use the entire panel of H2BNTac antibodies.

Reviewer #4

The authors have completely addressed all the concerns I raised, as well as concerns from the other referees. I am in full support of publishing these exciting findings!

We thank the reviewer for evaluating the work and for supporting the publication of the work.

Final Decision Letter:

23rd Feb 2023

Dear Chuna,

I am delighted to say that your manuscript "Acetylation of histone H2B marks active enhancers and predicts CBP/p300 target genes" has been accepted for publication in an upcoming issue of Nature Genetics.

Your paper will be published online after we receive your corrections and will appear in print in the next available issue. You can find out your date of online publication by contacting the Nature Press Office (press@nature.com) after sending your e-proof corrections. Now is the time to inform your Public Relations or Press Office about your paper, as they might be interested in promoting its publication. This will allow them time to prepare an accurate and satisfactory press release. Include your manuscript tracking number (NG-A60574R2) and the name of the journal, which they will need when they contact our Press Office.

Please note that *Nature Genetics* is a Transformative Journal (TJ). Authors may publish their research with us through the traditional subscription access route or make their paper immediately open access through payment of an article-processing charge (APC). Authors will not be required to make a final decision about access to their article until it has been accepted. [Find out more about Transformative Journals](https://www.springernature.com/gp/open-research/transformative-journals)

Authors may need to take specific actions to achieve [compliance with funder and institutional open access mandates](https://www.springernature.com/gp/open-research/funding/policy-compliance-faqs). If your research is supported by a funder that requires immediate open access (e.g. according to [Plan S principles](https://www.springernature.com/gp/open-research/plan-s-compliance)) then you should select the gold OA route, and we will direct you to the compliant route where possible. For authors selecting the subscription publication route, the journal's standard licensing terms will need to be accepted, including [self-archiving and license to publish](https://www.nature.com/nature-portfolio/editorial-policies/self-archiving-and-license-to-publish). Those licensing terms will supersede any other terms that the author or any third party may assert apply to any version of the manuscript.

Please note that Nature Portfolio offers an immediate open access option only for papers that were first submitted after 1 January, 2021.

Sincerely,

Tiago

Tiago Faial, PhD
Chief Editor
Nature Genetics
<https://orcid.org/0000-0003-0864-1200>